# *MOONSHOT*: A Framework for Multi-Objective Pruning of Vision and Large Language Models

**Gabriel Afriat**  *afriatg@mit.edu*
*Operations Research Center*
*Massachusetts Institute of Technology*

**Xiang Meng**  *mengx@mit.edu*
*Operations Research Center*
*Massachusetts Institute of Technology*

**Shibal Ibrahim** *  *shibal@google.com*
*Google*

**Hussein Hazimeh** [†]  *hh@ieee.org*
*OpenAI*

**Rahul Mazumder**  *rahulmaz@mit.edu*
*Sloan School of Management,*
*Operations Research Center*
*and MIT Center for Statistics*
*Massachusetts Institute of Technology*

**Reviewed on OpenReview:** *https://openreview.net/forum?id=Ew9s7veEQU*

## Abstract

Weight pruning is a common technique for compressing large neural networks. We focus on the challenging post-training one-shot setting, where a pre-trained model is compressed without any retraining. Existing one-shot pruning methods typically optimize a single objective, such as a layer-wise reconstruction loss or a second-order Taylor approximation of the training loss. We highlight that neither objective alone is consistently the most effective across architectures and sparsity levels. Motivated by this insight, we propose *MOONSHOT*, a general and flexible framework that extends any single-objective pruning method into a multi-objective formulation by jointly optimizing both the layer-wise reconstruction error and second-order Taylor approximation of the training loss. *MOONSHOT* acts as a wrapper around existing pruning algorithms. To enable this integration while maintaining scalability to billion-parameter models, we propose modeling decisions and introduce an efficient procedure for computing the inverse Hessian, preserving the efficiency of state-of-the-art one-shot pruners. When combined with state-of-the-art pruning methods on Llama-3.2 and Llama-2 models, *MOONSHOT* reduces C4 perplexity by up to 32.6% at 2:4 sparsity and improves zero-shot mean accuracy across seven classification benchmarks by up to 4.9 points. On Vision Transformers, it improves accuracy on ImageNet-1k by over 5 points at 70% sparsity, and on ResNet-50, it yields a 4-point gain at 90% sparsity. Our code is available at `https://github.com/mazumder-lab/MOONSHOT`.

---

[*]Work done while at MIT (Department of Electrical Engineering and Computer Science).
[†]Work done while at Google Research.

# 1 Introduction

Contemporary vision and language models have huge parameter counts (He et al., 2016; Dosovitskiy et al., 2021; Zhang et al., 2022), incurring significant computational costs during the inference phase. Pruning is a common strategy for compressing large neural networks. The aim is to remove a subset of weights by setting them to zero while maintaining relatively high predictive performance. Pruning can be either a) unstructured, where any individual weight can be set to zero (Han et al., 2015; Benbaki et al., 2023; Frantar et al., 2022; Kuznedelev et al., 2023; Frantar & Alistarh, 2023; Sun et al., 2024), b) structured, where entire rows and columns are set to zero (Ma et al., 2023; Meng et al., 2024b) or c) semi-structured where specific patterns are enforced, such as n:m sparsity, where n weights are set to zero within each block of m weights (Frantar et al., 2022; Kuznedelev et al., 2023; Frantar & Alistarh, 2023; Sun et al., 2024). In this work, we focus on these three different compression modes.

Various techniques have been proposed for pruning vision and large language models (Han et al., 2015; Frankle & Carbin, 2019; Yu et al., 2022; Frantar et al., 2022; Benbaki et al., 2023; Kuznedelev et al., 2023; Frantar & Alistarh, 2023; Meng et al., 2024a; Sun et al., 2024). Many existing methods rely on gradual pruning, where the model is fine-tuned on the original loss after every pruning stage to recover accuracy. However, for billion-scale models, such fine-tuning can be extremely expensive. In this context, recent works (Frantar & Alistarh, 2022; Frantar et al., 2022; Benbaki et al., 2023; Kuznedelev et al., 2023) have focused on the challenging task of *post-training* pruning in one-shot i.e., compressing a model without retraining based on a small amount of calibration data. In this paper, we focus on post-training one-shot pruning approaches, which are computationally attractive and particularly relevant for real-world applications.

When pruning a pre-determined fraction of the weights, various criteria are employed to preserve model accuracy or perplexity as much as possible, each leading to a different performance-sparsity trade-off. For example, weight magnitudes can be used as a criterion to decide which weights to prune and which to keep (Hanson & Pratt, 1988; Mozer & Smolensky, 1989; Gordon et al., 2020). However, magnitude-based pruning approaches rely extensively on expensive retraining to minimize the loss in performance. Another popular type of approach uses a local quadratic approximation of the original training loss to estimate the reduction in model performance. These approaches then approximately minimize this objective while imposing a sparsity constraint. This idea was introduced by LeCun et al. (1989b); Hassibi & Stork (1992b) through the Optimal Brain Surgeon (OBS) framework and built upon by various methods (Singh & Alistarh, 2020b; Frantar et al., 2021; Yu et al., 2022; Benbaki et al., 2023; Kuznedelev et al., 2023). A third prevalent criterion is based on the layer-wise OBS strategy (Dong et al., 2017; Frantar et al., 2022; Frantar & Alistarh, 2023; Sun et al., 2024; Meng et al., 2024b). In this approach, the pruning task is divided into layer-wise subproblems. For each layer, the goal is to minimize the squared reconstruction error between the original and pruned layer outputs subject to a sparsity constraint. While the OBS objective uses global information from the training loss of the pre-trained neural network to guide pruning, the layer-wise reconstruction loss uses more localized information in the embedding spaces.

To better understand the impact of pruning criteria on performance, we conducted a series of experiments across both vision and language models. On Vision Transformers, we evaluated CAP (Kuznedelev et al., 2023), which minimizes a second-order Taylor approximation of the training loss. On a convolutional neural network (ResNet-50 (He et al., 2015)), we considered OBC (Frantar et al., 2022), which uses the layer-wise reconstruction loss. For large language models, we evaluated SparseGPT (Frantar & Alistarh, 2023) and Wanda (Sun et al., 2024), both of which are designed around the layer-wise reconstruction objective. These methods have support for both unstructured and semi-structured pruning. To isolate the effect of the pruning criterion, we adapted each method to operate under the opposite objective: we evaluated CAP using the layer-wise reconstruction error, and OBC, CAP and Wanda using the second-order Taylor approximation of the training loss. These comparisons, illustrated in Table 1, revealed that neither criterion is uniformly superior. Depending on the architecture, pruning method, and sparsity level, either the layer-wise reconstruction error or second-order Taylor approximation of the training loss objective may yield better results. In several cases, the pruning methods performed better when paired with the objective they were not originally designed to optimize.

Table 1: Comparison between the second-order Taylor approximation of the training loss and the layer-wise reconstruction error objectives across different pruning methods, models and sparsity regimes. We either keep the original objective, indicated with a star *, as pruning criterion (approximation of the training loss for CAP and layer-wise reconstruction loss for OBC, Wanda and SparseGPT), or we replace it with the alternative single-objective criterion.

| Domain | Model | Method | Sparsity | Second-Order Taylor Approx. of Training Loss | Layer-Wise Reconst. Error |
|---|---|---|---|---|---|
| Language models C4 perplexity (↓) | Llama-3.2-1B | SparseGPT | 0.50 | $29.14 \pm 0.16$ | $\mathbf{27.15}^{*} \pm 0.23$ |
| | | Wanda | 0.50 | $\mathbf{30.48} \pm 0.14$ | $35.71^{*} \pm 0.21$ |
| | | Wanda | 0.60 | $\mathbf{88.87} \pm 1.62$ | $117.71^{*} \pm 0.87$ |
| | Llama-3.2-3B | SparseGPT | 0.50 | $18.12 \pm 0.06$ | $\mathbf{17.61}^{*} \pm 0.08$ |
| | | Wanda | 0.50 | $\mathbf{18.2} \pm 0.04$ | $18.88^{*} \pm 0.03$ |
| | | Wanda | 0.60 | $\mathbf{40.28} \pm 0.67$ | $41.98^{*} \pm 0.4$ |
| Vision models ImageNet-1k accuracy (↑) | DeiT-Tiny | CAP | 0.60 | $\mathbf{62.28}^{*} \pm 0.05$ | $54.18 \pm 0.15$ |
| | | | 2:4 | $\mathbf{52.28}^{*} \pm 0.04$ | $47.65 \pm 0.11$ |
| | DeiT-Small | CAP | 0.50 | $\mathbf{77.27}^{*} \pm 0.03$ | $76.56 \pm 0.04$ |
| | | | 2:4 | $69.65^{*} \pm 0.02$ | $\mathbf{70.25} \pm 0.04$ |
| | ResNet-50 | OBC | 0.50 | $50.88 \pm 25.39$ | $\mathbf{76.63}^{*} \pm 0.05$ |
| | | | 0.70 | $48.94 \pm 24.42$ | $\mathbf{74.73}^{*} \pm 0.03$ |

This indicates that *the two objectives capture complementary signals of parameter importance*, and relying on one alone can lead to suboptimal pruning decisions. Motivated by this insight, we propose a novel multi-objective optimization framework that jointly minimizes both the layer-wise reconstruction objective and second-order approximation of the training loss. This multi-objective optimization consistently improves the performance-sparsity trade-off of various state-of-the-art methods.

Extending state-of-the-art pruning methods to a multi-objective formulation introduces new challenges. These pruning algorithms typically approximate the objective using a quadratic form involving the Hessian of the model weights and require computing (or approximating) its inverse (Singh & Alistarh, 2020a; Frantar et al., 2022; Frantar & Alistarh, 2023; Kuznedelev et al., 2023; Sun et al., 2024; Meng et al., 2024b). To make this calculation efficient, these methods rely on approximations or exploit the Hessian structure, typically using a block-diagonal approximation. However, the Hessians associated with the layer-wise reconstruction loss and the second-order Taylor approximation of the training loss exhibit different structures, and existing algorithms adopt distinct block-diagonal formulations depending on the specific objective. As a result, combining the Hessians from different objectives directly impacts the block-diagonal approximation, introducing new challenges in adapting the single-objective pruning methods. In *MOONSHOT*, we propose some modeling decisions (as described in Figure 2) to adapt the existing algorithms to the new multi-objective formulation.

While a relatively straightforward adaptation is possible for smaller architectures, additional computational complexity appears in large-scale models. SparseGPT (Frantar & Alistarh, 2023) and OSSCAR (Meng et al., 2024b), for instance, are state-of-the-art pruning methods for one-shot pruning of LLMs, which minimize the layer-wise reconstruction error. SparseGPT has support for both unstructured and semi-structured pruning while OSSCAR is designed for structured pruning. Both methods achieve their efficiency by exploiting the structure of the layer-wise reconstruction objective: the Hessian for each layer is naturally block-diagonal, with each block corresponding to the Hessian of a row in the weight matrix. Notably, all the blocks are identical and depend only on the input data. This structure allows for a very efficient computation of both the Hessian and its inverse. In contrast, *MOONSHOT* combines the reconstruction loss with a second-order approximation of the training loss, resulting in a Hessian that is no longer block-diagonal and particularly large, especially for models with billions of parameters. Even imposing a block-diagonal approximation on the Hessian of the multi-objective formulation is insufficient: since the blocks along the diagonal differ, inverting the Hessian requires computing many more matrix inversions, and a naive computation quickly becomes prohibitively expensive in practice. To address these challenges, we develop an efficient method to scale the multi-objective formulation to modern large architectures. In particular, we propose a fast approximate method for computing the Hessian and its inverse, enabling compatibility with high-performance state-of-the-art pruning methods such as SparseGPT and OSSCAR.

Our framework is very flexible and can handle different pruning patterns. *MOONSHOT* can be used for any of the sparsity patterns supported by the underlying single-objective baseline. In this work, we consider unstructured, semi-structured 2:4, and structured sparsity. Unstructured pruning offers strong memory savings and can yield speedups on CPUs (NeuralMagic, 2021) and specialized hardware accelerators (Han et al., 2015; Dave et al., 2021), but typically requires high sparsity ratios to achieve speedups on GPUs (Gale et al., 2020), often at the cost of model performance. In contrast, n:m sparsity also enables efficient execution on modern GPUs even at moderate sparsity levels (Mishra et al., 2021), making it particularly well-suited for the sparsity regimes commonly used in large language models (Frantar & Alistarh, 2023). Finally, structured pruning yields direct speedups on GPUs and CPUs (Kurtic et al., 2023; Meng et al., 2024b), but is typically applied at much lower sparsity ratios, since maintaining accuracy becomes increasingly difficult at higher structured sparsity levels.

*MOONSHOT* is orthogonal to the other existing methods designed to improve single-objective pruning. In particular, prior works (Frantar et al., 2022; Kuznedelev et al., 2023; Lu et al., 2024; Yin et al., 2024) have shown that, in the case of unstructured pruning, a more principled distribution of the sparsity budget across layers can improve the performance of single-objective pruning approaches. In vision models, CAP (Kuznedelev et al., 2023) improves top-1 accuracy of DeiT-Tiny on ImageNet-1k by nearly 10 points (a relative gain of 22%) at 70% sparsity with non-uniform sparsity allocation. These improvements are even more critical for large language models, which typically suffer severe performance degradation when pruned beyond 50% sparsity unless the sparsity is distributed non-uniformly across layers. For example, Yin et al. (2024) report that, with OWL, a carefully optimized layer-wise sparsity allocation, it is possible to achieve 71.38% lower perplexity on WikiText using Wanda (Sun et al., 2024) at 70% sparsity. We show that when our method, *MOONSHOT*, is combined with non-uniform sparsity allocation strategies, we achieve additional improvements in the performance of the pruned model. On Llama-3.2-1B and Llama-3.2-3B across both SparseGPT and Wanda pruning baselines, *MOONSHOT* reduces C4 perplexity by up to an additional 25% and improves zero-shot mean accuracy by up to 1 additional point in comparison to the baselines with non-uniform sparsity allocation alone.

**Contributions.** We propose a novel optimization-based framework, which extends existing single-objective pruning approaches to a multi-objective formulation, enabling improved accuracy-sparsity trade-offs in the post-training one-shot pruning setting. Our contributions can be summarized as given below.

- We show that the layer-wise reconstruction loss and second-order Taylor approximation of the training loss result in different sparsity-accuracy trade-offs across architectures and sparsity levels. To the best of our knowledge, this is the first work to highlight those differences and systematically compare these two pruning objectives side by side.
- Motivated by this insight, we introduce a novel multi-objective optimization formulation to simultaneously minimize two objectives: a local quadratic approximation of the training loss and the layer-wise reconstruction error, subject to sparsity constraints. While these objectives have been considered in isolation, considering them simultaneously is new. Our framework, *MOONSHOT* (**M**ulti-**O**bjective **ON**e-**SHOT** pruning), provides a principled extension to existing single-objective pruning approaches, enabling them to operate under a multi-objective formulation.
- We introduce a set of modeling choices and algorithmic adaptations that extend single-objective pruning methods to a multi-objective setting. For applications to large language models, we propose an efficient procedure for computing the inverse Hessian in our multi-objective formulation. This fast computation is essential for preserving the scalability of existing pruning methods. Our implementation of *MOONSHOT*-SparseGPT prunes Llama-3.2-3B in under 40 minutes and Llama-3.2-1B in 8 minutes on a single GPU, showing that the multi-objective formulation remains efficient at the relevant billion-parameter scale.
- We validate our proposed method across diverse domains and applications:

  (i) We evaluate *MOONSHOT* on large language models, including Llama-3.2-1B, Llama-3.2-3B (Grattafiori et al., 2024) and Llama-2-13b-chat-hf (Touvron et al., 2023). *MOONSHOT* improves the performance of state-of-the-art pruning methods such as SparseGPT (Frantar & Alistarh, 2023) and Wanda (Sun et al., 2024) in the post-training one-shot setting, under (a) unstructured sparsity (including non-uniform allocations via OWL (Yin et al., 2024) and AlphaPruning (Lu et al., 2024)), (b) semi-structured $n{:}m$ sparsity. It also improves OSSCAR (Meng et al., 2024b) in the structured

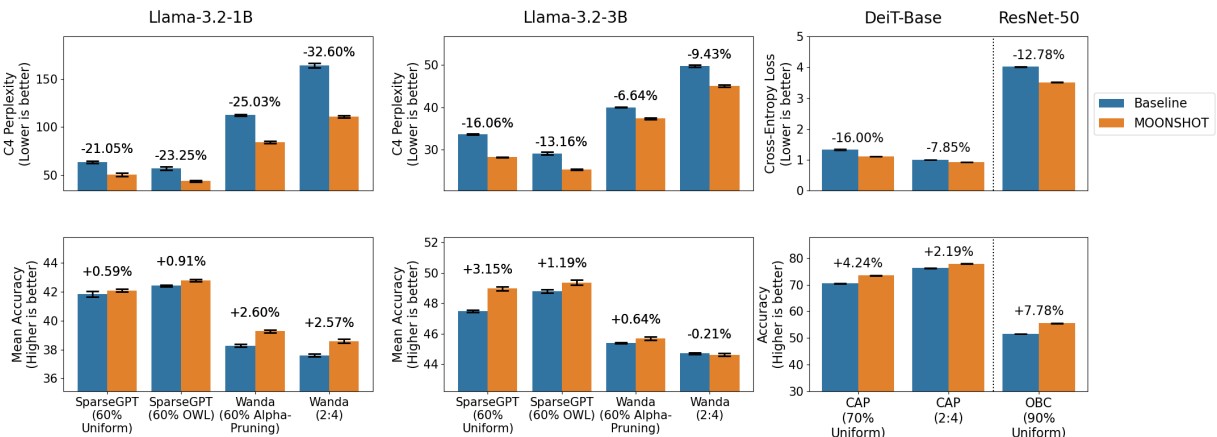

Figure 1: Impact of *MOONSHOT* on SparseGPT/Wanda (Llama-3.2) and CAP/OBC (DeiT-Base, ResNet-50) across sparsity regimes. For vision models, mean cross-entropy and ImageNet-1k accuracy are reported; for LLMs, perplexity on C4 along with mean zero-shot accuracy over seven classification tasks. Results are averaged over three seeds with standard errors.

pruning case. On Llama-3.2-1B and 3B, *MOONSHOT* reduces C4 test perplexity (Raffel et al., 2020) of Wanda by up to 32.6% at 2:4 sparsity, achieves over 20% perplexity reduction for both SparseGPT and Wanda at 60% and 2:4 sparsity, and improves the mean accuracy across seven classification benchmarks by up to 1.5 points (see Figure 1 and Table 3). At 10% structured sparsity, *MOONSHOT* reduces C4 perplexity by up to 11% and improves the mean accuracy by up to 4.9 points. For Llama-2-13b-chat-hf, *MOONSHOT* reduces C4 perplexity by up to 14% and similarly improves the mean accuracy by up to 1.5 points at 70% unstructured sparsity (see Table 3).

(ii) We also evaluate *MOONSHOT* on computer vision benchmarks, including Vision Transformers (DeiT-Tiny, DeiT-Small, and DeiT-Base) and a convolutional model (ResNet-50). Across these models, our approach improves state-of-the-art methods such as CAP (Kuznedelev et al., 2023) and OBC (Frantar et al., 2022) in the unstructured and $n:m$ sparsity regimes. In particular, on ImageNet-1k (Deng et al., 2009), it improves CAP by over 5 points in accuracy at 70% sparsity and 2 points at 2:4 sparsity, and improves OBC by 4 points at 90% sparsity (see Figure 1 and Table 2).

## 2 Multi-Objective Pruning

Our framework aims to improve existing single-objective network pruning methods by considering both the layer-wise reconstruction error and second-order approximation of the training loss. We first introduce these single-objective loss functions. We then formulate a new multi-objective optimization problem and propose *MOONSHOT*, which adapts state-of-the-art algorithms to minimize this new objective. To maintain the efficiency of the original LLM pruning methods, we finally propose a highly efficient approach for computing the inverse Hessian, a key component for algorithms like SparseGPT (Frantar & Alistarh, 2023) and OSSCAR (Meng et al., 2024b).

### 2.1 Layer-wise pruning objectives

Consider the task of pruning a neural network with $L$ layers. For any given layer $l \in [L]$, layer-wise pruning approaches (Nagel et al., 2020; Frantar et al., 2022; Kuznedelev et al., 2023; Frantar & Alistarh, 2023; Sun et al., 2024) aim to zero out some parameters and potentially adjust the remaining weights in the $l$-th layer to minimize the performance drop as much as possible. More formally, given the original pre-trained weights in the $l$-th layer, layer-wise pruning targets the following discrete optimization problem:

$$\min_{W^{(l)}} \quad \mathcal{L}(W^{(l)}, \widehat{W}^{(l)}) \quad \text{s.t.} \quad \mathcal{S}(W^{(l)}) \leq S^{(l)},\tag{1}$$

where $\mathcal{S}(W^{(l)}) \leq S^{(l)}$ denotes the sparsity constraint, which depends on the sparsity type (unstructured, structured or n:m) and budget, and $\mathcal{L}(W^{(l)}, \widehat{W}^{(l)})$ the loss function, which measures the performance drop when $\widehat{W}^{(l)}$ in the $l$-th layer is replaced by $W^{(l)}$. Usually, the loss function uses a set of $N$ training samples $\{X_i\}_{i=1}^N$. As we describe below, there are two commonly used loss functions $\mathcal{L}$ in the one-shot pruning literature.

**Layer-wise reconstruction error.** Various existing layer-wise compression frameworks (Dong et al., 2017; He et al., 2017; Hubara et al., 2021; Frantar et al., 2022; Frantar & Alistarh, 2023; Sun et al., 2024) evaluate the pruned network's performance by examining changes in the output of the pruned layer. Their goal is to minimize the squared error loss between the layer's outputs generated by $W^{(l)}$ and $\widehat{W}^{(l)}$ on a training set. For a linear layer[1] $l$ with input dimension $d_{\text{in}}^{(l)}$ and output dimension $d_{out}^{(l)}$, we represent its input over $N$ training samples as a $d_{\text{in}}^{(l)} \times N$ matrix $X^{(l)}$. This reconstruction loss $(\mathcal{L}_R^{(l)})$ can be written as follows:

$$\mathcal{L}_R^{(l)}(W^{(l)}) := \left\| W^{(l)} X^{(l)} - \widehat{W}^{(l)} X^{(l)} \right\|_F^2. \tag{2}$$

By ensuring that the outputs of the pruned layers remain close to those of the corresponding dense layers, this objective preserves the functional behavior of each layer, thereby maintaining the overall integrity of the model. As the entire network is constructed through the composition of its individual layers, the layer-wise reconstruction error can be viewed as a local approximation of the training loss.

**Second-order Taylor approximation and Fisher loss.** Another line of work (Hassibi & Stork, 1992a; Singh & Alistarh, 2020b; Benbaki et al., 2023) considers the impact of pruning weights on the (global) training loss. They consider a second-order Taylor approximation of the training loss $\mathcal{L}_{\text{Tr}}$ around the pre-trained weights $\widehat{W}$ using a second-order Taylor expansion. Typically, we set $\nabla \mathcal{L}_{\text{Tr}}(\widehat{W}) = 0$ as $\widehat{W}$ is assumed to be a stationary point of the training loss.

Since computing the full Hessian is expensive, earlier work (Hassibi & Stork, 1992a) uses an approximation based on the empirical Fisher information matrix: $\nabla^2 \mathcal{L}_{\text{Tr}}(\widehat{W}) \approx H = \frac{1}{N} \sum_{i=1}^N \nabla \ell_i(\widehat{W}) \nabla \ell_i(\widehat{W})^\top$, where $\nabla \ell_i(\widehat{W})$ denotes the gradient of the network for weights $\widehat{W}$ on the $i$-th training sample. In the case of layer-wise pruning, we prune a layer $l$ while keeping the weights for all the other layers fixed, and the second-order Taylor approximation leads to the following Fisher loss:

$$\mathcal{L}_F^{(l)}(W^{(l)}) := \left( \text{vec}(W^{(l)}) - \text{vec}(\widehat{W}^{(l)}) \right)^\top H^{(l)} \left( \text{vec}(W^{(l)}) - \text{vec}(\widehat{W}^{(l)}) \right), \tag{3}$$

where $\text{vec}(W^{(l)})$ and $\text{vec}(\widehat{W}^{(l)})$ indicate the vector form of the pruned and dense weights in layer $l$ respectively, and $H^{(l)}$ denotes the submatrix of the approximated Hessian corresponding to the weights in layer $l$.

The Fisher loss $\mathcal{L}_F^{(l)}(W^{(l)})$ provides a more global view of the network's behavior (since this is based on computing the gradient of the entire pre-trained model) as opposed to the layer-wise reconstruction error which focuses on the outputs of individual layers.

**Pruning objective proposed in *MOONSHOT*.** Leveraging the merits of pruning based on the reconstruction of the layer outputs and the second-order Taylor approximation of the training loss, our framework introduces the task of pruning layer $l$ as a multi-objective optimization problem, defined as follows:

$$\min_{W^{(l)}} \ \left( \mathcal{L}_R^{(l)}(W^{(l)}), \mathcal{L}_F^{(l)}(W^{(l)}) \right) \quad \text{s.t.} \ \ \mathcal{S}(W^{(l)}) \leq S^{(l)} \tag{4}$$

This approach offers two benefits: (i) Targeting multiple objectives enhances the accuracy of the pruned networks beyond what is achievable with a single-objective (e.g. layer-wise reconstruction error or Fisher loss). (ii) By considering multiple objectives simultaneously and therefore leveraging more information, pruning becomes more robust, maintaining high performance even when one of the single objectives, $\mathcal{L}_R^{(l)}(W^{(l)})$ or $\mathcal{L}_F^{(l)}(W^{(l)})$, fails to accurately capture the network's overall performance.

---

[1] Convolutional layers can be processed in similar ways.

## 2.2 Reformulation as cardinality constrained convex quadratic problem

We consider a weighted combination of the two individual objectives to address the multi-objective pruning formulation in equation 4. To ensure a balanced consideration of $\mathcal{L}_R^{(l)}(W^{(l)})$ and $\mathcal{L}_F^{(l)}(W^{(l)})$, which might differ widely in magnitude, we normalize these objectives relative to their values at $\mathbf{0}$, the weight matrix filled with zeros. For $\lambda \in [0,1]$, we set the objective as:

$$\mathcal{L}_\lambda^{(l)} := (\lambda/\mathcal{L}_R^{(l)}(\mathbf{0}))\mathcal{L}_R^{(l)} + ((1-\lambda)/\mathcal{L}_F^{(l)}(\mathbf{0}))\mathcal{L}_F^{(l)} \tag{5}$$

In the following, we show how existing baselines can be adapted to the multi-objective formulation: we first explain how the block-diagonal structure of the Hessian arises in these methods, then how it can be preserved in the multi-objective case, and finally show how to reduce the multi-objective formulation to a quadratic problem under sparsity constraints, which can be addressed by existing pruning algorithms.

**Block-diagonal representation.** The layer-wise reconstruction loss can be rewritten (Frantar et al., 2022):

$$\mathcal{L}_R^{(l)}(W^{(l)}) = \sum_{i=1}^{d_{out}} \left\| W_{i,:}^{(l)\top} X^{(l)} - \widehat{W}_{i,:}^{(l)\top} X^{(l)} \right\|_2^2 \tag{6}$$

with $W_{i,:}^{(l)}$ and $\widehat{W}_{i,:}^{(l)}$ the $i$-th row of $W^{(l)}$ and $\widehat{W}^{(l)}$ respectively.

This allows us to express the layer-wise reconstruction loss in the following quadratic form:

$$\mathcal{L}_R^{(l)}(W^{(l)}) = \left( \text{vec}(W^{(l)}) - \text{vec}(\widehat{W}^{(l)}) \right)^\top H_R^{(l)} \left( \text{vec}(W^{(l)}) - \text{vec}(\widehat{W}^{(l)}) \right) \tag{7}$$

with $H_R^{(l)} = \text{Diag}\left( X^{(l)}(X^{(l)})^T, \ldots, X^{(l)}(X^{(l)})^T \right)$, a block-diagonal matrix containing $X^{(l)}(X^{(l)})^T$ $d_{out}$ times.

This exact block-diagonal structure enables Hessian computations to scale to large architectures, where a dense Hessian would be intractable. It also makes the objective separable, which can be exploited to improve the efficiency of pruning algorithms (Frantar et al., 2022; Frantar & Alistarh, 2023).

Similarly, for computational reasons, the existing single-objective pruning algorithms using the Fisher loss also assume $H^{(l)}$ in equation 3 to be block-diagonal (Singh & Alistarh, 2020b; Benbaki et al., 2023; Kuznedelev et al., 2023). Specifically, we can write $H^{(l)}$ in Fisher loss as $\text{Diag}(H_1^{(l)}, H_2^{(l)}, \cdots, H_K^{(l)})$ with $K$ the number of blocks. Using the quadratic expressions from equation 3 and equation 7, the weighted loss from equation 5 becomes:

$$\mathcal{L}_\lambda^{(l)}(W^{(l)}) = \left( \text{vec}(W^{(l)}) - \text{vec}(\widehat{W}^{(l)}) \right)^\top \left( \frac{\lambda}{\mathcal{L}_R^{(l)}(\mathbf{0})} H_R^{(l)} + \frac{1-\lambda}{\mathcal{L}_F^{(l)}(\mathbf{0})} H^{(l)} \right) \left( \text{vec}(W^{(l)}) - \text{vec}(\widehat{W}^{(l)}) \right) \tag{8}$$

$$= \frac{\lambda}{\mathcal{L}_R^{(l)}(\mathbf{0})} \sum_{i=1}^{d_{out}^{(l)}} \left( W_{i,:}^{(l)} - \widehat{W}_{i,:}^{(l)} \right)^T X^{(l)}(X^{(l)})^T \left( W_{i,:}^{(l)} - \widehat{W}_{i,:}^{(l)} \right) + \frac{1-\lambda}{\mathcal{L}_F^{(l)}(\mathbf{0})} \sum_{k=1}^{K} (w_k^{(l)} - \widehat{w}_k^{(l)})^\top H_k^{(l)} (w_k^{(l)} - \widehat{w}_k^{(l)}) \tag{9}$$

where $w_k^{(l)}$ and $\widehat{w}_k^{(l)}$ denote the weights of $W^{(l)}$ and $\widehat{W}^{(l)}$ corresponding to block $k$ in $H^{(l)}$ respectively.

**Adapting existing baselines.** If $H_R^{(l)}$ and $H^{(l)}$ use different block sizes, the original block structure would be altered. To isolate the impact of the multi-objective formulation while preserving the effectiveness of the baseline, *MOONSHOT* enforces the block size of the original baseline. As described in Figure 2, there are two main cases:

- For algorithms focusing only on the Fisher loss, such as CAP (Kuznedelev et al., 2023), we can apply a block-diagonal approximation to $H_R$, specifically to $X^{(l)}(X^{(l)})^T$, ensuring that each block of $H_R^{(l)}$ aligns in size with the corresponding block of $H^{(l)}$. In this case, we maintain the original block-diagonal approximation of $H^{(l)}$ assumed by the single-objective baseline.

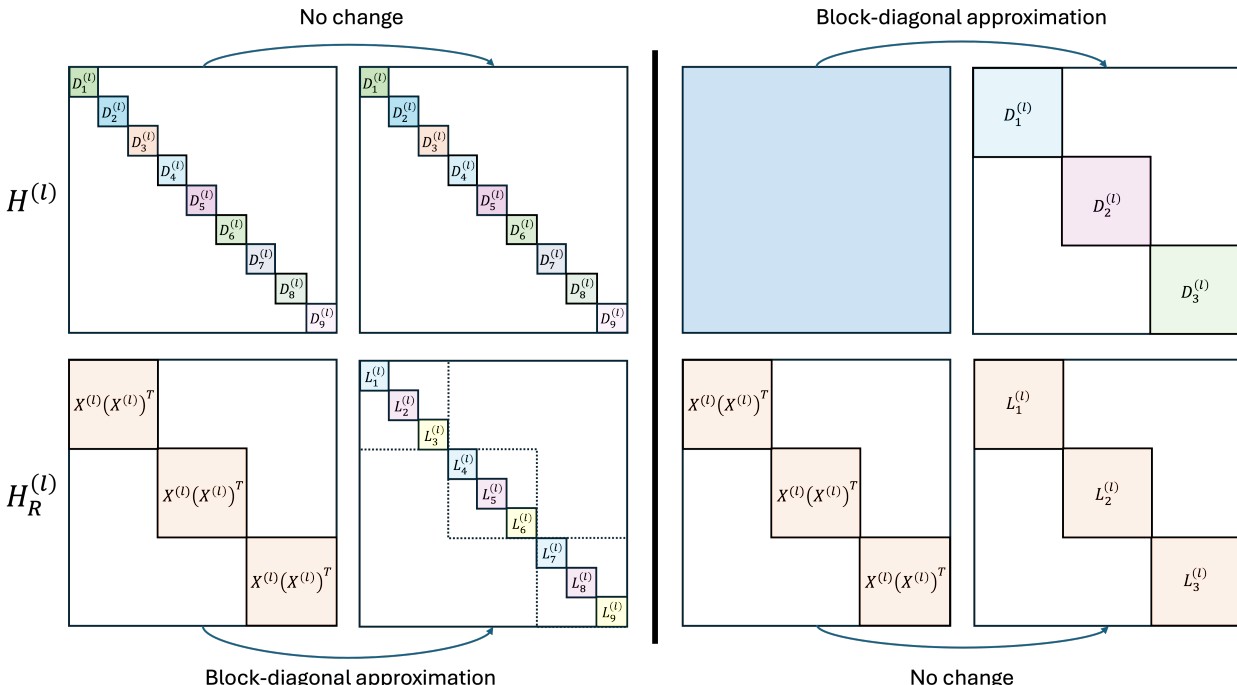

Figure 2: Depending on the block-diagonal approximation assumed by the single-objective algorithm, *MOONSHOT* matches the size of the original block-diagonal approximation to the Hessian of the other objective. [Left] Case 1: When adapting a Fisher-objective algorithm, we keep the block-diagonal approximation of $H^{(l)}$ from the baseline unchanged, and perform a block-diagonal approximation on $H_R^{(l)}$ (more precisely on $X^{(l)}(X^{(l)})^T$). [Right] Case 2: When adapting a layer-wise reconstruction objective algorithm, we keep the exact block-diagonal form of $H_R^{(l)}$, as in the original baseline, and perform a block-diagonal approximation on $H^{(l)}$.

- For algorithms that focus solely on the layer-wise reconstruction loss, such as OBC (Frantar et al., 2022), SparseGPT (Frantar & Alistarh, 2023) and OSSCAR (Meng et al., 2024b), we can set $K$ to $d_{\text{out}}^{(l)}$, ensuring that each block of $H^{(l)}$ matches the dimensions of $X^{(l)}(X^{(l)})^T$. In this case, we maintain the exact block-diagonal structure of $H_R^{(l)}$ without further approximation, as in the original baseline.

Finally, for Wanda (Sun et al., 2024) which uses a diagonal approximation of $X^{(l)}(X^{(l)})^T$, we use a diagonal approximation of $H^{(l)}$ as well.

In all cases, we write in the following $H_R^{(l)} = \text{Diag}\left(L_1^{(l)}, \dots, L_K^{(l)}\right)$ ($L_k^{(l)}$ denotes either a block of the block-diagonal approximation of $X^{(l)}(X^{(l)})^T$ or $X^{(l)}(X^{(l)})^T$ itself, depending on the baseline).

**Quadratic formulation.** Let $F_k^{(l)} = \frac{\lambda}{\mathcal{L}_R^{(l)}(\mathbf{0})} L_k^{(l)} + \frac{1-\lambda}{\mathcal{L}_F^{(l)}(\mathbf{0})} H_k^{(l)}$. The multi-objective formulation in equation 4 can be reformulated as the following quadratic optimization problem under sparsity constraints:

$$\min_{W^{(l)}} \quad \mathcal{L}_\lambda^{(l)}(W^{(l)}) = \sum_{k=1}^{K} (w_k^{(l)} - \widehat{w}_k^{(l)})^\top F_k^{(l)} (w_k^{(l)} - \widehat{w}_k^{(l)}) \qquad \text{s.t.} \qquad \mathcal{S}(W^{(l)}) \leq S^{(l)}, \tag{10}$$

Most single-objective pruning methods reduce to solving a separable quadratic problem with sparsity constraints (Frantar & Alistarh, 2023; Kuznedelev et al., 2023; Frantar et al., 2022; Meng et al., 2024b). At this point, the formulation resembles the single-objective case (equation 3 and equation 7), which means

---

**Algorithm 1** Efficient Computation of the Block-Diagonal Hessian Inverse

---

**Input:** Layer input matrix $X^{(l)} \in \mathbb{R}^{d_{\text{in}}^{(l)} \times N}$, per-sample gradients $A_k^{(l)} \in \mathbb{R}^{d_{\text{in}}^{(l)} \times N}$ for each block $k = 1, \ldots, K$, multi-objective weight $\lambda \in [0,1]$.

1: Compute base inverse:

$$J_0 \leftarrow \left( \frac{\lambda}{\mathcal{L}_R^{(l)}(\mathbf{0})} X^{(l)}(X^{(l)})^T \right)^{-1}$$

2: **for** each block $k = 1, \ldots, K$ **do**

3:     Compute $G_k^{(l)}$ using the Woodbury identity (see Appendix A.3):

$$G_k^{(l)} \leftarrow J_0 - \left( \frac{1-\lambda}{N\mathcal{L}_F^{(l)}(\mathbf{0})} \right) J_0 A_k^{(l)} \left( I_N + \frac{1-\lambda}{N\mathcal{L}_F^{(l)}(\mathbf{0})} A_k^{(l)\top} J_0 A_k^{(l)} \right)^{-1} A_k^{(l)\top} J_0$$

**Output:** Block-diagonal Hessians inverse $\{(F_k^{(l)})^{-1}\}_{k=1}^K = \{G_k^{(l)}\}_{k=1}^K$

---

that existing single-objective baselines can, in principle, be applied to the new multi-objective setting. However, as we show in the following section, a direct application in the case of LLMs is intractable and requires additional adaptations.

### 2.3 Efficient inverse Hessian computation for LLMs

Most state-of-the-art single-objective pruning methods (Frantar et al., 2022; Kuznedelev et al., 2023; Frantar & Alistarh, 2023; Sun et al., 2024; Meng et al., 2024b) require access to the inverse Hessian $(F_k^{(l)})^{-1}$ for each layer $l$ and block $k$ in order to compute the impact of pruning a weight $w_p$ on the objective, as well as the potential update on the support $\delta_p$, as described in the OBS algorithm (Hassibi & Stork, 1992b):

$$p = \text{argmin}_p \frac{[w_k^{(l)}]_p^2}{[(F_k^{(l)})^{-1}]_{p,p}}, \quad \delta_p = -\frac{[w_k^{(l)}]_p}{[(F_k^{(l)})^{-1}]_{p,p}} [(F_k^{(l)})^{-1}]_{:,p} \tag{11}$$

In the case of structured pruning, this formula can be extended to determine the impact of pruning an entire column (Kurtic et al., 2023; Meng et al., 2024b) but still requires computing $(F_k^{(l)})^{-1}$. While this inverse Hessian can be computed efficiently for vision models (as in OBC (Frantar et al., 2022) and CAP (Kuznedelev et al., 2023)), the computation becomes significantly more challenging in the context of LLMs. This is due to the larger number of blocks and the higher dimensions of $F_k^{(l)}$. Indeed, state-of-the-art layer-wise pruning algorithms like SparseGPT (Frantar & Alistarh, 2023) and OSSCAR (Meng et al., 2024b) use the layer-wise reconstruction loss with no further block-diagonal approximation: $L_1^{(l)} = \cdots = L_K^{(l)} = X^{(l)}(X^{(l)})^T$. This design requires just one matrix inversion of $X^{(l)}(X^{(l)})^T$, which makes the algorithm practical even for large models. In our multi-objective formulation, by contrast, each block $F_k^{(l)}$ is different due to the Fisher loss component. Therefore, a naive adaptation of such algorithms to the multi-objective formulation would require computing K matrix inversions (one for each block) instead of one, which would significantly slow down the algorithm. However, by leveraging the structure of our multi-objective formulation, we can compute the inverse Hessian efficiently. In particular, for a layer $l$, the Hessian component from the layer-wise reconstruction error $X^{(l)}(X^{(l)})^T$ does not depend on the block $k$. In addition, the size of calibration sets $N$ for LLMs is often small (128 for SparseGPT, Wanda and OSSCAR). Consequently, the Hessian component coming from the Fisher loss $H_k^{(l)}$ is a matrix of low rank $r \leq N \ll d_{\text{in}}^{(l)}$ that can be written as $H_k^{(l)} = \frac{1}{N} A_k^{(l)} A_k^{(l)T}$ with $A_k^{(l)} = \begin{bmatrix} \nabla \ell_{1,k}^{(l)} & \nabla \ell_{2,k}^{(l)} & \cdots & \nabla \ell_{N,k}^{(l)} \end{bmatrix} \in \mathbb{R}^{d_{\text{in}}^{(l)} \times N}$ ($d_{\text{in}}^{(l)}$ is the block-size in this case). We therefore propose to compute $G_k^{(l)} = (F_k^{(l)})^{-1}$, necessary for the state-of-the-art OBS strategy used in SparseGPT, OBC, Wanda, CAP, OSSCAR, following the procedure described in Algorithm 1. Our exact adaptation of the SparseGPT algorithm, denoted *MOONSHOT*-SparseGPT, is provided in Appendix A.2.

Here, $I_N \in \mathbb{R}^{N \times N}$ is the identity matrix, and $I_N + \left( \frac{1-\lambda}{N \mathcal{L}_F^{(l)}(\mathbf{0})} \right) A_k^{(l)} J_0 A_k^{(l)T}$ is of size $N \times N$. Therefore, the $N \times N$ matrix inversion and the Woodbury identity (Woodbury & of Statistics, 1950) can be computed very efficiently (at most 5-6 seconds for the largest layers of Llama-3.2-3B) in the case we are interested in ($N = 128$). In particular, we observe a speedup of up to 6 times in comparison to inverting all the blocks using the standard matrix inversion via Cholesky decomposition (used in SparseGPT and OBC for example). While previous works also use the Woodbury identity to compute the Hessian inverse in the literature (Singh & Alistarh, 2020a; Kurtic et al., 2022), we extend in this paper its application to the billion-parameter scale and adapt it to the multi-objective pruning setting. The exact derivation of the update in Algorithm 1 is provided in Appendix A.3. It is important to note that the steps described above enable exact computation of the Hessian inverse for the block-diagonal approximation shown in Figure 2.

## 3 Experiments

### 3.1 Models and Datasets

We evaluate the performance of our method on a wide range of models and baselines. Thus we prune:

- Several Llama models: Llama-3.2-1B (1B parameters) (Grattafiori et al., 2024), Llama-3.2-3B (3B parameters) (Grattafiori et al., 2024) and Llama-2-13b-chat-hf (Touvron et al., 2023) (13B parameters) using SparseGPT (Frantar & Alistarh, 2023), Wanda (Sun et al., 2024) and OSSCAR (Meng et al., 2024b)
- The DeiT Vision Transformers (Touvron et al., 2021): DeiT-Tiny (5.7M parameters), DeiT-Small (22.1M parameters) and DeiT-Base (86.6M parameters) using CAP (Kuznedelev et al., 2023)
- A Convolutional Neural Network (He et al., 2015): ResNet-50 (25.6M parameters) using OBC (Frantar et al., 2022)

We additionally prune the Instruct variants of Llama-3.2-1B and Llama-3.2-3B using SparseGPT and Wanda and report the results in Appendix A.9.

OSSCAR (Meng et al., 2024b) greedily prunes columns by optimizing a layer-wise reconstruction objective, with an optional local-search refinement. In our experiments, we use OSSCAR's default hyperparameters and focus on OSSCAR's greedy pruning step.

For pruning, we use 128 samples for the LLMs from the C4 (Raffel et al., 2020) dataset, and 4096 samples from ImageNet-1k (Deng et al., 2009) for the vision models. While we focus on the test accuracy on ImageNet-1k for the vision models, we focus on both perplexity and zero-shot performance for Llama-3.2. Following previous work (Frantar & Alistarh, 2023; Sun et al., 2024), we compute the test perplexity on WikiText2 (Merity et al., 2016) and PTB (Marcus et al., 1994). Additionally, we assess the zero-shot accuracy of the pruned LLMs on a variety of common-sense reasoning datasets, including BoolQ (Clark et al., 2019), PIQA (Bisk et al., 2020), HellaSwag (Zellers et al., 2019), WinoGrande (Sakaguchi et al., 2021), ARC-easy (Clark et al., 2018), ARC-challenge (Clark et al., 2018), and OpenBookQA (Mihaylov et al., 2018). In addition to mean performance across the seven classification benchmarks, we also report the win rate, i.e., the percentage of benchmarks on which one method outperforms the other.

### 3.2 Setup

We prune Llama-3.2-3B and Llama-2-13b-chat-hf on a single NVIDIA A100 GPU (80 GB) and Llama-3.2-1B on a single NVIDIA L40 GPU (40 GB). For the vision models, we use four NVIDIA L40 GPUs (40 GB each) and prune the layers across the four devices. *MOONSHOT* is implemented in PyTorch (Paszke et al., 2019).

### 3.3 Implementation Details

**Pruning blocks of rows with *MOONSHOT*-SparseGPT and *MOONSHOT*-OSSCAR.** Unlike the single-objective setting where the Hessian is block-diagonal with identical blocks repeated, $H = \mathrm{Diag}(XX^\top, \dots, XX^\top)$, *MOONSHOT*-SparseGPT and *MOONSHOT*-OSSCAR with $\lambda \neq 1$ use a Hessian with row-dependent blocks, $H = \mathrm{Diag}(F_1, \dots, F_K)$. In the case of the largest layers, storing all the blocks

simultaneously can become infeasible under GPU memory constraints. Our adaptation of SparseGPT and OSSCAR prunes the rows by blocks of size $K_p$. The exact adaptation of SparseGPT is described in Appendix A.2.

**Efficient Backsolve for OSSCAR.** After the greedy column-selection step, OSSCAR performs a backsolve (i.e., it computes the optimal weights on the support). To preserve efficiency in the multi-objective setting, we exploit problem structure to perform this backsolve efficiently. Additional details are provided in Appendix A.4.

**Pruning Efficiency.** In the case of the attention layers, a sufficiently large $K_p$ can be used; however, for the much larger projection layers, $K_p$ often needs to be reduced, which can increase runtime due to reduced parallelism. Moreover, the projection layers typically require a larger $H$, further increasing computational cost. To maintain the efficiency of the original method in the case of LLMs, only the self-attention layers are pruned using the multi-objective formulation. Concretely, this corresponds to `q_proj`, `k_proj`, `v_proj`, and `o_proj` for SparseGPT, and `o_proj` for OSSCAR (OSSCAR prunes only the `down_proj` and `o_proj` matrices.) The projection layers are pruned using the layer-wise reconstruction loss only. For SparseGPT, we select $K_p$ such that at least 50% of rows are pruned at a time for Llama-3.2-1B and Llama-3.2-3B, and fix $K_p = 512$ for Llama-2-13b-chat-hf. For OSSCAR, $K_p$ corresponds to 50% of the rows for Llama-3.2-1B and 25% of the rows for Llama-3.2-3B. An evaluation of *MOONSHOT*'s effectiveness with the pruning times of each method is included in Appendix A.5.

We additionally evaluate the impact of applying *MOONSHOT* across both the attention and projection layers of Llama, both in terms of performance and computational cost, in Appendix A.8.

**Hessian Recomputation.** Fisher-based methods typically compute $H^{(l)}$ once, as recomputation requires per-sample gradients, and they report results with Hessian recomputation as a more costly alternative (Benbaki et al., 2023; Kuznedelev et al., 2023). In contrast, layer-wise reconstruction-based methods like SparseGPT and Wanda recompute $H_R^{(l)}$ after each block of layers, as $H_R^{(l)}$ depends only on the input data and can be recomputed with relatively low overhead (Frantar & Alistarh, 2023; Sun et al., 2024). In this paper, due to the multi-objective formulation, we follow the standard approach in the Fisher-based literature, and compute the Hessian (inverse) once. We also report results with Hessian recomputation after each block for SparseGPT and Wanda in Appendix A.7.

**Selecting $\lambda$ in equation 10.** The results provided in Tables 2, 3 and in Figure 4 are obtained by selecting the best value of $\lambda$ based on the training loss for vision models and training perplexity for LLMs. For the vision models, we evaluate $\lambda \in \{0.0, 0.25, 0.5, 0.75, 1.0\}$, and for the Llama-3.2 models, we test $\lambda \in \{0.0, 0.1, 0.25, 0.5, 0.75, 0.9, 1.0\}$. For Llama-2-13b-chat-hf, we only test $\lambda \in \{0.9, 1.0\}$.

While $\lambda$ is selected via tuning, Section 4.1 shows that $\lambda \in (0, 1)$ - that is, beyond the standard single-objective baselines ($\lambda = 0$ or 1) - almost always lead to better performance. In addition, $\lambda = 0.5$ for vision models and $\lambda = 0.9$ for LLMs serve as simple and effective defaults in resource- or time-constrained scenarios. For typical pruning use cases, where pruning is performed once offline and the resulting model is used across multiple downstream tasks or applications, investing in hyperparameter tuning can further enhance the performance gains achieved by *MOONSHOT*.

**Hyperparameters.** Additional information on the hyperparameters used for the baselines and *MOON-SHOT* is provided in Appendix A.6.

### 3.4 Main Results

Tables 2 and 3 report results at relevant sparsity levels: 10% structured and 60%/70% unstructured for LLMs, 70% unstructured for DeiT models, and 90% unstructured for ResNet-50, in addition to 2:4 semi-structured sparsity. Across all settings, *MOONSHOT* consistently outperforms the baseline, yielding statistically significant improvements. Comprehensive results across architectures, sparsity regimes and $\lambda$ values are available in Appendix A.12.

**Vision Models.** Table 2 shows that on DeiT-Small, *MOONSHOT* improves test accuracy on ImageNet-1k by up to 5.5 points compared to CAP at 70% unstructured sparsity. This indicates that the Fisher-based

Hessian used in CAP is insufficiently informative at this level of compression. In contrast, our multi-objective formulation yields a more stable and informative Hessian, resulting in a much higher quality pruned model. DeiT-Tiny and DeiT-Base also show consistent improvements of 1–3 points across both unstructured and semi-structured sparsity settings. For ResNet-50, *MOONSHOT* improves accuracy by 4 points at 90% unstructured sparsity compared to OBC, and further improves performance at 2:4 sparsity.

**Language Models.** Table 3 shows that on Llama-3.2-1B, *MOONSHOT* lowers test perplexity on C4 by up to 54 points with Wanda at 2:4 sparsity and by 13 points with SparseGPT at 60% unstructured sparsity. These improvements extend across other language modeling benchmarks (WikiText2, PTB) and generalize to downstream classification tasks, where mean accuracy often improves by up to 1 point. Similar results are observed for Llama-3.2-3B and Llama-2-13b-chat-hf, and *MOONSHOT* improves the mean accuracy of these models by up to 1.5 points at 60% and 70% unstructured sparsity respectively.

Importantly, *MOONSHOT* complements existing sparsity allocation strategies. When combined with AlphaPruning or OWL, it yields additional performance gains. For example, on Llama-3.2-1B at 60% unstructured sparsity, combining *MOONSHOT* with OWL leads to a further 13-point reduction in C4 perplexity and a 0.4-point increase in mean downstream accuracy.

In the case of structured pruning, the gains are particularly high, with up to 30% lower perplexity on WikiText2, 22% on PTB, and 11% on C4, together with a +4.9 point improvement in mean accuracy.

Finally, in terms of win rate, *MOONSHOT* outperforms the baseline on most benchmarks across sparsity regimes, architectures, and pruning baselines.

# 4 Ablation studies

## 4.1 Selecting $\lambda$

To demonstrate the efficacy of our proposed multi-objective formulation, we try different values of $\lambda$ in equation 10. $\lambda$ determines the balance between the layer-wise reconstruction error and the Fisher loss.

Figure 3 below, and Figure 5 in Appendix A.10, illustrate that setting neither $\lambda = 0.0$ nor $\lambda = 1.0$ achieves the best results on ResNet-50, DeiT-Base, and the Llama-3.2 models. The results are striking for the Llama models, for which the test perplexity on C4 is substantially lower in the multi-objective regime than the single objective regime. An intermediate value of $\lambda$ that leverages the advantages of both loss functions is more effective. Furthermore, $\lambda$ seems to require minimal tuning, as a value of $\lambda$ other than 0 and 1 is often sufficient to achieve near-optimal performance. $\lambda = 0.5$ for vision models and $\lambda = 0.9$ for LLMs are relatively good choices across all architectures and sparsity levels tested.

## 4.2 Performance of *MOONSHOT* Across Sparsity Regimes

Figure 4 shows that *MOONSHOT* consistently outperforms the baselines across all tested sparsity levels. Performance gains are especially pronounced at higher sparsity levels, where preserving original performance is increasingly difficult. Additional results can be found in Appendix A.11.

Table 2: Impact of *MOONSHOT* on CAP for the DeiT models (left) and OBC for ResNet-50 (right) across unstructured and 2:4 sparsity levels. ImageNet-1k accuracies over 3 seeds are averaged with standard errors.

| Sparsity | Method | DeiT Tiny | DeiT Small | DeiT Base |
|---|---|---|---|---|
| Dense | - | 72.14 | 79.83 | 81.80 |
| 0.7 | CAP | $44.22 \pm 0.32$ | $57.50 \pm 0.83$ | $70.44 \pm 0.15$ |
| 0.7 | *MOONSHOT*-CAP | $\mathbf{45.05} \pm 0.20$ | $\mathbf{62.97} \pm 0.15$ | $\mathbf{73.42} \pm 0.06$ |
| 2:4 | CAP | $52.28 \pm 0.04$ | $69.65 \pm 0.02$ | $76.21 \pm 0.07$ |
| 2:4 | *MOONSHOT*-CAP | $\mathbf{54.20} \pm 0.15$ | $\mathbf{71.54} \pm 0.08$ | $\mathbf{77.88} \pm 0.05$ |

| Sparsity | Method | ResNet-50 |
|---|---|---|
| Dense | - | 77.11 |
| 0.9 | OBC | $51.52 \pm 0.07$ |
| 0.9 | *MOONSHOT*-OBC | $\mathbf{55.52} \pm 0.09$ |
| 2:4 | OBC | $75.46 \pm 0.03$ |
| 2:4 | *MOONSHOT*-OBC | $\mathbf{75.50} \pm 0.03$ |

Table 3: Impact for the Llama-3.2 models of *MOONSHOT* on SparseGPT/Wanda at 60% unstructured sparsity (including with OWL and AlphaPruning), 2:4 sparsity, and OSSCAR at 10% structured sparsity. We also include Llama-2-13b-chat-hf at 70% unstructured sparsity. The perplexities on C4, WikiText2 and PTB, as well as the zero-shot accuracies, are averaged over 3 seeds with standard errors. Mean performance and win rate are computed over the 7 zero-shot downstream classification tasks.

(a) Llama-3.2-1B

| Sparsity | Method | C4↓ | WikiText2↓ | PTB↓ | BoolQ↑ | HellaSwag↑ | WinoGrande↑ | ARC-e↑ | ARC-c↑ | PIQA↑ | OBQA↑ | Mean↑ | Win Rate↑ |
|---|---|---|---|---|---|---|---|---|---|---|---|---|---|
| Dense | - | 14.02 | 9.75 | 17.59 | 64.01 | 47.73 | 60.14 | 65.15 | 31.23 | 74.32 | 26.4 | 52.71 | - |
| 0.1 (structured) | OSSCAR | 43.00±1.28 | 43.91±0.60 | 106.62±8.57 | 52.61±1.63 | 35.82±0.29 | 54.17±1.11 | 45.19±7.83 | 24.23±2.04 | 65.65±3.68 | 15.93±0.33 | 41.94±1.98 | 19.05±12.60 |
| | *MOONSHOT*-OSSCAR | 38.04±0.58 | 30.93±0.92 | 86.73±9.88 | 56.55±0.90 | 37.89±1.11 | 55.51±0.37 | 58.84±0.38 | 28.58±0.51 | 71.49±0.06 | 18.93±0.48 | 46.83±0.42 | 80.95±12.60 |
| 0.6 | SparseGPT | 63.63±1.18 | 54.60±1.00 | 81.11±3.99 | 60.67±0.59 | 32.16±0.20 | 54.46±0.53 | 44.94±0.11 | 21.47±0.48 | 62.21±0.20 | 17.07±0.41 | 41.85±0.20 | 42.86±8.25 |
| | *MOONSHOT*-SparseGPT | 50.28±1.99 | 39.13±1.54 | 60.14±2.90 | 62.36±0.12 | 32.49±0.13 | 53.09±0.18 | 46.49±0.38 | 21.30±0.24 | 63.22±0.17 | 15.73±0.55 | 42.10±0.11 | 57.14±8.25 |
| 0.6 (Alpha-Pruning) | SparseGPT | 61.05±0.77 | 52.80±0.43 | 78.27±3.64 | 62.08±0.26 | 32.00±0.08 | 53.88±0.25 | 45.29±0.49 | 22.01±0.59 | 62.02±0.42 | 17.60±0.42 | 42.13±0.06 | 33.33±9.52 |
| | *MOONSHOT*-SparseGPT | 49.31±1.10 | 38.44±0.52 | 60.32±1.20 | 62.29±0.05 | 32.53±0.06 | 53.70±0.55 | 46.30±0.30 | 21.99±0.15 | 63.13±0.10 | 16.60±0.12 | 42.36±0.14 | 66.67±9.52 |
| 0.6 (OWL) | SparseGPT | 56.82±1.68 | 49.54±1.22 | 68.73±3.48 | 62.20±0.11 | 32.86±0.05 | 53.67±0.33 | 44.14±0.71 | 23.46±0.05 | 62.59±0.21 | 18.00±0.76 | 42.42±0.07 | 33.33±17.17 |
| | *MOONSHOT*-SparseGPT | 43.58±0.91 | 35.72±0.17 | 53.02±0.91 | 62.20±0.04 | 33.66±0.16 | 53.54±0.55 | 46.37±0.23 | 23.41±0.08 | 64.04±0.03 | 16.40±0.20 | 42.80±0.07 | 66.67±17.17 |
| 2:4 | SparseGPT | 53.59±0.35 | 42.56±0.37 | 63.79±0.19 | 61.42±0.21 | 31.68±0.05 | 53.83±0.37 | 44.04±0.23 | 21.47±0.44 | 61.79±0.40 | 15.00±0.20 | 41.32±0.08 | 57.14±14.29 |
| | *MOONSHOT*-SparseGPT | 50.99±0.47 | 38.00±0.58 | 59.32±1.55 | 61.98±0.29 | 31.47±0.21 | 53.09±0.27 | 45.37±0.50 | 20.28±0.58 | 62.13±0.19 | 14.33±0.35 | 41.24±0.20 | 42.86±14.29 |
| 0.6 | Wanda | 117.71±0.87 | 84.73±0.73 | 119.64±1.00 | 58.96±1.39 | 28.86±0.03 | 51.35±0.49 | 38.82±0.32 | 18.94±0.26 | 59.05±0.18 | 13.93±0.24 | 38.56±0.12 | 19.05±4.76 |
| | *MOONSHOT*-Wanda | 86.55±1.67 | 63.57±1.61 | 98.44±3.98 | 61.56±0.29 | 29.53±0.06 | 51.64±0.23 | 40.40±0.17 | 19.60±0.06 | 61.12±0.08 | 13.40±0.20 | 39.61±0.07 | 80.95±4.76 |
| 0.6 (Alpha-Pruning) | Wanda | 112.33±1.03 | 80.75±1.03 | 120.89±0.73 | 58.01±1.10 | 28.92±0.09 | 51.22±0.47 | 37.56±0.16 | 19.51±0.21 | 59.32±0.04 | 13.40±0.40 | 38.28±0.10 | 14.29±8.25 |
| | *MOONSHOT*-Wanda | 84.21±1.04 | 63.30±0.92 | 101.96±2.98 | 61.69±0.24 | 29.56±0.08 | 52.33±0.37 | 39.28±0.29 | 19.65±0.06 | 60.32±0.13 | 12.07±0.58 | 39.27±0.09 | 85.71±8.25 |
| 0.6 (OWL) | Wanda | 99.38±1.37 | 73.00±0.37 | 111.24±1.83 | 61.28±0.28 | 29.88±0.11 | 52.07±0.82 | 40.22±0.09 | 20.82±0.05 | 60.03±0.20 | 14.80±0.12 | 39.87±0.17 | 33.33±4.76 |
| | *MOONSHOT*-Wanda | 73.97±0.19 | 58.81±0.28 | 100.07±1.96 | 61.81±0.08 | 30.47±0.07 | 51.51±0.37 | 40.92±0.22 | 20.71±0.15 | 60.36±0.19 | 13.27±0.35 | 39.86±0.15 | 66.67±4.76 |
| 2:4 | Wanda | 164.32±2.37 | 114.73±2.32 | 190.58±1.50 | 57.28±1.00 | 28.32±0.03 | 51.51±0.15 | 35.76±0.26 | 18.34±0.09 | 58.41±0.35 | 13.67±0.07 | 37.61±0.10 | 28.57±0.00 |
| | *MOONSHOT*-Wanda | 110.74±1.17 | 78.55±1.14 | 126.91±1.66 | 61.08±0.57 | 28.51±0.05 | 50.62±0.25 | 38.41±0.26 | 19.43±0.08 | 59.36±0.33 | 12.67±0.37 | 38.58±0.14 | 71.43±0.00 |

(b) Llama-3.2-3B

| Sparsity | Method | C4↓ | WikiText2↓ | PTB↓ | BoolQ↑ | HellaSwag↑ | WinoGrande↑ | ARC-e↑ | ARC-c↑ | PIQA↑ | OBQA↑ | Mean↑ | Win Rate↑ |
|---|---|---|---|---|---|---|---|---|---|---|---|---|---|
| Dense | - | 11.34 | 7.81 | 13.54 | 72.72 | 55.30 | 69.22 | 74.37 | 42.41 | 76.71 | 31.20 | 60.28 | - |
| 0.1 (structured) | OSSCAR | 16.80±0.46 | 13.76±0.85 | 22.34±0.43 | 64.38±0.86 | 49.03±0.86 | 61.30±0.74 | 65.82±0.64 | 34.10±0.53 | 75.70±0.78 | 27.33±0.41 | 53.95±0.08 | 33.33±4.76 |
| | *MOONSHOT*-OSSCAR | 16.56±0.38 | 13.28±0.55 | 22.10±0.28 | 64.91±1.00 | 49.50±0.62 | 61.43±0.47 | 66.39±0.83 | 34.41±0.33 | 75.54±0.45 | 26.80±0.76 | 54.14±0.20 | 66.67±4.76 |
| 0.6 | SparseGPT | 33.63±0.14 | 26.12±0.23 | 42.69±0.73 | 66.82±0.60 | 38.14±0.14 | 60.91±0.65 | 53.89±0.11 | 26.28±0.18 | 67.75±0.34 | 18.47±0.44 | 47.47±0.08 | 4.76±4.76 |
| | *MOONSHOT*-SparseGPT | 28.23±0.11 | 22.46±0.17 | 35.63±0.68 | 67.76±0.35 | 39.13±0.07 | 61.01±0.23 | 57.59±0.92 | 27.79±0.71 | 69.44±0.13 | 20.00±0.53 | 48.96±0.12 | 95.24±4.76 |
| 0.6 (Alpha-Pruning) | SparseGPT | 34.19±0.40 | 26.06±0.64 | 42.82±0.13 | 68.17±0.43 | 38.62±0.16 | 61.98±0.81 | 53.37±0.70 | 26.00±0.23 | 68.06±0.38 | 19.80±0.90 | 48.00±0.38 | 14.29±8.25 |
| | *MOONSHOT*-SparseGPT | 28.64±0.25 | 22.17±0.19 | 35.48±1.52 | 68.73±0.11 | 39.34±0.18 | 62.27±0.52 | 56.52±0.25 | 26.93±0.37 | 69.13±0.32 | 20.33±0.24 | 49.04±0.06 | 85.71±8.25 |
| 0.6 (OWL) | SparseGPT | 29.15±0.31 | 23.58±0.40 | 36.58±0.78 | 66.64±0.77 | 39.89±0.16 | 61.96±0.40 | 55.57±0.41 | 27.53±0.12 | 68.72±0.14 | 21.20±0.35 | 48.79±0.11 | 28.57±8.25 |
| | *MOONSHOT*-SparseGPT | 25.31±0.12 | 20.85±0.11 | 31.39±0.69 | 67.41±0.47 | 40.64±0.13 | 61.62±0.09 | 57.39±0.78 | 28.16±0.62 | 69.73±0.29 | 20.60±0.50 | 49.36±0.17 | 71.43±8.25 |
| 2:4 | SparseGPT | 30.00±0.29 | 24.40±0.23 | 38.32±0.74 | 65.64±0.70 | 38.31±0.08 | 59.93±0.13 | 55.63±1.10 | 26.14±0.42 | 68.34±0.33 | 20.87±0.35 | 47.83±0.18 | 52.38±9.52 |
| | *MOONSHOT*-SparseGPT | 28.79±0.29 | 23.21±0.32 | 35.94±0.73 | 65.58±0.08 | 38.06±0.13 | 59.30±0.41 | 55.99±0.63 | 26.56±0.37 | 67.94±0.21 | 20.93±0.55 | 47.77±0.18 | 47.62±9.52 |
| 0.6 | Wanda | 41.98±0.40 | 30.56±0.32 | 51.00±0.45 | 64.82±0.35 | 35.12±0.07 | 56.56±0.46 | 50.58±0.41 | 23.83±0.12 | 65.58±0.19 | 16.93±0.07 | 44.77±0.10 | 38.10±4.76 |
| | *MOONSHOT*-Wanda | 37.73±0.19 | 27.71±0.26 | 46.47±0.08 | 61.33±0.91 | 35.53±0.08 | 54.83±0.14 | 52.53±0.29 | 24.69±0.21 | 66.81±0.03 | 16.60±0.23 | 44.62±0.12 | 61.90±4.76 |
| 0.6 (Alpha-Pruning) | Wanda | 40.03±0.04 | 29.19±0.20 | 50.24±0.27 | 65.93±0.24 | 35.95±0.01 | 57.51±0.09 | 51.09±0.16 | 24.32±0.36 | 66.03±0.15 | 16.87±0.24 | 45.39±0.03 | 38.10±4.76 |
| | *MOONSHOT*-Wanda | 37.37±0.21 | 27.08±0.16 | 47.01±0.29 | 63.38±0.65 | 36.05±0.10 | 57.51±0.56 | 54.15±0.27 | 25.43±0.18 | 66.96±0.36 | 16.27±0.18 | 45.68±0.10 | 61.90±4.76 |
| 0.6 (OWL) | Wanda | 37.35±0.14 | 27.93±0.23 | 44.25±0.74 | 67.26±0.07 | 37.18±0.13 | 59.06±0.73 | 51.89±0.34 | 25.54±0.23 | 66.47±0.10 | 17.20±0.12 | 46.37±0.18 | 33.33±9.52 |
| | *MOONSHOT*-Wanda | 34.56±0.11 | 25.59±0.06 | 40.41±0.58 | 63.73±0.92 | 37.40±0.12 | 58.93±0.35 | 54.31±0.07 | 25.68±0.25 | 67.05±0.25 | 17.00±0.23 | 46.30±0.13 | 66.67±9.52 |
| 2:4 | Wanda | 49.79±0.26 | 35.90±0.37 | 68.16±0.29 | 64.29±0.13 | 34.16±0.06 | 55.88±0.36 | 50.88±0.23 | 25.28±0.03 | 65.25±0.10 | 17.13±0.24 | 44.70±0.06 | 38.10±12.60 |
| | *MOONSHOT*-Wanda | 45.10±0.26 | 32.47±0.50 | 61.19±0.23 | 61.60±0.86 | 34.41±0.16 | 55.51±0.66 | 52.53±0.18 | 25.60±0.38 | 65.40±0.17 | 17.20±0.61 | 44.61±0.09 | 61.90±12.60 |

(c) Llama-2-13b-chat-hf

| Sparsity | Method | C4↓ | WikiText2↓ | PTB↓ | BoolQ↑ | HellaSwag↑ | WinoGrande↑ | ARC-e↑ | ARC-c↑ | PIQA↑ | OBQA↑ | Mean↑ | Win Rate↑ |
|---|---|---|---|---|---|---|---|---|---|---|---|---|---|
| Dense | - | 8.49 | 6.11 | 50.36 | 81.65 | 60.71 | 71.11 | 77.53 | 46.16 | 77.91 | 35.20 | 64.32 | - |
| 0.7 | SparseGPT | 27.62±0.31 | 24.87±0.41 | 460.75±16.25 | 71.86±1.18 | 38.59±0.1 | 59.93±0.27 | 53.65±1.11 | 28.21±0.16 | 67.16±0.54 | 22.73±0.07 | 48.88±0.28 | 19.05±4.76 |
| | *MOONSHOT*-SparseGPT | 23.67±0.06 | 20.95±0.59 | 368.57±5.75 | 75.71±0.5 | 40.01±0.27 | 61.12±0.42 | 57.41±0.72 | 28.16±0.23 | 68.64±0.28 | 21.73±0.77 | 50.4±0.3 | 80.95±4.76 |
| 0.7 | Wanda | 46.15±0.16 | 47.85±1.0 | 629.11±9.28 | 64.05±0.04 | 32.17±0.13 | 54.22±0.28 | 43.95±0.27 | 20.71±0.17 | 61.35±0.3 | 17.0±0.12 | 41.92±0.07 | 19.05±4.76 |
| | *MOONSHOT*-Wanda | 41.0±0.25 | 38.38±1.31 | 607.22±8.48 | 66.68±0.13 | 34.11±0.08 | 54.75±0.38 | 48.22±0.27 | 21.16±0.09 | 64.4±0.1 | 14.53±0.52 | 43.41±0.1 | 80.95±4.76 |

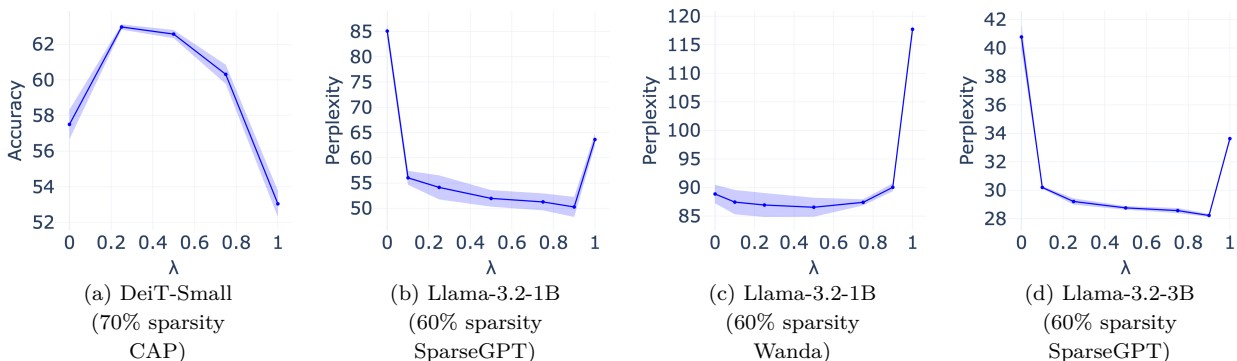

Figure 3: Performance of *MOONSHOT* across values of $\lambda$ on DeiT-Small using CAP (70% sparsity) and Llama-3.2 models using SparseGPT/Wanda (60% and 2:4 sparsity). ImageNet-1k accuracies for DeiT-Small and C4 perplexities for the Llama-3.2 models are averaged over 3 seeds with standard errors.

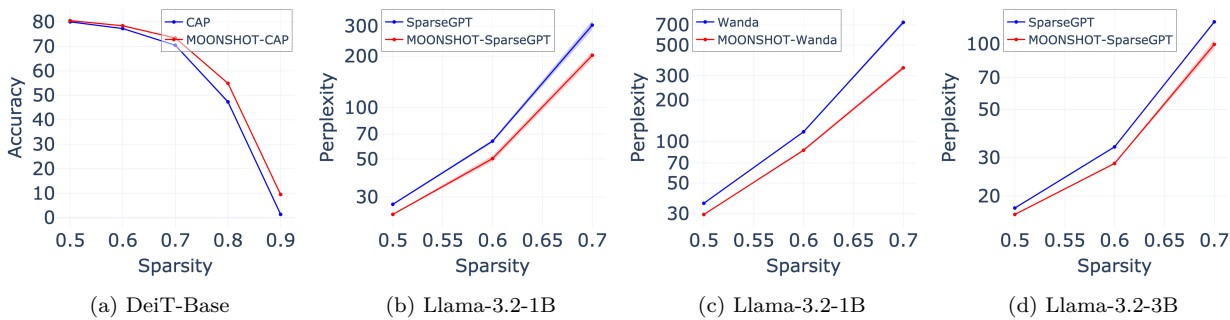

Figure 4: Impact of *MOONSHOT* across sparsity levels on CAP for DeiT-Base, and SparseGPT/Wanda on the Llama-3.2 models. ImageNet-1k accuracies for DeiT-Base and C4 perplexities for the Llama-3.2 models are averaged over 3 seeds with standard errors.

## 5    Conclusion

We present *MOONSHOT*, a framework that replaces the traditional single-objective formulation in one-shot pruning algorithms with a multi-objective approach. By incorporating both the layer-wise reconstruction loss (a local objective) and the second-order Taylor approximation of the training loss (a global objective), *MOONSHOT* significantly enhances the performance of state-of-the-art single-objective algorithms. Beyond these performance improvements, our work shows that generalizing existing pruning algorithms to a multi-objective framework can be done efficiently to scale to modern large language models, making it a compelling approach for real-world applications.

### Acknowledgments

We thank Google and Office of Naval Research for partially supporting this research. Additionally, we thank Google for providing us with Google Cloud Credits to run some of the computational experiments reported in this paper.

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

# A  Appendix

## A.1  Related Work

Many techniques have been proposed in order to prune a neural network to a desired sparsity. While some methods emphasize the use of gradual pruning to recover accuracy (Han et al., 2015; Gale et al., 2019; Singh & Alistarh, 2020a; Blalock et al., 2020; Benbaki et al., 2023), others attempt to prune during training or at initialization (Louizos et al., 2017; Frankle & Carbin, 2019; Lee et al., 2019; Liu et al., 2019; Lee et al., 2020; Wang et al., 2020; Renda et al., 2020; Frankle et al., 2021; Sui et al., 2021; Zhang et al., 2021). While effective, these methods require extensive retraining and are often too costly or impractical in resource-constrained settings with large models. Therefore, we focus on post-training one-shot pruning of large models in this work.

**Post-training One-Shot Pruning.** In the post-training one-shot pruning literature, we identify three main types of approaches that are generally proposed: (i) Magnitude-based methods (Hanson & Pratt, 1988; Mozer & Smolensky, 1989; Gordon et al., 2020) use the weight magnitudes to determine their importances and whether or not they should be pruned. Since magnitude alone may not be the best proxy for weight relevance, alternatives have been proposed. (ii) Second-order approaches, such as OBD and OBS (Optimal Brain

Damage/Surgeon) (LeCun et al., 1989a; Hassibi & Stork, 1992a), consider a local quadratic approximation of the loss around the pre-trained weights. These methods employ impact-based pruning, removing weights based on the estimated effect of their removal on the loss function. This line of work uses a second-order Taylor approximation of the training loss and the empirical Fisher information as a proxy for the Hessian. Singh & Alistarh (2020b) proposed a block-diagonal approximation of the empirical Fisher matrix for scaling the OBS framework to modern vision model sizes. Yu et al. (2022) propose to select weights to prune based on their joint rather than individual impact on the loss. (iii) Layer-wise pruning methods adapt the OBS framework to the layer-wise reconstruction objective. Dong et al. (2017) prunes each layer independently to overcome the computational challenge of computing the per-sample gradients needed in OBS. With the Optimal Brain Compression (OBC) framework, Frantar et al. (2022) adapts the OBS algorithm (Hassibi & Stork, 1992b) to the layer-wise reconstruction error and proposes rank-1 updates of the Hessian for efficient pruning.

**Pruning Vision Models.** While the pruning techniques mentioned previously are generally applicable, they have been investigated primarily in the context of Convolutional Neural Networks (CNNs). For instance, in the case of the ResNet architecture (He et al., 2015), OBC (Frantar et al., 2022) represents a state-of-the-art post-training one-shot pruning approach. Kuznedelev et al. (2023), with their Correlation Aware Pruner (CAP), adapted the greedy OBS algorithm used in OBC (Frantar et al., 2022) to Vision Transformers. This approach is a state-of-the-art method for post-training one-shot pruning in Vision Transformers.

**Pruning Large Language Models.** Pruning is particularly important for Large Language Models, which can have billions of parameters. In the case of unstructured and semi-structured pruning, state-of-the-art methods include SparseGPT (Frantar & Alistarh, 2023) and Wanda (Sun et al., 2024), both of which build on the OBC/OBS (Frantar et al., 2022; Hassibi & Stork, 1992b) framework. To make pruning more scalable, SparseGPT prunes the weight matrix by groups of columns. This enables to reduce the number of computations during pruning by considering only the Hessian for the weights within the support. However, SparseGPT still relies on a block-diagonal approximation of the Hessian. Wanda simplifies this further by using a diagonal matrix instead. Both methods focus solely on minimizing the layer-wise reconstruction loss. In the case of structured pruning, ZipLM (Kurtic et al., 2023) uses the layer-wise reconstruction error to guide the pruning of attention heads and projection layers. Building on a careful reformulation of this objective, Meng et al. (2024b) introduce OSSCAR, a more scalable and stronger baseline for one-shot structured pruning. OSSCAR greedily selects columns for removal to reduce the reconstruction objective, and optionally applies a local-search refinement step. OSSCAR is a state-of-the-art one-shot structured pruning method for LLMs.

**Non-Uniform Sparsity.** LLMs typically suffer significant degradation beyond 50% uniform sparsity. This has motivated recent work on non-uniform sparsity for LLMs, which assigns different sparsity levels to different layers to better preserve model quality under the same global sparsity constraint. OWL (Yin et al., 2024) improves pruning by allocating layer-wise sparsity based on the distribution of outlier weights, leading to notable gains for methods like SparseGPT and Wanda. AlphaPruning (Lu et al., 2024) takes a more principled approach and uses heavy-tailed self-regularization theory to measure how well each layer is trained. After quantifying the heavy-tail distribution of the weights, it assigns lower sparsity to the better trained layers.

## A.2 *MOONSHOT*-SparseGPT Algorithm

We present below our adaptation of the SparseGPT algorithm (Frantar & Alistarh, 2023), denoted *MOONSHOT*-SparseGPT.

A key component of SparseGPT is the use of the Hessian inverse. In the multi-objective setting, however, directly computing matrix inverses for every block is computationally infeasible. To overcome this, we replace Step 2 in Algorithm 2 with the more efficient procedure outlined in Algorithm 1. We also note that the larger Hessian size makes it challenging to prune all blocks simultaneously. To address this, we perform pruning in parallel over $K_p$ blocks at a time. This means that the uniform sparsity budget is applied within each group of $K_p$ blocks rather than across the entire weight matrix. While this introduces a more local form of sparsity allocation, choosing $K_p$ sufficiently large (all the blocks for Llama-3.2-1B and at least 50% of them for Llama-3.2-3B) ensures that the effect is minimal in practice.

---

**Algorithm 2** *MOONSHOT*-SparseGPT

---

**Input:** Layer weight matrix $\widehat{W}^{(l)} \in \mathbb{R}^{d_{\text{out}}^{(l)} \times d_{\text{in}}^{(l)}}$, layer input matrix $X^{(l)} \in \mathbb{R}^{d_{\text{in}}^{(l)} \times N}$, per-sample gradients $A_k^{(l)} \in \mathbb{R}^{d_{\text{in}}^{(l)} \times N}$ for each block $k = 1, \ldots, d_{\text{out}}^{(l)}$, multi-objective weight $\lambda \in [0,1]$, lazy batch-update block-size B, adaptive mask selection blocksize $B_s$, number of blocks to prune in parallel $K_P$ and sparsity level $p$.

1: Initialize Pruned Weights: $W^{(l)} \leftarrow \widehat{W}^{(l)}$
2: Initialize Binary Pruning Mask: $M \leftarrow \mathbb{1}_{d_{\text{out}}^{(l)} \times d_{\text{in}}^{(l)}}$
3: Initialize Block Errors: $E \leftarrow \mathbb{0}_{d_{\text{out}}^{(l)} \times B}$
4: **for** $k_p = 0, K_p, 2K_P, \ldots$ **do**
5:  Compute Hessian Inverse $\{G_k^{(l)}\}_{k=k_p}^{k_p+K_p} = \{F_k^{(l)^{-1}}\}_{k=k_p}^{k_p+K_p}$ using Algorithm 1 (tensor format $G^{(l)} \in \mathbb{R}^{K_p \times d_{\text{in}}^{(l)} \times d_{\text{in}}^{(l)}}$)
6:  Compute Cholesky Decomposition for each block $G_k^{(l)} \leftarrow \text{Cholesky}(G_k^{(l)})^T$
7:  **for** $i = 0, B, 2B, \ldots$ **do**
8:   **for** $j = i, \ldots i + B - 1$ **do**
9:    **if** $j \bmod B_s = 0$ **then**
10:     $M_{k_p:k_p+K_p, j:(j+B_s)} \leftarrow$ mask of $(1-p)\%$ weights $w_c \in W_{k_p:k_p+K_p, j:(j+B_s)}^{(l)}$
11:      with largest $w_c^2 / \left[G^{(l)}\right]^2_{k_p:k_p+K_p, c, c}$
12:    Compute Pruning Errors: $E_{k_p:k_p+K_p, j-i} \leftarrow W_{k_p:k_p+K_p, j}^{(l)} / \left[G^{(l)}\right]_{k_p:k_p+K_p, j, j}$
13:    Freeze Weights: $E_{k_p:k_p+K_p, i, j-i} \leftarrow \left(\mathbb{1}_{d_{\text{out}}^{(l)} \times d_{\text{in}}^{(l)}} - M_{k_p:k_p+K_p, j}\right) \cdot E_{k_p:k_p+K_p, j-i}$
14:    Weights update: $W_{k_p:k_p+K_p, j:(i+B)}^{(l)} \leftarrow W_{k_p:k_p+K_p, j:(i+B)}^{(l)} - E_{k_p:k_p+K_p, j-i} \cdot G_{k_p:k_p+K_p, j, j:(i+B)}^{(l)}$
15:   Weights update: $W_{k_p:k_p+K_p, (i+B):}^{(l)} \leftarrow W_{k_p:k_p+K_p, (i+B):}^{(l)} - E_{k_p:k_p+K_p, :} \cdot G_{k_p:k_p+K_p, i:(i+B), (i+B):}^{(l)}$
16:  Set pruned weights to 0: $W_{k_p:k_p+K_p, :}^{(l)} \leftarrow W_{k_p:k_p+K_p, :}^{(l)} \cdot M_{k_p:k_p+K_p, :}$

**Output:** Pruned weights $W^{(l)}$

---

### A.3 Woodbury Update in 1

Let $A \in \mathbb{R}^{n \times n}$, $C \in \mathbb{R}^{k \times k}$, $U \in \mathbb{R}^{n \times k}$ and $V \in \mathbb{R}^{k \times n}$. The Woodbury matrix identity (Woodbury & of Statistics, 1950) gives us that:

$$(A + UCV)^{-1} = A^{-1} - A^{-1}U(C + VA^{-1}U)^{-1}VA^{-1} \tag{12}$$

In our case, we want to compute $G_k^{(l)} = (F_k^{(l)})^{-1} = \left(\frac{\lambda}{\mathcal{L}_R^{(l)}(0)} L_k^{(l)} + \frac{1-\lambda}{\mathcal{L}_F^{(l)}(0)} H_k^{(l)}\right)^{-1}$

We focus in Algorithm 1 on methods like SparseGPT and OSSCAR, which use the layer-wise reconstruction error objective to scale to billion-parameter-LLMs (without further diagonal approximation on the exact block-diagonal Hessian). For these pruning baselines, $L_1^{(l)} = \cdots = L_K^{(l)} = X^{(l)}(X^{(l)})^T$ (with $K = d_{\text{out}}^{(l)}$). In addition, as seen in Section 2.3, we can write $H_k^{(l)} = \frac{1}{N} A_k^{(l)} A_k^{(l)^T}$. Therefore, $G_k^{(l)}$ can be rewritten as:

$$G_k^{(l)} = \left(\frac{\lambda}{\mathcal{L}_R^{(l)}(0)} X^{(l)}(X^{(l)})^T + \frac{1-\lambda}{N\mathcal{L}_F^{(l)}(0)} A_k^{(l)} A_k^{(l)^T}\right)^{-1} \tag{13}$$

This is the same form as equation 12 with:

$$A = \frac{\lambda}{\mathcal{L}_R^{(l)}(0)} X^{(l)}(X^{(l)})^T = J_0^{-1}, \qquad U = \frac{1-\lambda}{N\mathcal{L}_F^{(l)}(0)} A_k^{(l)}, \qquad C = I_N, \qquad V = \left(A_k^{(l)}\right)^T$$

Table 4: Pruning times (in seconds) of *MOONSHOT* over 3 seeds, for different values of $\lambda$, architectures, and pruning baselines.

| Sparsity | $\lambda$ | OBC | CAP | | | Wanda | | SparseGPT | |
|---|---|---|---|---|---|---|---|---|---|
| | | ResNet-50 | DeiT-Tiny | DeiT-Small | DeiT-Base | Llama-3.2-1B | Llama-3.2-3B | Llama-3.2-1B | Llama-3.2-3B |
| Unstruc-tured | 0.00 | $7360.65 \pm 8.35$ | $71.89 \pm 7.91$ | $268.58 \pm 23.98$ | $1303.13 \pm 22.99$ | $69.59 \pm 0.41$ | $403.51 \pm 37.9$ | $473.03 \pm 1.95$ | $2181.89 \pm 128.72$ |
| | 0.25 | $7392.68 \pm 23.96$ | $65.63 \pm 1.12$ | $240.06 \pm 1.73$ | $1186.93 \pm 9.11$ | $68.94 \pm 0.47$ | $407.2 \pm 38.62$ | $474.86 \pm 1.48$ | $2269.66 \pm 94.02$ |
| | 0.50 | $7370.94 \pm 1.0$ | $66.77 \pm 1.51$ | $226.0 \pm 6.68$ | $1185.66 \pm 14.01$ | $69.63 \pm 0.64$ | $464.06 \pm 48.64$ | $472.11 \pm 1.07$ | $2287.58 \pm 88.63$ |
| | 0.75 | $7389.37 \pm 9.43$ | $67.58 \pm 2.02$ | $227.61 \pm 0.77$ | $1257.79 \pm 87.92$ | $70.23 \pm 0.83$ | $420.58 \pm 47.86$ | $471.77 \pm 0.19$ | $2165.25 \pm 171.46$ |
| | 1.00 | $\mathbf{7146.09} \pm 5.34$ | $\mathbf{21.62} \pm 1.15$ | $\mathbf{43.07} \pm 0.89$ | $\mathbf{247.25} \pm 3.04$ | $\mathbf{21.0} \pm 0.36$ | $\mathbf{70.66} \pm 7.23$ | $\mathbf{73.9} \pm 0.29$ | $\mathbf{307.73} \pm 11.31$ |
| Semi-Structured (2:4) | 0.00 | $3965.42 \pm 13.45$ | $60.19 \pm 0.9$ | $229.22 \pm 2.39$ | $1236.61 \pm 8.72$ | $71.0 \pm 0.54$ | $413.24 \pm 53.15$ | $478.6 \pm 1.14$ | $2197.32 \pm 142.89$ |
| | 0.25 | $4033.31 \pm 85.1$ | $62.06 \pm 1.11$ | $217.56 \pm 1.78$ | $1123.35 \pm 13.82$ | $71.62 \pm 0.39$ | $442.05 \pm 35.45$ | $477.58 \pm 0.25$ | $2345.26 \pm 86.07$ |
| | 0.50 | $3956.29 \pm 4.2$ | $62.35 \pm 1.41$ | $218.5 \pm 2.5$ | $1110.06 \pm 9.58$ | $72.62 \pm 1.04$ | $474.96 \pm 6.32$ | $476.66 \pm 2.13$ | $2362.17 \pm 59.99$ |
| | 0.75 | $3974.67 \pm 5.72$ | $65.18 \pm 4.0$ | $213.7 \pm 2.33$ | $1102.55 \pm 3.82$ | $73.87 \pm 0.33$ | $433.84 \pm 77.23$ | $477.92 \pm 0.51$ | $2237.57 \pm 216.16$ |
| | 1.00 | $\mathbf{3737.4} \pm 5.68$ | $\mathbf{16.31} \pm 1.21$ | $\mathbf{29.38} \pm 0.1$ | $\mathbf{178.98} \pm 2.99$ | $\mathbf{22.74} \pm 0.11$ | $\mathbf{80.78} \pm 6.02$ | $\mathbf{79.01} \pm 0.13$ | $\mathbf{333.86} \pm 13.01$ |

Therefore, using equation 12, equation 13 becomes:

$$
G_k^{(l)} = \left( \frac{\lambda}{\mathcal{L}_R^{(l)}(0)} X^{(l)}(X^{(l)})^T + \frac{1-\lambda}{N\mathcal{L}_F^{(l)}(0)} A_k^{(l)} A_k^{(l)T} \right)^{-1}
$$

$$
= J_0 - J_0 \left( \frac{1-\lambda}{N\mathcal{L}_F^{(l)}(0)} A_k^{(l)} \right) \left( I_N + A_k^{(l)T} J_0 \left( \frac{1-\lambda}{N\mathcal{L}_F^{(l)}(0)} \right) A_k^{(l)} \right)^{-1} A_k^{(l)T} J_0
$$

This expression of $G_k^{(l)}$ is the same as the one used in Algorithm 1.

### A.4 Efficient backsolve for OSSCAR

After greedy selection, OSSCAR performs a backsolve to update the weights on the support of remaining columns $S$. With the original single-objective, the Hessian has repeated blocks, yielding the closed form

$$
W_{S,:}^* = [XX^\top]_{S,S}^{-1}[XX^\top]_{S,:}\widehat{W}.
$$

With *MOONSHOT* (when $\lambda \neq 1$), the Hessian becomes block-diagonal with row-dependent blocks, $H = \mathrm{Diag}(F_1, \ldots, F_K)$, where each $F_k$ has the same shape as $XX^\top$. A direct extension would require inverting all $K$ matrices, which is unnecessarily expensive. Instead, we compute the Cholesky decomposition for each block and use an efficient solver in PyTorch for the system:

$$
\mathrm{Diag}\big([F_1]_{S,S}, \ldots, [F_K]_{S,S}\big) \mathrm{vec}(W_{S,:}^*) = H_{S,:} \mathrm{vec}(\widehat{W}),
$$

where $\mathrm{vec}(W)$ denotes the vector form of the weight matrix $W$.

### A.5 Pruning Time

The pruning times obtained when applying *MOONSHOT* on the different baselines are described in Table 4. OBC and CAP allow pruning at multiple sparsity levels in a single run, with minimal additional cost compared to pruning at a single level. For these methods, we report the total pruning time needed to obtain pruned weights for all target sparsities: 0.5, 0.6, 0.7, 0.8, and 0.9. SparseGPT and Wanda only support pruning one sparsity level at a time, but their runtime is not affected by the chosen sparsity. We therefore report their pruning time at sparsity 0.6.

We note that the baselines correspond to $\lambda = 0$ for CAP and $\lambda = 1$ for OBC, Wanda and SparseGPT. We observe that *MOONSHOT* incur no to almost no additional computational overhead to CAP and OBC for the DeiT models and ResNet-50 respectively. With *MOONSHOT*-SparseGPT, we achieve pruning times of under 40 minutes for Llama-3.2-3B and under 8 minutes for Llama-3.2-1B. While this is a slowdown compared to SparseGPT's original pruning times of 5 and 1.2 minutes respectively, the increase is reasonable given the significant performance improvements. Moreover, since the ultimate goal is to deploy a more compact and

efficient model post-pruning, the slightly longer pruning time, viewed as a one-time cost, is a worthwhile trade-off. For Wanda, pruning times are similarly manageable, with 8 minutes for Llama-3.2-3B and 1.2 minutes for Llama-3.2-1B, compared to 1.2 minutes and 20 seconds respectively for the single-objective version.

## A.6 Additional Hyperparameters

**Dampening Term**. As noted in previous work (Frantar et al., 2022; Frantar & Alistarh, 2023; Kuznedelev et al., 2023), the Hessian is not always invertible in practice. To address this issue, a common approach is to add a dampening factor $\mu$ to the diagonal of the Hessian, ensuring it is positive definite. However, selecting $\mu$ is non-trivial: if $\mu$ is too small, numerical instabilities can persist, while a large $\mu$ may degrade algorithm performance. Following SparseGPT (Frantar & Alistarh, 2023), we set $\mu$ to 1% of the mean of the diagonal elements of the Hessian. For OBC (Frantar et al., 2022) and CAP (Kuznedelev et al., 2023), we use the diagonal of each block $F_k^{(l)}$ in the multi-objective formulation equation 10, while for SparseGPT, we use the diagonal of $X^{(l)}(X^{(l)})^T$ only to maintain the efficiency of the inverse Hessian computation described in Section 2.3. For Wanda (Sun et al., 2024) only, $\mu$ is set to 0 as recommended by the authors (no inverse Hessian is required).

**OWL and AlphaPruning.** Following the optimal parameters found by the authors of OWL (Yin et al., 2024) on Llama-7B, we use $\lambda^{(\mathrm{OWL})} = 0.08$ and $M = 5$. For AlphaPruning, we fixed $\tau = 0.05$ for Llama-3.2-1B and $\tau = 0.1$ for Llama-3.2-3B.

## A.7 Hessian Recomputation

In this section, we report additional results with Hessian recomputation after each block of layers for SparseGPT and Wanda. Following prior Fisher-based work (Benbaki et al., 2023), we denote these variants as *MOONSHOT*-SparseGPT++ and *MOONSHOT*-Wanda++. Although more costly than computing the Hessian once, our implementation remains tractable: we compute per-sample gradients for one block of layers at a time and update the dataset after pruning each block to reduce further gradient costs (using only the remaining dense layers). With this approach, *MOONSHOT*-SparseGPT++ prunes Llama-3.2-1B in under 11 minutes (vs. under 2 minutes for standard SparseGPT), and *MOONSHOT*-Wanda++ prunes Llama-3.2-1B in under 4 minutes (vs. around 1 minute for standard Wanda) on a single L40 GPU (40 GB). Comprehensive results are presented in Tables 5 and 6.

Table 5: Test perplexity on C4, WikiText2 and PTB and zero-shot accuracy of Llama-3.2-1B using *MOONSHOT*-SparseGPT++. Zero-shot mean performance and win rate are computed over the 7 classification tasks. Results are averaged over 3 seeds with standard errors.

| Sparsity | λ | C4 ↓ | WikiText2 ↓ | PTB ↓ | BoolQ ↑ | HellaSwag ↑ | WinoGrande ↑ | ARC-e ↑ | ARC-c ↑ | PIQA ↑ | OBQA ↑ | Mean ↑ | Win Rate ↑ |
|---|---|---|---|---|---|---|---|---|---|---|---|---|---|
| Dense | - | 14.02 | 9.75 | 17.59 | 64.01 | 47.73 | 60.14 | 65.15 | 31.23 | 74.32 | 26.4 | 52.71 | - |
| 0.5 | 0.0 | 27.37 ± 0.02 | 20.3 ± 0.03 | 33.19 ± 0.31 | 62.25 ± 0.19 | 37.0 ± 0.15 | 55.01 ± 0.88 | 53.47 ± 0.4 | 24.43 ± 0.8 | 67.43 ± 0.08 | 18.0 ± 0.12 | 45.37 ± 0.14 | 9.52 ± 9.52 |
|  | 0.1 | 24.02 ± 0.1 | 17.71 ± 0.11 | 28.99 ± 0.26 | 62.5 ± 0.26 | 38.37 ± 0.03 | 55.3 ± 0.37 | 56.17 ± 0.3 | 25.63 ± 0.19 | **68.57** ± 0.19 | 19.2 ± 0.12 | 46.53 ± 0.04 | 28.57 ± 8.25 |
|  | 0.25 | 23.8 ± 0.12 | 17.57 ± 0.01 | 28.86 ± 0.02 | **63.03** ± 0.11 | 38.71 ± 0.03 | 54.67 ± 0.37 | 55.2 ± 0.26 | 26.65 ± 0.08 | 68.17 ± 0.17 | 20.33 ± 0.44 | 46.68 ± 0.02 | 33.33 ± 12.6 |
|  | 0.5 | 23.6 ± 0.06 | 17.38 ± 0.02 | 28.72 ± 0.12 | 62.8 ± 0.46 | 38.91 ± 0.05 | 55.14 ± 0.16 | 55.61 ± 0.52 | 25.97 ± 0.33 | 68.41 ± 0.22 | 19.73 ± 0.57 | 46.65 ± 0.07 | 23.81 ± 12.6 |
|  | 0.75 | 23.45 ± 0.06 | **17.23** ± 0.03 | 28.43 ± 0.11 | 62.94 ± 0.35 | 38.93 ± 0.05 | 55.96 ± 0.34 | 55.56 ± 0.06 | **27.05** ± 0.34 | 67.85 ± 0.19 | 19.8 ± 0.46 | 46.87 ± 0.07 | 42.86 ± 8.25 |
|  | 0.9 | **23.39** ± 0.05 | **17.23** ± 0.02 | **28.16** ± 0.16 | 62.4 ± 0.14 | 39.11 ± 0.12 | 55.72 ± 0.39 | **56.44** ± 0.26 | 26.22 ± 0.16 | 68.55 ± 0.42 | 20.73 ± 0.93 | 47.03 ± 0.1 | **47.62** ± 26.51 |
|  | 1.0 | 26.35 ± 0.05 | 19.26 ± 0.23 | 30.72 ± 0.2 | 62.86 ± 1.0 | **39.27** ± 0.06 | **56.2** ± 0.06 | 54.64 ± 0.06 | 26.62 ± 0.3 | 68.44 ± 0.11 | **21.27** ± 0.24 | **47.04** ± 0.24 | - |
| 0.6 | 0.0 | 59.74 ± 0.72 | 48.97 ± 0.26 | 72.66 ± 3.54 | 60.47 ± 0.82 | 30.38 ± 0.19 | 51.43 ± 0.62 | 41.68 ± 0.58 | 20.73 ± 0.57 | 60.57 ± 0.13 | 14.13 ± 0.48 | 39.92 ± 0.34 | 4.76 ± 4.76 |
|  | 0.1 | 47.21 ± 0.16 | 37.08 ± 0.66 | 53.82 ± 0.38 | 62.1 ± 0.1 | 31.97 ± 0.12 | 52.22 ± 0.25 | 45.86 ± 0.98 | 21.56 ± 0.54 | 63.02 ± 0.09 | 15.6 ± 0.6 | 41.76 ± 0.25 | 52.38 ± 9.52 |
|  | 0.25 | 46.93 ± 0.41 | 36.73 ± 0.2 | 52.18 ± 0.65 | 61.89 ± 0.27 | 32.34 ± 0.11 | 52.04 ± 0.13 | 45.9 ± 0.62 | 21.9 ± 0.5 | 63.44 ± 0.65 | 15.73 ± 0.29 | 41.89 ± 0.09 | 52.38 ± 4.76 |
|  | 0.5 | 46.31 ± 0.42 | 35.07 ± 0.29 | 50.84 ± 0.64 | 62.16 ± 0.15 | 32.5 ± 0.13 | **53.01** ± 0.55 | 45.66 ± 0.71 | 21.53 ± 0.2 | **63.89** ± 0.13 | 16.47 ± 0.66 | 42.17 ± 0.3 | **71.43** ± 14.29 |
|  | 0.75 | 46.19 ± 0.4 | **34.85** ± 0.78 | 51.53 ± 1.17 | 62.05 ± 0.07 | **32.68** ± 0.11 | 52.64 ± 0.32 | **46.65** ± 0.38 | **21.96** ± 0.33 | 63.51 ± 0.53 | 16.2 ± 0.35 | **42.24** ± 0.13 | 57.14 ± 8.25 |
|  | 0.9 | **46.08** ± 0.79 | 35.6 ± 0.44 | **50.36** ± 1.23 | 61.67 ± 0.35 | 32.55 ± 0.16 | 52.67 ± 0.64 | 46.34 ± 0.55 | 21.53 ± 0.52 | 63.46 ± 0.64 | 16.87 ± 0.18 | 42.16 ± 0.14 | 66.67 ± 12.6 |
|  | 1.0 | 61.36 ± 0.86 | 49.75 ± 1.61 | 64.21 ± 1.1 | **62.35** ± 0.12 | 31.76 ± 0.23 | 52.64 ± 0.16 | 44.4 ± 0.59 | 21.9 ± 0.58 | 62.37 ± 0.56 | **17.67** ± 0.55 | 41.87 ± 0.11 | - |
| 0.7 | 0.0 | 168.52 ± 1.54 | 156.79 ± 3.73 | 190.61 ± 10.89 | 54.75 ± 3.4 | 27.18 ± 0.16 | 50.07 ± 0.16 | 32.72 ± 0.72 | 18.66 ± 0.59 | 56.4 ± 0.07 | 11.73 ± 0.33 | 35.93 ± 0.52 | 28.57 ± 0.0 |
|  | 0.1 | 130.06 ± 1.93 | 107.57 ± 2.09 | 135.44 ± 4.14 | 61.88 ± 0.1 | 27.76 ± 0.06 | **50.83** ± 0.46 | 34.09 ± 0.18 | 18.69 ± 0.6 | 57.13 ± 0.2 | 11.53 ± 0.37 | 37.41 ± 0.2 | 47.62 ± 17.17 |
|  | 0.25 | 127.78 ± 1.31 | 103.16 ± 1.48 | **125.71** ± 4.58 | 61.47 ± 0.26 | 27.78 ± 0.1 | 49.3 | **34.83** ± 0.42 | 17.55 ± 0.51 | 56.71 ± 0.15 | 12.2 ± 0.35 | 37.12 ± 0.08 | 23.81 ± 4.76 |
|  | 0.5 | **127.22** ± 1.21 | **101.14** ± 0.6 | 130.32 ± 3.42 | 60.84 ± 0.58 | 27.73 ± 0.06 | 49.46 ± 0.55 | 34.39 ± 0.68 | 18.43 ± 0.31 | **57.25** ± 0.33 | 11.6 ± 0.6 | 37.1 ± 0.1 | 38.1 ± 12.6 |
|  | 0.75 | 132.13 ± 0.92 | 103.3 ± 2.55 | 130.82 ± 5.53 | **62.04** ± 0.12 | **27.94** ± 0.09 | 49.8 ± 0.27 | 34.5 ± 0.07 | **18.8** ± 0.37 | 56.64 ± 0.35 | 11.47 ± 0.37 | 37.31 ± 0.13 | 47.62 ± 9.52 |
|  | 0.9 | 135.42 ± 1.28 | 105.49 ± 1.42 | 134.09 ± 2.7 | 61.31 ± 0.59 | 27.92 ± 0.06 | 50.78 ± 0.43 | 34.57 ± 0.19 | 18.52 ± 0.21 | 57.18 ± 0.23 | 11.27 ± 0.27 | 37.36 ± 0.12 | **57.14** ± 16.5 |
|  | 1.0 | 159.35 ± 11.39 | 143.16 ± 4.52 | 174.23 ± 10.38 | 61.44 ± 0.45 | 27.86 ± 0.04 | **50.83** ± 0.41 | 34.53 ± 0.68 | 18.71 ± 0.47 | 56.0 ± 0.41 | **13.67** ± 0.18 | **37.58** ± 0.08 | - |
| 2:4 | 0.0 | 53.47 ± 0.38 | 40.62 ± 0.39 | 64.24 ± 0.82 | 62.09 ± 0.22 | 30.43 ± 0.1 | 52.59 ± 0.59 | 42.61 ± 0.2 | 19.94 ± 0.28 | 60.75 ± 0.35 | 14.2 ± 0.72 | 40.37 ± 0.04 | 19.05 ± 4.76 |
|  | 0.1 | 43.81 ± 0.17 | 32.47 ± 0.2 | 51.62 ± 0.17 | 62.16 ± 0.1 | 31.67 ± 0.11 | 53.14 ± 0.55 | 44.39 ± 0.87 | 20.16 ± 0.38 | 62.3 ± 0.32 | 14.93 ± 0.48 | 41.25 ± 0.2 | 23.81 ± 4.76 |
|  | 0.25 | 43.39 ± 0.09 | **32.08** ± 0.31 | 51.18 ± 0.48 | **62.21** ± 0.12 | 31.72 ± 0.03 | 53.62 ± 0.89 | 46.04 ± 0.17 | 19.94 ± 0.33 | 62.01 ± 0.09 | 15.2 ± 0.95 | 41.53 ± 0.09 | **47.62** ± 4.76 |
|  | 0.5 | **43.07** ± 0.12 | 32.2 ± 0.48 | 50.95 ± 0.2 | 62.11 ± 0.12 | 31.83 ± 0.05 | 53.88 ± 0.43 | 46.07 ± 0.48 | 21.02 ± 0.35 | 61.95 ± 0.17 | 15.13 ± 0.18 | 41.71 ± 0.11 | **47.62** ± 9.52 |
|  | 0.75 | 43.19 ± 0.22 | 32.17 ± 0.07 | 50.55 ± 0.57 | 61.95 ± 0.14 | 31.86 ± 0.06 | 53.67 ± 0.12 | 45.62 ± 0.61 | 20.56 ± 0.3 | **62.42** ± 0.17 | 14.67 ± 0.57 | 41.54 ± 0.11 | 42.86 ± 14.29 |
|  | 0.9 | 43.27 ± 0.1 | **32.08** ± 0.29 | **50.51** ± 0.65 | 61.75 ± 0.48 | 31.8 ± 0.08 | 54.72 ± 0.65 | **46.34** ± 0.17 | 20.42 ± 0.58 | 61.64 ± 0.21 | 14.87 ± 0.44 | 41.65 ± 0.07 | 33.33 ± 4.76 |
|  | 1.0 | 45.73 ± 0.77 | 34.12 ± 0.55 | 52.81 ± 0.61 | 61.64 ± 0.39 | **32.18** ± 0.19 | **54.93** ± 0.57 | 45.74 ± 0.91 | **21.5** ± 0.69 | 61.81 ± 0.17 | **16.2** ± 0.92 | **42.0** ± 0.4 | - |

Table 6: Test perplexity on C4, WikiText2 and PTB and zero-shot accuracy of Llama-3.2-1B using *MOON-SHOT*-Wanda++. Zero-shot mean performance and win rate are computed over the 7 classification tasks. Results are averaged over 3 seeds with standard errors.

| Sparsity | λ | C4 ↓ | WikiText2 ↓ | PTB ↓ | BoolQ ↑ | HellaSwag ↑ | WinoGrande ↑ | ARC-e ↑ | ARC-c ↑ | PIQA ↑ | OBQA ↑ | Mean ↑ | Win Rate ↑ |
|---|---|---|---|---|---|---|---|---|---|---|---|---|---|
| Dense | - | 14.02 | 9.75 | 17.59 | 64.01 | 47.73 | 60.14 | 65.15 | 31.23 | 74.32 | 26.4 | 52.71 | - |
| 0.5 | 0.0 | 31.13 ± 0.14 | 21.86 ± 0.1 | 38.27 ± 0.35 | 61.99 ± 0.08 | 35.08 ± 0.06 | 54.35 ± 0.09 | 54.45 ± 0.16 | 24.09 ± 0.38 | 66.09 ± 0.23 | 16.6 ± 0.23 | 44.66 ± 0.1 | 66.67 ± 4.76 |
|  | 0.1 | 30.56 ± 0.26 | 21.46 ± 0.15 | 37.5 ± 0.53 | 62.12 ± 0.12 | 35.3 ± 0.07 | 54.09 ± 0.41 | 54.91 ± 0.43 | 24.29 ± 0.14 | 66.74 ± 0.13 | 16.47 ± 0.24 | 44.84 ± 0.05 | 66.67 ± 4.76 |
|  | 0.25 | 30.37 ± 0.09 | 21.13 ± 0.06 | 37.4 ± 0.32 | 61.96 ± 0.12 | 35.37 ± 0.01 | 54.43 ± 0.53 | **54.97** ± 0.46 | 24.6 ± 0.25 | **66.79** ± 0.19 | 17.53 ± 0.47 | 45.09 ± 0.07 | 80.95 ± 4.76 |
|  | 0.5 | 30.08 ± 0.21 | 20.83 ± 0.06 | 36.74 ± 0.46 | 62.08 ± 0.12 | 35.59 ± 0.02 | **54.93** ± 0.41 | 54.77 ± 0.27 | 25.06 ± 0.2 | 66.36 ± 0.23 | **17.67** ± 0.29 | 45.21 ± 0.07 | **95.24** ± 4.76 |
|  | 0.75 | 29.91 ± 0.26 | 20.61 ± 0.17 | **36.63** ± 0.45 | 62.13 ± 0.14 | 35.73 ± 0.07 | 54.62 ± 0.43 | 54.45 ± 0.19 | **25.26** ± 0.13 | 66.61 ± 0.13 | 17.6 ± 0.0 | 45.2 ± 0.09 | 90.48 ± 9.52 |
|  | 0.9 | **29.78** ± 0.2 | **20.54** ± 0.1 | 36.69 ± 0.3 | **62.22** ± 0.06 | **35.85** ± 0.03 | 54.56 ± 0.56 | 54.95 ± 0.2 | 24.94 ± 0.15 | 66.63 ± 0.13 | 17.13 ± 0.57 | **45.23** ± 0.09 | 90.48 ± 9.52 |
|  | 1.0 | 34.21 ± 0.21 | 23.47 ± 0.14 | 40.89 ± 0.37 | 61.3 ± 0.34 | 35.1 ± 0.07 | 54.56 ± 0.56 | 51.61 ± 0.14 | 24.37 ± 0.24 | 65.07 ± 0.19 | 17.4 ± 0.42 | 44.2 ± 0.13 | - |
| 0.6 | 0.0 | 93.22 ± 0.76 | 65.58 ± 0.41 | 91.86 ± 1.37 | 61.99 ± 0.1 | 28.55 ± 0.06 | 51.07 ± 0.43 | 39.63 ± 0.04 | 18.83 ± 0.52 | 58.96 ± 0.38 | 12.87 ± 0.41 | 38.84 ± 0.13 | 76.19 ± 9.52 |
|  | 0.1 | 90.14 ± 1.95 | 62.95 ± 0.88 | 90.3 ± 0.91 | 62.16 ± 0.04 | 28.75 ± 0.08 | 50.99 ± 0.25 | 40.28 ± 0.21 | 19.37 ± 0.18 | 59.7 ± 0.5 | 12.6 ± 0.2 | 39.12 ± 0.1 | 85.71 ± 8.25 |
|  | 0.25 | 87.37 ± 1.18 | **62.37** ± 0.47 | 87.48 ± 1.14 | **62.17** ± 0.06 | 28.85 ± 0.12 | 51.51 ± 0.89 | 40.17 ± 0.18 | 18.94 ± 0.05 | 59.5 ± 0.18 | 12.73 ± 0.13 | 39.13 ± 0.14 | 80.95 ± 4.76 |
|  | 0.5 | **87.1** ± 1.68 | 62.86 ± 0.3 | 88.14 ± 1.31 | 62.08 ± 0.06 | 28.89 ± 0.06 | 51.38 ± 0.3 | 39.76 ± 0.22 | 19.28 ± 0.05 | 59.9 ± 0.11 | 13.47 ± 0.29 | 39.25 ± 0.06 | 90.48 ± 4.76 |
|  | 0.75 | 87.58 ± 0.79 | 62.94 ± 1.14 | **85.97** ± 0.77 | 62.06 ± 0.13 | 29.0 ± 0.05 | **51.67** ± 0.07 | 40.01 ± 0.2 | 19.25 ± 0.25 | 59.7 ± 0.12 | 13.2 ± 0.2 | 39.27 ± 0.01 | 90.48 ± 4.76 |
|  | 0.9 | 89.06 ± 0.44 | 63.87 ± 0.25 | 89.52 ± 0.28 | 61.62 ± 0.34 | **29.13** ± 0.03 | 50.41 ± 0.21 | **40.71** ± 0.07 | **19.54** ± 0.1 | **60.08** ± 0.14 | **13.6** ± 0.12 | **39.3** ± 0.11 | 90.48 ± 4.76 |
|  | 1.0 | 109.62 ± 1.07 | 79.05 ± 0.37 | 107.92 ± 1.11 | 58.75 ± 1.04 | 28.43 ± 0.07 | 50.49 ± 0.48 | 37.61 ± 0.24 | 19.31 ± 0.19 | 58.03 ± 0.13 | 12.73 ± 0.18 | 37.91 ± 0.2 | - |
| 0.7 | 0.0 | 382.23 ± 8.94 | 274.94 ± 10.62 | 298.74 ± 11.35 | 42.25 ± 0.96 | 26.9 ± 0.04 | 48.67 ± 0.56 | **30.35** ± 0.36 | 18.43 ± 0.3 | 54.95 ± 0.13 | 11.4 ± 0.12 | 33.28 ± 0.05 | 57.14 ± 8.25 |
|  | 0.1 | 352.6 ± 7.68 | 254.96 ± 2.39 | 274.48 ± 14.3 | 39.62 ± 0.51 | 26.9 ± 0.01 | 49.2 ± 0.33 | 30.29 ± 0.2 | 18.32 ± 0.27 | 55.11 ± 0.14 | **12.13** ± 0.18 | 33.08 ± 0.18 | 57.14 ± 8.25 |
|  | 0.25 | 368.59 ± 38.77 | 251.25 ± 22.88 | 277.37 ± 16.15 | **45.3** ± 3.24 | **27.02** ± 0.02 | 49.78 ± 0.29 | 29.97 ± 0.32 | 19.08 ± 0.12 | 54.9 ± 0.22 | 11.93 ± 0.27 | 33.03 ± 0.23 | 52.38 ± 4.76 |
|  | 0.5 | **349.39** ± 17.94 | **246.09** ± 15.9 | **273.31** ± 18.89 | 39.83 ± 0.55 | 26.89 ± 0.06 | **50.28** ± 0.66 | 29.66 ± 0.48 | 17.95 ± 0.16 | 54.82 ± 0.22 | 11.8 ± 0.4 | 33.03 ± 0.23 | 33.33 ± 9.52 |
|  | 0.75 | 368.59 ± 5.73 | 252.62 ± 7.82 | 292.4 ± 7.11 | 44.65 ± 2.05 | 26.82 ± 0.01 | 49.91 ± 0.26 | 29.76 ± 0.27 | 18.43 ± 0.27 | 55.22 ± 0.24 | 12.0 ± 0.31 | 33.83 ± 0.35 | **66.67** ± 12.6 |
|  | 0.9 | 352.34 ± 3.1 | 249.04 ± 2.44 | 287.06 ± 4.69 | 40.15 ± 0.93 | 26.84 ± 0.04 | 48.88 ± 0.16 | 29.94 ± 0.26 | 18.12 ± 0.32 | **55.24** ± 0.02 | 11.8 ± 0.64 | 33.0 ± 0.23 | 52.38 ± 9.52 |
|  | 1.0 | 467.13 ± 14.94 | 436.87 ± 28.98 | 507.38 ± 37.85 | 40.84 ± 1.23 | 26.45 ± 0.1 | 50.28 ± 0.37 | 29.31 ± 0.12 | **19.28** ± 0.44 | 55.17 ± 0.3 | 12.13 ± 0.44 | 33.35 ± 0.09 | - |
| 2:4 | 0.0 | 96.63 ± 1.95 | 69.45 ± 0.97 | 97.33 ± 1.97 | 61.39 ± 0.43 | 28.54 ± 0.06 | 48.7 ± 0.43 | 38.43 ± 0.41 | **18.89** ± 0.57 | 59.48 ± 0.13 | 12.47 ± 0.13 | 38.27 ± 0.03 | 71.43 ± 0.0 |
|  | 0.1 | 96.35 ± 1.78 | 69.13 ± 0.85 | 98.94 ± 1.31 | 61.51 ± 0.04 | 28.56 ± 0.12 | 51.3 ± 0.16 | 39.13 ± 0.26 | 18.46 ± 0.5 | 59.39 ± 0.16 | 12.0 ± 0.42 | 38.62 ± 0.08 | **76.19** ± 9.52 |
|  | 0.25 | **94.01** ± 1.72 | **65.79** ± 0.34 | **93.54** ± 0.96 | 62.17 ± 0.02 | 28.56 ± 0.09 | **51.41** ± 0.28 | 39.52 ± 0.12 | 18.63 ± 0.19 | 59.07 ± 0.38 | 12.27 ± 0.07 | 38.8 ± 0.04 | **76.19** ± 9.52 |
|  | 0.5 | 96.97 ± 2.48 | 67.35 ± 1.33 | 99.48 ± 4.12 | 60.93 ± 0.85 | 28.54 ± 0.13 | 50.3 ± 0.93 | **39.6** ± 0.62 | 18.34 ± 0.1 | **59.54** ± 0.24 | 12.87 ± 0.52 | 38.59 ± 0.31 | 71.43 ± 0.0 |
|  | 0.75 | 99.94 ± 2.36 | 68.35 ± 1.74 | 101.87 ± 2.6 | **62.2** ± 0.06 | 28.65 ± 0.12 | 50.57 ± 0.3 | 39.06 ± 0.17 | 18.03 ± 0.1 | 59.34 ± 0.36 | 13.33 ± 0.58 | 38.74 ± 0.15 | **76.19** ± 4.76 |
|  | 0.9 | 97.52 ± 2.0 | 67.32 ± 1.09 | 101.73 ± 0.39 | 61.95 ± 0.32 | **28.76** ± 0.06 | 50.59 ± 0.08 | 39.45 ± 0.13 | 18.6 ± 0.21 | 59.1 ± 0.55 | **13.73** ± 0.41 | **38.88** ± 0.04 | **76.19** ± 9.52 |
|  | 1.0 | 115.08 ± 2.36 | 81.35 ± 0.77 | 125.66 ± 3.77 | 55.77 ± 2.58 | 28.2 ± 0.03 | 49.8 ± 0.77 | 37.43 ± 0.14 | 18.77 ± 0.37 | 58.29 ± 0.1 | 13.73 ± 0.44 | 37.43 ± 0.38 | - |

The multi-objective formulation remains consistently beneficial under Hessian recomputation. *MOON-SHOT*-SparseGPT++ outperforms *MOONSHOT*-SparseGPT (see Table 15), supporting that *MOONSHOT* is complementary to other techniques for improving pruning algorithms. *MOONSHOT*-Wanda++ does not always outperform *MOONSHOT*-Wanda (see Table 16), which may be due to Wanda's stronger approximations (e.g., diagonal Hessian approximation), leading to poorer solutions and less reliable outcomes.

## A.8 Pruning all the layers of Llama with *MOONSHOT*

In the paper, *MOONSHOT* is applied only to attention layers for efficiency reasons. In this section, we investigate the impact of applying *MOONSHOT* to all the layers of Llama-3.2-1B and Llama-3.2-3B both in terms of computation time and performance gains. As mentioned in Section 3, $K_p$ (see Algorithm 2) needs to be reduced in order to prune the larger projection layers. In particular, we use the following values of $K_p$:

Table 7: Number $K_p$ of rows (Algorithm 2) pruned at the same time for *MOONSHOT*-SparseGPT ($\lambda \neq 1$).

| Layer | Llama-3.2-1B | | Llama-3.2-3B | |
|---|---|---|---|---|
|  | $K_p$ | $n_{\text{blocks}}$ | $K_p$ | $n_{\text{blocks}}$ |
| q_proj | 2048 | 2048 | 1536 | 3072 |
| k_proj | 512 | 512 | 512 | 512 |
| v_proj | 512 | 512 | 512 | 512 |
| o_proj | 2048 | 2048 | 1536 | 3072 |
| gate_proj | 1024 | 8192 | 1024 | 8192 |
| up_proj | 1024 | 8192 | 1024 | 8192 |
| down_proj | 64 | 2048 | 192 | 3072 |

Llama-3.2-1B is pruned using a single L40 (40GB) and Llama-3.2-3B using a single A100 (80GB). Empirically, applying *MOONSHOT* to projection layers too (gate_proj, up_proj, down_proj) can yield additional gains:

Table 8: Impact of *MOONSHOT* on SparseGPT/Wanda on the LlaMA-3.2 models at 60% unstructured sparsity. The perplexities on C4, WikiText2 and PTB, as well as the zero-shot accuracies are averaged over 3 seeds with standard errors. Mean performance and win rate are computed over the 7 zero-shot downstream classification tasks.

(a) Llama-3.2-1B

| Sparsity | Method | C4 ↓ | WikiText2 ↓ | PTB ↓ | BoolQ ↑ | HellaSwag ↑ | WinoGrande ↑ | ARC-e ↑ | ARC-c ↑ | PIQA ↑ | OBQA ↑ | Mean ↑ | Win Rate ↑ |
|---|---|---|---|---|---|---|---|---|---|---|---|---|---|
| 0.6 | SparseGPT | 63.63 ± 1.18 | 54.60 ± 1.00 | 81.11 ± 3.99 | 60.67 ± 0.59 | 32.16 ± 0.20 | **54.46 ± 0.53** | 44.94 ± 0.11 | 21.47 ± 0.48 | 62.21 ± 0.20 | **17.07 ± 0.41** | 41.85 ± 0.20 | - |
|  | *MOONSHOT*-SparseGPT | 50.28 ± 1.99 | 39.13 ± 1.54 | 60.14 ± 2.90 | **62.36 ± 0.12** | 32.49 ± 0.13 | 53.09 ± 0.18 | 46.49 ± 0.38 | 21.30 ± 0.24 | 63.22 ± 0.17 | 15.73 ± 0.55 | 42.10 ± 0.11 | 57.14 ± 8.25 |
|  | *MOONSHOT* (all)-SparseGPT | **42.96 ± 0.27** | **34.29 ± 0.46** | **54.92 ± 1.22** | 62.09 ± 0.13 | **32.88 ± 0.08** | 53.14 ± 0.78 | **46.87 ± 0.14** | **21.50 ± 0.57** | **63.73 ± 0.43** | 16.93 ± 0.44 | **42.45 ± 0.22** | **71.43 ± 8.25** |
| 0.6 | Wanda | 117.71 ± 0.87 | 84.73 ± 0.73 | 119.64 ± 1.00 | 58.96 ± 1.39 | 28.86 ± 0.03 | 51.35 ± 0.09 | 38.82 ± 0.32 | 18.94 ± 0.26 | 59.05 ± 0.18 | **13.93 ± 0.24** | 38.56 ± 0.12 | - |
|  | *MOONSHOT*-Wanda | 86.55 ± 1.67 | 63.57 ± 1.61 | 98.44 ± 3.98 | 61.56 ± 0.29 | 29.53 ± 0.06 | 51.64 ± 0.23 | 40.40 ± 0.17 | **19.60 ± 0.06** | 61.12 ± 0.08 | 13.40 ± 0.20 | 39.61 ± 0.07 | 80.95 ± 4.76 |
|  | *MOONSHOT* (all)-Wanda | **85.66 ± 0.92** | **63.12 ± 0.29** | **94.39 ± 1.78** | **61.81 ± 0.18** | **29.62 ± 0.05** | **51.88 ± 0.38** | **41.27 ± 0.22** | 19.11 ± 0.13 | **61.19 ± 0.17** | 12.73 ± 0.47 | **39.66 ± 0.06** | **85.71 ± 8.25** |

(b) Llama-3.2-3B

| Sparsity | Method | C4 ↓ | WikiText2 ↓ | PTB ↓ | BoolQ ↑ | HellaSwag ↑ | WinoGrande ↑ | ARC-e ↑ | ARC-c ↑ | PIQA ↑ | OBQA ↑ | Mean ↑ | Win Rate ↑ |
|---|---|---|---|---|---|---|---|---|---|---|---|---|---|
| 0.6 | SparseGPT | 33.63 ± 0.14 | 26.12 ± 0.23 | 42.69 ± 0.73 | 66.82 ± 0.60 | 38.14 ± 0.14 | 60.91 ± 0.65 | 53.89 ± 0.11 | 26.28 ± 0.18 | 67.75 ± 0.34 | 18.47 ± 0.44 | 47.47 ± 0.08 | - |
|  | *MOONSHOT*-SparseGPT | 28.23 ± 0.11 | 22.46 ± 0.17 | 35.63 ± 0.68 | **67.76 ± 0.35** | 39.13 ± 0.07 | 61.01 ± 0.23 | 57.59 ± 0.92 | **27.79 ± 0.71** | 69.44 ± 0.13 | **20.00 ± 0.53** | **48.96 ± 0.12** | **95.24 ± 4.76** |
|  | *MOONSHOT* (all)-SparseGPT | **26.26 ± 0.12** | **20.67 ± 0.16** | **33.01 ± 0.86** | 65.66 ± 1.76 | **39.60 ± 0.06** | **61.19 ± 0.43** | **58.25 ± 0.79** | 27.45 ± 1.13 | **69.57 ± 0.07** | **20.00 ± 0.31** | 48.82 ± 0.38 | 80.95 ± 9.52 |
| 0.6 | Wanda | 41.98 ± 0.40 | 30.56 ± 0.32 | 51.00 ± 0.45 | **64.82 ± 0.35** | 35.12 ± 0.07 | **56.56 ± 0.46** | 50.58 ± 0.41 | 23.83 ± 0.12 | 65.58 ± 0.19 | 16.93 ± 0.07 | **44.77 ± 0.10** | - |
|  | *MOONSHOT*-Wanda | **37.73 ± 0.19** | **27.71 ± 0.26** | **46.47 ± 0.08** | 61.33 ± 0.91 | 35.53 ± 0.08 | 54.83 ± 0.14 | **52.53 ± 0.29** | 24.69 ± 0.21 | 66.81 ± 0.03 | 16.60 ± 0.23 | 44.62 ± 0.12 | 61.90 ± 4.76 |
|  | *MOONSHOT* (all)-Wanda | 37.83 ± 0.04 | 27.84 ± 0.18 | 47.37 ± 0.57 | 60.40 ± 0.51 | **35.54 ± 0.07** | 56.30 ± 0.21 | 52.24 ± 0.17 | **24.72 ± 0.16** | **66.85 ± 0.35** | **17.20 ± 0.23** | 44.75 ± 0.14 | **76.19 ± 4.76** |

With the exception of Wanda on Llama-3.2-3B, we observe consistent additional improvements when extending *MOONSHOT* to projection layers, indicating that the multi-objective formulation is beneficial beyond the attention blocks. However, pruning time increases by 8× for Wanda and 12× for SparseGPT on Llama-3.2-1B. For Llama-3.2-3B, pruning time increases by 8× for Wanda and 6× for SparseGPT. This increase is due to the smaller feasible $K_p$ and the increased computations with the larger Hessian.

Pruning projection layers with *MOONSHOT* is thus a practical compute/memory trade-off: depending on available resources, users may choose to apply *MOONSHOT* only to attention layers for efficiency, or extend it to projection layers for additional performance gains. Since pruning is typically performed once as an offline step, the extra runtime can be justified when resources permit, given the consistent improvements we observe.

## A.9 Pruning Llama-3.2 Instruct Models

We provide in this section additional results for the Instruct version of Llama-3.2-1B and Llama-3.2-3B.

Table 9: Test perplexity on C4, WikiText2 and PTB and zero-shot accuracy of Llama-3.2-1B-Instruct using *MOONSHOT*-SparseGPT. Zero-shot mean performance and win rate are computed over the 7 classification tasks. Results are averaged over 3 seeds with standard errors.

| Sparsity | Method | C4 ↓ | WikiText2 ↓ | PTB ↓ | BoolQ ↑ | HellaSwag ↑ | WinoGrande ↑ | ARC-e ↑ | ARC-c ↑ | PIQA ↑ | OBQA ↑ | Mean ↑ | Win Rate ↑ |
|---|---|---|---|---|---|---|---|---|---|---|---|---|---|
| Dense | - | 21.31 | 13.16 | 25.69 | 69.36 | 45.11 | 59.67 | 68.31 | 35.75 | 74.16 | 24.80 | 53.88 | - |
| 0.5 | 0.0 | 35.63 ± 0.15 | 26.56 ± 0.26 | 47.38 ± 0.53 | 62.87 ± 0.32 | 36.55 ± 0.18 | 54.33 ± 0.37 | 55.99 ± 0.12 | 25.91 ± 0.47 | 67.16 ± 0.25 | 20.33 ± 0.27 | 46.16 ± 0.09 | 0.0 ± 0.0 |
|  | 0.1 | 31.72 ± 0.06 | 22.93 ± 0.07 | 42.37 ± 0.55 | 63.22 ± 0.18 | 38.0 ± 0.05 | 54.46 ± 0.43 | 58.57 ± 0.16 | 27.9 ± 0.44 | 68.82 ± 0.17 | 20.8 ± 0.53 | 47.4 ± 0.08 | 38.1 ± 4.76 |
|  | 0.25 | 31.3 ± 0.07 | 22.52 ± 0.05 | 41.28 ± 0.48 | 63.19 ± 0.12 | 38.23 ± 0.09 | 54.91 ± 0.03 | 58.52 ± 0.33 | 28.04 ± 0.24 | 68.7 ± 0.21 | 20.2 ± 0.4 | 47.4 ± 0.08 | 42.86 ± 0.0 |
|  | 0.5 | 31.17 ± 0.04 | 22.43 ± 0.1 | 41.27 ± 0.61 | 63.2 ± 0.08 | 38.28 ± 0.03 | 54.38 ± 0.73 | 58.61 ± 0.15 | **28.73 ± 0.21** | **69.04 ± 0.14** | 21.2 ± 0.35 | 47.64 ± 0.07 | 38.1 ± 4.76 |
|  | 0.75 | 31.01 ± 0.03 | **22.26 ± 0.06** | 40.84 ± 0.78 | 63.82 ± 0.21 | 38.37 ± 0.07 | 54.78 ± 0.51 | 58.95 ± 0.18 | 27.7 ± 0.12 | 68.48 ± 0.13 | 20.8 ± 0.46 | 47.56 ± 0.07 | 47.62 ± 4.76 |
|  | 0.9 | **30.97 ± 0.03** | **22.26 ± 0.09** | **40.2 ± 0.02** | 63.74 ± 0.15 | **38.45 ± 0.04** | 55.12 ± 0.78 | **59.15 ± 0.11** | 28.58 ± 0.52 | 68.97 ± 0.11 | 20.93 ± 0.27 | **47.85 ± 0.15** | **61.9 ± 12.6** |
|  | 1.0 | 33.94 ± 0.18 | 24.56 ± 0.15 | 44.76 ± 0.34 | **63.98 ± 0.35** | 38.39 ± 0.06 | **55.93 ± 0.13** | 57.48 ± 0.14 | 27.53 ± 0.48 | 68.35 ± 0.43 | **22.27 ± 0.13** | 47.7 ± 0.07 | - |
| 0.6 | 0.0 | 73.52 ± 0.28 | 63.14 ± 0.29 | 97.29 ± 0.62 | 62.21 ± 0.15 | 30.95 ± 0.17 | 51.99 ± 0.66 | 43.2 ± 0.68 | 21.05 ± 0.3 | 60.75 ± 0.04 | 15.47 ± 0.55 | 40.8 ± 0.14 | 14.29 ± 8.25 |
|  | 0.1 | 54.01 ± 0.28 | 43.44 ± 0.9 | 69.89 ± 1.4 | 62.27 ± 0.06 | 32.69 ± 0.07 | 51.75 ± 0.5 | 47.95 ± 0.69 | **23.04 ± 0.49** | 62.88 ± 0.32 | **17.33 ± 0.41** | 42.56 ± 0.13 | 42.86 ± 8.25 |
|  | 0.25 | 52.92 ± 0.29 | 42.57 ± 0.84 | 70.33 ± 1.27 | 62.4 ± 0.08 | 32.97 ± 0.06 | 51.64 ± 0.43 | 48.74 ± 0.67 | 22.53 ± 0.26 | 63.13 ± 0.33 | 16.47 ± 0.75 | 42.55 ± 0.15 | 47.62 ± 4.76 |
|  | 0.5 | 52.01 ± 0.14 | 41.49 ± 0.78 | 67.8 ± 1.32 | 62.26 ± 0.05 | 33.06 ± 0.05 | **52.72 ± 0.16** | 49.06 ± 0.83 | 22.87 ± 0.2 | 63.62 ± 0.1 | 16.27 ± 0.57 | 42.84 ± 0.13 | 57.14 ± 0.0 |
|  | 0.75 | 51.13 ± 0.18 | 40.71 ± 0.74 | 66.39 ± 0.56 | 62.35 ± 0.15 | 33.11 ± 0.19 | 52.57 ± 0.25 | 49.45 ± 0.79 | 22.84 ± 0.27 | 63.76 ± 0.27 | 17.07 ± 0.41 | 43.02 ± 0.3 | **76.19 ± 12.6** |
|  | 0.9 | **50.87 ± 0.22** | **40.47 ± 0.43** | **66.03 ± 0.58** | 62.46 ± 0.13 | **33.29 ± 0.11** | 52.07 ± 0.03 | **49.54 ± 0.8** | **23.04 ± 0.34** | **64.15 ± 0.3** | 16.67 ± 0.07 | **43.03 ± 0.22** | 71.43 ± 16.5 |
|  | 1.0 | 61.34 ± 0.64 | 51.01 ± 1.18 | 77.49 ± 1.38 | **62.57 ± 0.05** | 32.81 ± 0.09 | 51.96 ± 0.44 | 47.98 ± 0.53 | 22.75 ± 0.23 | 63.22 ± 0.65 | 16.27 ± 0.41 | 42.51 ± 0.06 | - |
| 0.7 | 0.0 | 339.24 ± 24.3 | 347.52 ± 25.13 | 561.79 ± 42.06 | 54.19 ± 1.67 | 27.34 ± 0.12 | 49.41 ± 0.85 | 31.72 ± 0.27 | 18.52 ± 0.47 | 55.42 ± 0.54 | 13.0 ± 0.42 | 35.66 ± 0.29 | 4.76 ± 4.76 |
|  | 0.1 | 169.63 ± 5.41 | 151.44 ± 4.14 | 263.45 ± 20.91 | 61.02 ± 0.5 | 28.14 ± 0.16 | **51.93 ± 1.22** | 36.43 ± 0.36 | **19.45 ± 0.2** | 57.54 ± 0.51 | 14.47 ± 0.93 | 38.43 ± 0.27 | 80.95 ± 4.76 |
|  | 0.25 | 158.23 ± 2.87 | **147.81 ± 4.46** | 247.05 ± 8.59 | **61.97 ± 0.11** | 28.27 ± 0.12 | 50.96 ± 0.37 | 36.25 ± 0.44 | 19.11 ± 0.3 | 57.47 ± 0.21 | 14.27 ± 0.41 | 38.33 ± 0.2 | 80.95 ± 12.6 |
|  | 0.5 | 155.75 ± 2.38 | 154.07 ± 8.74 | 245.63 ± 12.03 | 61.62 ± 0.11 | 28.4 ± 0.18 | 51.33 ± 0.54 | 36.53 ± 0.65 | 19.23 ± 0.32 | 57.94 ± 0.11 | 14.93 ± 0.29 | 38.57 ± 0.13 | 80.95 ± 12.6 |
|  | 0.75 | 154.39 ± 2.31 | 156.53 ± 7.65 | 249.79 ± 14.84 | 61.35 ± 0.11 | 28.4 ± 0.08 | 51.7 ± 0.55 | 37.37 ± 1.03 | 19.31 ± 0.12 | **58.31 ± 0.24** | **15.07 ± 0.37** | **38.79 ± 0.13** | 85.71 ± 8.25 |
|  | 0.9 | **152.47 ± 1.6** | 159.61 ± 7.77 | **235.26 ± 9.45** | 61.53 ± 0.04 | **28.49 ± 0.03** | 51.38 ± 0.24 | **37.85 ± 0.66** | 19.08 ± 0.16 | 58.23 ± 0.31 | 14.2 ± 0.12 | 38.68 ± 0.15 | **90.48 ± 9.52** |
|  | 1.0 | 220.95 ± 7.06 | 278.28 ± 18.56 | 393.47 ± 37.16 | 60.43 ± 0.48 | 27.92 ± 0.1 | 50.51 ± 0.83 | 33.26 ± 0.48 | 19.11 ± 0.26 | 56.67 ± 0.07 | 13.67 ± 0.18 | 37.37 ± 0.21 | - |

Table 10: Test perplexity on C4, WikiText2 and PTB and zero-shot accuracy of Llama-3.2-1B-Instruct using *MOONSHOT*-Wanda. Zero-shot mean performance and win rate are computed over the 7 classification tasks. Results are averaged over 3 seeds with standard errors.

| Sparsity | Method | C4 ↓ | WikiText2 ↓ | PTB ↓ | BoolQ ↑ | HellaSwag ↑ | WinoGrande ↑ | ARC-e ↑ | ARC-c ↑ | PIQA ↑ | OBQA ↑ | Mean ↑ | Win Rate ↑ |
|---|---|---|---|---|---|---|---|---|---|---|---|---|---|
| Dense | - | 21.31 | 13.16 | 25.69 | 69.36 | 45.11 | 59.67 | 68.31 | 35.75 | 74.16 | 24.80 | 53.88 | - |
| 0.5 | 0.0 | 41.06 ± 0.24 | 29.22 ± 0.09 | 53.47 ± 0.42 | 62.19 ± 0.11 | 35.7 ± 0.01 | 53.67 ± 0.25 | 56.87 ± 0.43 | 26.14 ± 0.22 | 66.52 ± 0.16 | 16.8 ± 0.23 | 45.41 ± 0.1 | 42.86 ± 0.0 |
|  | 0.1 | 40.41 ± 0.11 | 28.61 ± 0.06 | 52.63 ± 0.41 | 62.85 ± 0.37 | 35.92 ± 0.05 | 53.64 ± 0.27 | 56.69 ± 0.42 | 26.11 ± 0.44 | 66.56 ± 0.1 | 16.87 ± 0.29 | 45.52 ± 0.07 | 61.9 ± 12.6 |
|  | 0.25 | 39.73 ± 0.1 | **27.87 ± 0.09** | 51.39 ± 0.4 | 62.61 ± 0.31 | 35.99 ± 0.06 | 54.2 ± 0.38 | 57.1 ± 0.28 | 26.22 ± 0.32 | 66.65 ± 0.16 | 17.0 ± 0.31 | 45.68 ± 0.06 | 52.38 ± 9.52 |
|  | 0.5 | **39.58 ± 0.28** | **27.87 ± 0.19** | **51.26 ± 0.5** | 62.75 ± 0.23 | 36.12 ± 0.01 | 53.96 ± 0.09 | **57.25 ± 0.33** | 26.02 ± 0.52 | 66.72 ± 0.16 | 17.07 ± 0.33 | 45.7 ± 0.1 | 61.9 ± 9.52 |
|  | 0.75 | 39.91 ± 0.27 | 28.13 ± 0.1 | 51.4 ± 0.12 | 62.95 ± 0.59 | 36.22 ± 0.04 | 54.06 ± 0.68 | 56.9 ± 0.15 | **26.39 ± 0.35** | **67.05 ± 0.3** | 17.27 ± 0.24 | 45.83 ± 0.22 | **66.67 ± 17.17** |
|  | 0.9 | 40.16 ± 0.17 | 28.28 ± 0.03 | 52.16 ± 0.18 | **63.06 ± 0.27** | **36.34 ± 0.0** | 54.72 ± 0.48 | 56.96 ± 0.05 | 26.28 ± 0.2 | 66.74 ± 0.13 | 17.87 ± 0.07 | **46.0 ± 0.1** | **66.67 ± 12.6** |
|  | 1.0 | 46.5 ± 0.11 | 33.63 ± 0.04 | 59.43 ± 0.11 | 62.62 ± 0.18 | 35.77 ± 0.08 | **55.99 ± 0.52** | 54.05 ± 0.24 | 25.97 ± 0.34 | 65.89 ± 0.06 | **18.07 ± 0.44** | 45.48 ± 0.2 | - |
| 0.6 | 0.0 | 106.89 ± 2.02 | 85.13 ± 2.17 | 115.83 ± 3.22 | 62.2 ± 0.02 | 29.38 ± 0.1 | 51.54 ± 0.36 | 41.41 ± 0.37 | 18.66 ± 0.06 | 59.47 ± 0.03 | 14.0 ± 0.2 | 39.52 ± 0.01 | 47.62 ± 4.76 |
|  | 0.1 | 100.94 ± 1.72 | 78.78 ± 1.76 | 109.13 ± 2.96 | 62.15 ± 0.02 | 29.53 ± 0.12 | 52.14 ± 0.46 | 41.46 ± 0.37 | 19.25 ± 0.23 | 60.08 ± 0.14 | 14.6 ± 0.46 | 39.89 ± 0.03 | 66.67 ± 4.76 |
|  | 0.25 | 96.71 ± 1.01 | 74.61 ± 1.17 | **103.93 ± 2.76** | 62.15 ± 0.01 | 29.71 ± 0.06 | 52.54 ± 0.59 | 42.19 ± 0.46 | 19.48 ± 0.27 | 60.37 ± 0.34 | 14.8 ± 0.12 | 40.18 ± 0.23 | 76.19 ± 12.6 |
|  | 0.5 | **96.18 ± 1.15** | **72.81 ± 1.32** | 104.85 ± 2.75 | 62.27 ± 0.01 | 29.91 ± 0.09 | 52.83 ± 0.37 | 42.68 ± 0.33 | 19.8 ± 0.26 | 60.45 ± 0.17 | **15.2 ± 0.23** | 40.45 ± 0.13 | **90.48 ± 4.76** |
|  | 0.75 | 97.72 ± 0.56 | 74.27 ± 1.04 | 106.11 ± 2.83 | 62.2 ± 0.02 | 30.21 ± 0.1 | **53.01 ± 0.32** | 43.17 ± 0.23 | 19.97 ± 0.34 | 60.43 ± 0.27 | 14.6 ± 0.23 | 40.51 ± 0.1 | 76.19 ± 9.52 |
|  | 0.9 | 99.91 ± 1.39 | 77.19 ± 1.06 | 105.98 ± 1.66 | **62.31 ± 0.06** | **30.29 ± 0.07** | 52.2 ± 0.38 | **43.74 ± 0.21** | **20.42 ± 0.21** | **60.57 ± 0.07** | 14.2 ± 0.12 | **40.53 ± 0.06** | 76.19 ± 4.76 |
|  | 1.0 | 154.55 ± 0.94 | 129.18 ± 1.52 | 141.38 ± 1.56 | 61.73 ± 0.09 | 29.62 ± 0.02 | 52.28 ± 0.29 | 39.69 ± 0.29 | 19.8 ± 0.64 | 59.16 ± 0.26 | 15.0 ± 0.12 | 39.61 ± 0.21 | - |
| 0.7 | 0.0 | 455.59 ± 8.85 | 467.43 ± 20.46 | 477.51 ± 15.39 | 49.42 ± 3.01 | 26.75 ± 0.08 | 49.33 ± 0.2 | 30.09 ± 0.33 | **18.34 ± 0.27** | 55.02 ± 0.1 | 12.27 ± 0.27 | 34.46 ± 0.45 | 71.43 ± 8.25 |
|  | 0.1 | 444.16 ± 9.19 | 437.37 ± 24.56 | 450.97 ± 22.14 | 53.84 ± 2.97 | 26.93 ± 0.02 | 50.51 ± 0.3 | 30.22 ± 0.26 | 18.0 ± 0.21 | 55.24 ± 0.11 | 12.0 ± 0.5 | 35.25 ± 0.34 | 71.43 ± 0.0 |
|  | 0.25 | **438.43 ± 21.47** | 416.16 ± 37.44 | 428.54 ± 38.04 | 55.31 ± 1.87 | 27.02 ± 0.03 | 50.91 ± 0.6 | 30.65 ± 0.14 | 18.15 ± 0.14 | 55.55 ± 0.24 | 12.4 ± 0.12 | 35.71 ± 0.18 | 80.95 ± 9.52 |
|  | 0.5 | 454.78 ± 12.84 | **413.61 ± 22.88** | **425.15 ± 18.8** | 58.4 ± 0.18 | 27.09 ± 0.04 | 51.04 ± 0.74 | 30.64 ± 0.11 | 17.92 ± 0.34 | 55.57 ± 0.24 | 12.93 ± 0.18 | 36.23 ± 0.08 | 85.71 ± 8.25 |
|  | 0.75 | 481.74 ± 11.64 | 466.41 ± 26.96 | 426.24 ± 10.96 | 57.66 ± 1.42 | 27.0 ± 0.05 | 52.67 ± 0.59 | **31.02 ± 0.18** | 18.2 ± 0.11 | 55.77 ± 0.14 | 12.67 ± 0.33 | **36.43 ± 0.24** | **90.48 ± 9.52** |
|  | 0.9 | 583.01 ± 11.37 | 559.07 ± 16.93 | 513.8 ± 22.06 | 56.83 ± 0.84 | **27.15 ± 0.04** | 51.91 ± 0.25 | 31.0 ± 0.13 | 17.72 ± 0.11 | **55.89 ± 0.52** | 12.73 ± 0.33 | 36.18 ± 0.08 | 76.19 ± 4.76 |
|  | 1.0 | 1797.15 ± 176.66 | 2120.64 ± 117.11 | 2225.24 ± 246.18 | 52.1 ± 3.48 | 26.62 ± 0.06 | 49.25 ± 0.33 | 28.3 ± 0.2 | 18.26 ± 0.2 | 54.39 ± 0.15 | **13.0 ± 0.23** | 34.56 ± 0.45 | - |

Table 11: Test perplexity on C4, WikiText2 and PTB and zero-shot accuracy of Llama-3.2-3B-Instruct using *MOONSHOT*-SparseGPT. Zero-shot mean performance and win rate are computed over the 7 classification tasks. Results are averaged over 3 seeds with standard errors.

| Sparsity | Method | C4 ↓ | WikiText2 ↓ | PTB ↓ | BoolQ ↑ | HellaSwag ↑ | WinoGrande ↑ | ARC-e ↑ | ARC-c ↑ | PIQA ↑ | OBQA ↑ | Mean ↑ | Win Rate ↑ |
|---|---|---|---|---|---|---|---|---|---|---|---|---|---|
| Dense | - | 16.49 | 11.04 | 20.42 | 78.47 | 52.24 | 67.40 | 73.99 | 43.43 | 75.79 | 27.40 | 59.82 | - |
| 0.5 | 0.0 | 24.55 ± 0.14 | 19.22 ± 0.44 | 33.43 ± 1.03 | 74.55 ± 0.01 | 43.43 ± 0.16 | 61.64 ± 0.25 | 64.37 ± 0.31 | 33.25 ± 0.6 | 70.51 ± 0.22 | 21.73 ± 0.48 | 52.78 ± 0.18 | 4.76 ± 4.76 |
|  | 0.1 | 22.61 ± 0.02 | 17.45 ± 0.28 | 29.93 ± 0.2 | 75.52 ± 0.32 | 44.79 ± 0.1 | 62.88 ± 0.37 | 65.05 ± 0.43 | 34.3 ± 0.34 | **71.85 ± 0.1** | 22.0 ± 0.2 | 53.82 ± 0.11 | 38.1 ± 4.76 |
|  | 0.25 | 22.45 ± 0.01 | 17.21 ± 0.14 | 29.67 ± 0.09 | 75.77 ± 0.4 | 44.81 ± 0.02 | 63.27 ± 0.31 | 65.5 ± 0.56 | 34.3 ± 0.44 | 71.8 ± 0.32 | 22.13 ± 0.29 | 53.94 ± 0.2 | 38.1 ± 12.6 |
|  | 0.5 | 22.41 ± 0.03 | 17.12 ± 0.18 | 29.42 ± 0.03 | **76.13 ± 0.04** | 44.87 ± 0.05 | 63.04 ± 0.52 | 65.74 ± 0.53 | **34.33 ± 0.37** | 71.84 ± 0.16 | 21.93 ± 0.64 | 53.98 ± 0.16 | **42.86 ± 14.29** |
|  | 0.75 | 22.38 ± 0.04 | 17.09 ± 0.17 | **29.21 ± 0.24** | 75.95 ± 0.15 | 44.88 ± 0.02 | 62.93 ± 0.21 | 66.11 ± 0.09 | 34.07 ± 0.31 | 71.47 ± 0.15 | 22.13 ± 0.59 | 53.94 ± 0.14 | 38.1 ± 4.76 |
|  | 0.9 | **22.37 ± 0.02** | **17.04 ± 0.14** | 29.24 ± 0.27 | 75.81 ± 0.13 | 44.89 ± 0.01 | 63.35 ± 0.13 | **66.16 ± 0.06** | **34.33 ± 0.14** | 71.65 ± 0.05 | 22.33 ± 0.24 | 54.08 ± 0.02 | **42.86 ± 8.25** |
|  | 1.0 | 23.11 ± 0.05 | 17.73 ± 0.26 | 30.01 ± 0.16 | 76.06 ± 0.34 | **45.08 ± 0.1** | **64.59 ± 0.35** | 64.52 ± 0.21 | 34.22 ± 0.6 | 71.16 ± 0.27 | **23.33 ± 0.44** | **54.14 ± 0.2** | - |
| 0.6 | 0.0 | 52.02 ± 0.56 | 49.18 ± 2.2 | 79.4 ± 5.34 | 67.55 ± 0.77 | 34.8 ± 0.18 | 56.41 ± 0.37 | 49.13 ± 0.8 | 24.06 ± 0.13 | 64.78 ± 0.79 | 16.73 ± 0.44 | 44.78 ± 0.15 | 9.52 ± 4.76 |
|  | 0.1 | 38.86 ± 0.36 | 35.07 ± 0.68 | 56.0 ± 1.4 | 70.11 ± 1.18 | 36.57 ± 0.06 | 59.54 ± 0.85 | 54.11 ± 0.57 | 26.68 ± 0.73 | 65.81 ± 0.21 | 17.8 ± 0.64 | 47.23 ± 0.34 | 61.9 ± 4.76 |
|  | 0.25 | 37.96 ± 0.15 | 34.29 ± 0.7 | 54.26 ± 1.11 | 70.08 ± 0.75 | 36.84 ± 0.24 | 58.83 ± 0.71 | 55.3 ± 0.51 | 27.13 ± 0.31 | **66.43 ± 0.39** | **18.07 ± 0.75** | 47.53 ± 0.39 | 76.19 ± 12.6 |
|  | 0.5 | 37.92 ± 0.41 | 33.55 ± 0.67 | 53.92 ± 1.47 | 69.87 ± 0.84 | **36.9 ± 0.1** | **59.93 ± 0.43** | 55.23 ± 0.57 | 27.36 ± 0.51 | 66.41 ± 0.31 | 17.93 ± 0.57 | 47.66 ± 0.37 | **90.48 ± 4.76** |
|  | 0.75 | 37.58 ± 0.25 | 33.69 ± 0.44 | 54.45 ± 0.97 | 70.32 ± 0.96 | **36.9 ± 0.08** | 59.69 ± 0.64 | **55.84 ± 0.58** | 27.36 ± 0.36 | 66.25 ± 0.31 | 17.47 ± 0.77 | **47.69 ± 0.31** | 76.19 ± 4.76 |
|  | 0.9 | **37.31 ± 0.18** | **33.4 ± 0.5** | **53.89 ± 1.03** | **70.37 ± 1.18** | **36.9 ± 0.12** | 59.67 ± 0.75 | 55.57 ± 0.16 | **27.9 ± 0.47** | 65.96 ± 0.4 | 17.33 ± 0.52 | 47.67 ± 0.24 | 76.19 ± 4.76 |
|  | 1.0 | 41.2 ± 0.1 | 38.64 ± 0.82 | 61.84 ± 1.86 | 69.73 ± 0.98 | 36.61 ± 0.18 | 59.3 ± 0.54 | 53.27 ± 0.16 | 26.22 ± 0.32 | 65.16 ± 0.23 | 17.27 ± 0.57 | 46.79 ± 0.34 | - |
| 0.7 | 0.0 | 249.7 ± 6.99 | 301.94 ± 8.0 | 364.31 ± 8.83 | 61.95 ± 0.09 | 27.16 ± 0.06 | 49.99 ± 0.09 | 30.89 ± 0.74 | 18.03 ± 0.12 | 56.31 ± 0.39 | 12.27 ± 0.24 | 36.66 ± 0.19 | 9.52 ± 9.52 |
|  | 0.1 | 134.51 ± 3.39 | 142.92 ± 1.32 | 190.67 ± 16.05 | 62.59 ± 0.12 | 28.53 ± 0.15 | 51.83 ± 0.64 | 35.16 ± 0.42 | 19.11 ± 0.61 | 58.12 ± 0.05 | 12.0 ± 0.23 | 38.19 ± 0.11 | 57.14 ± 8.25 |
|  | 0.25 | 130.54 ± 1.15 | 136.69 ± 3.78 | 186.8 ± 16.49 | 62.7 ± 0.23 | 28.57 ± 0.11 | 51.51 ± 1.24 | 35.17 ± 0.39 | 19.0 ± 0.74 | 58.29 ± 0.35 | 11.87 ± 0.29 | 38.16 ± 0.22 | 57.14 ± 8.25 |
|  | 0.5 | 125.49 ± 2.34 | 131.09 ± 3.91 | 181.17 ± 15.41 | 62.59 ± 0.19 | 28.79 ± 0.11 | 52.01 ± 0.62 | 35.97 ± 0.4 | **19.2 ± 0.57** | **58.65 ± 0.23** | 12.0 ± 0.2 | **38.46 ± 0.21** | 66.67 ± 4.76 |
|  | 0.75 | **121.85 ± 1.19** | **126.32 ± 2.39** | 177.98 ± 15.26 | 62.6 ± 0.47 | 28.77 ± 0.1 | 51.83 ± 0.39 | **36.27 ± 0.41** | 18.69 ± 0.49 | 58.23 ± 0.44 | 12.33 ± 0.37 | 38.39 ± 0.27 | 61.9 ± 9.52 |
|  | 0.9 | 123.54 ± 1.3 | 131.66 ± 1.76 | **175.18 ± 11.33** | 62.63 ± 0.17 | **28.83 ± 0.05** | **52.33 ± 0.83** | 35.69 ± 0.46 | **19.2 ± 0.44** | 58.14 ± 0.2 | 12.2 ± 0.12 | 38.43 ± 0.21 | **71.43 ± 8.25** |
|  | 1.0 | 136.3 ± 0.13 | 156.05 ± 2.0 | 193.42 ± 6.92 | **62.92 ± 0.51** | 28.22 ± 0.14 | 52.12 ± 0.61 | 34.69 ± 0.09 | 18.86 ± 0.13 | 57.29 ± 0.35 | **12.73 ± 0.48** | 38.12 ± 0.18 | - |

Table 12: Test perplexity on C4, WikiText2 and PTB and zero-shot accuracy of Llama-3.2-3B-Instruct using *MOONSHOT*-Wanda. Zero-shot mean performance and win rate are computed over the 7 classification tasks. Results are averaged over 3 seeds with standard errors.

| Sparsity | Method | C4 ↓ | WikiText2 ↓ | PTB ↓ | BoolQ ↑ | HellaSwag ↑ | WinoGrande ↑ | ARC-e ↑ | ARC-c ↑ | PIQA ↑ | OBQA ↑ | Mean ↑ | Win Rate ↑ |
|---|---|---|---|---|---|---|---|---|---|---|---|---|---|
| Dense | - | 16.49 | 11.04 | 20.42 | 78.47 | 52.24 | 67.40 | 73.99 | 43.43 | 75.79 | 27.40 | 59.82 | - |
| 0.5 | 0.0 | 25.54 ± 0.11 | 19.79 ± 0.17 | 34.17 ± 0.36 | 73.04 ± 0.34 | 42.38 ± 0.08 | 60.8 ± 0.59 | 66.26 ± 0.23 | 33.73 ± 0.28 | 70.98 ± 0.05 | 21.47 ± 0.24 | 52.67 ± 0.22 | 38.1 ± 4.76 |
|  | 0.1 | 25.15 ± 0.07 | 19.26 ± 0.1 | 33.32 ± 0.19 | 73.21 ± 0.33 | 42.57 ± 0.05 | 62.04 ± 0.21 | 66.02 ± 0.09 | 33.79 ± 0.27 | 70.96 ± 0.1 | **22.33 ± 0.07** | 52.99 ± 0.12 | 42.86 ± 0.0 |
|  | 0.25 | 24.86 ± 0.07 | 19.06 ± 0.2 | 32.71 ± 0.21 | 73.79 ± 0.27 | 42.7 ± 0.05 | 61.88 ± 0.28 | 66.22 ± 0.15 | 34.19 ± 0.29 | **71.42 ± 0.02** | 21.93 ± 0.24 | 53.16 ± 0.13 | 47.62 ± 4.76 |
|  | 0.5 | 24.53 ± 0.06 | 18.73 ± 0.14 | 32.3 ± 0.13 | 73.37 ± 0.38 | 42.88 ± 0.06 | 62.77 ± 0.47 | 66.33 ± 0.15 | 34.7 ± 0.27 | 71.27 ± 0.11 | 22.07 ± 0.13 | 53.34 ± 0.07 | 57.14 ± 0.0 |
|  | 0.75 | 24.48 ± 0.13 | 18.57 ± 0.15 | 31.99 ± 0.23 | 73.86 ± 0.51 | 42.9 ± 0.07 | 62.67 ± 0.18 | 66.53 ± 0.0 | 35.01 ± 0.2 | 71.4 ± 0.13 | 21.93 ± 0.18 | 53.43 ± 0.08 | **61.9 ± 4.76** |
|  | 0.9 | **24.35 ± 0.08** | **18.46 ± 0.01** | **31.62 ± 0.17** | 73.86 ± 0.26 | 43.2 ± 0.01 | 63.59 ± 0.09 | **66.4 ± 0.1** | **35.32 ± 0.21** | 71.25 ± 0.02 | 21.6 ± 0.23 | **53.6 ± 0.04** | 52.38 ± 4.76 |
|  | 1.0 | 24.76 ± 0.01 | 18.7 ± 0.04 | 32.27 ± 0.13 | **74.28 ± 0.34** | **43.43 ± 0.06** | **63.85 ± 0.16** | 64.66 ± 0.12 | 34.5 ± 0.22 | 70.26 ± 0.16 | 21.27 ± 0.35 | 53.18 ± 0.04 | - |
| 0.6 | 0.0 | 73.41 ± 4.15 | 66.23 ± 4.51 | 101.71 ± 3.95 | 63.9 ± 0.4 | 32.57 ± 0.25 | 55.67 ± 0.43 | 49.89 ± 0.66 | 23.83 ± 0.52 | 63.64 ± 0.56 | 14.47 ± 0.24 | 43.42 ± 0.31 | 19.05 ± 12.6 |
|  | 0.1 | 67.82 ± 1.91 | 61.5 ± 2.44 | 94.64 ± 1.55 | 64.48 ± 0.19 | 33.09 ± 0.18 | 55.62 ± 0.26 | 50.74 ± 0.95 | 24.94 ± 0.38 | 64.18 ± 0.34 | 15.0 ± 0.5 | 44.01 ± 0.29 | 23.81 ± 9.52 |
|  | 0.25 | 63.87 ± 0.75 | 57.93 ± 1.4 | 90.5 ± 1.03 | 64.69 ± 0.04 | 33.55 ± 0.18 | 56.75 ± 0.21 | 51.32 ± 0.51 | 24.91 ± 0.45 | 64.24 ± 0.19 | 15.33 ± 0.07 | 44.4 ± 0.2 | 33.33 ± 4.76 |
|  | 0.5 | 60.41 ± 0.83 | 54.03 ± 1.15 | 85.55 ± 0.97 | 65.38 ± 0.13 | 33.92 ± 0.17 | 56.91 ± 0.41 | 51.02 ± 0.58 | 25.17 ± 0.32 | 64.25 ± 0.17 | 15.87 ± 0.18 | 44.65 ± 0.09 | 47.62 ± 4.76 |
|  | 0.75 | 57.64 ± 0.24 | 50.62 ± 0.77 | 82.07 ± 0.91 | 65.89 ± 0.24 | 34.25 ± 0.06 | 56.99 ± 0.16 | 51.5 ± 0.26 | **25.74 ± 0.23** | 64.33 ± 0.15 | **16.67 ± 0.27** | 45.05 ± 0.08 | 76.19 ± 9.52 |
|  | 0.9 | **56.47 ± 0.27** | **49.09 ± 0.69** | **80.34 ± 0.05** | **66.41 ± 0.4** | **34.49 ± 0.06** | **57.51 ± 0.27** | 51.46 ± 0.09 | 25.65 ± 0.19 | **64.54 ± 0.21** | 16.2 ± 0.35 | **45.18 ± 0.13** | **80.95 ± 12.6** |
|  | 1.0 | 57.74 ± 0.74 | 50.82 ± 0.81 | 82.33 ± 0.32 | 65.54 ± 0.49 | 34.33 ± 0.07 | 57.14 ± 0.24 | 48.72 ± 0.28 | 25.06 ± 0.24 | 63.98 ± 0.23 | **16.67 ± 0.07** | 44.49 ± 0.06 | - |
| 0.7 | 0.0 | 323.12 ± 19.39 | 340.56 ± 32.95 | 302.69 ± 10.61 | 39.43 ± 0.25 | 26.81 ± 0.07 | 49.46 ± 0.73 | 31.14 ± 0.19 | **18.54 ± 0.28** | 55.82 ± 0.41 | 11.67 ± 0.18 | 33.27 ± 0.18 | 42.86 ± 8.25 |
|  | 0.1 | 301.09 ± 12.36 | 311.9 ± 24.51 | 288.06 ± 3.24 | 41.1 ± 0.2 | 26.86 ± 0.1 | 49.7 ± 0.6 | 31.1 ± 0.11 | 18.15 ± 0.2 | 56.33 ± 0.16 | 12.0 ± 0.2 | 33.6 ± 0.14 | 47.62 ± 12.6 |
|  | 0.25 | 290.05 ± 12.47 | 296.65 ± 19.83 | 286.03 ± 6.45 | 45.08 ± 0.64 | 26.93 ± 0.05 | 49.83 ± 0.17 | 31.14 ± 0.29 | 17.95 ± 0.22 | 56.31 ± 0.17 | 11.4 ± 0.31 | 34.09 ± 0.15 | 38.1 ± 9.52 |
|  | 0.5 | 277.25 ± 10.52 | 279.64 ± 12.43 | 282.93 ± 6.27 | 46.76 ± 1.4 | 27.1 ± 0.05 | 49.96 ± 0.05 | 31.59 ± 0.12 | 17.61 ± 0.08 | 56.57 ± 0.11 | 11.87 ± 0.05 | 34.49 ± 0.26 | 42.86 ± 8.25 |
|  | 0.75 | 264.81 ± 3.25 | 266.54 ± 8.53 | 280.08 ± 6.69 | 47.18 ± 0.96 | 27.26 ± 0.03 | **50.3 ± 0.25** | 31.75 ± 0.11 | 18.0 ± 0.3 | 56.66 ± 0.33 | 11.73 ± 0.13 | 34.7 ± 0.21 | **52.38 ± 4.76** |
|  | 0.9 | 255.35 ± 1.85 | **258.39 ± 3.53** | 268.83 ± 4.87 | 45.51 ± 1.63 | 27.38 ± 0.04 | **50.3 ± 0.15** | **31.8 ± 0.2** | 17.78 ± 0.19 | **57.05 ± 0.3** | 11.47 ± 0.13 | 34.47 ± 0.29 | **52.38 ± 12.6** |
|  | 1.0 | **234.03 ± 3.39** | 271.85 ± 7.51 | **264.91 ± 5.09** | **55.83 ± 0.19** | **27.42 ± 0.04** | 48.8 ± 0.18 | 31.0 ± 0.18 | 17.83 ± 0.25 | 56.84 ± 0.18 | **12.2 ± 0.2** | **35.7 ± 0.07** | - |

## A.10 Additional Ablations on $\lambda$

In Section 3, we analyze the sensitivity of $\lambda$ and find that the optimum almost never occurs at the extremes $\lambda \in (0, 1)$. Across all architectures, pruning baselines, and sparsity regimes we tested, intermediate values consistently outperform the single-objective endpoints, as shown in Figure 5. This pattern supports the idea that balancing the reconstruction and Fisher terms yields a more robust pruning criterion than either alone.

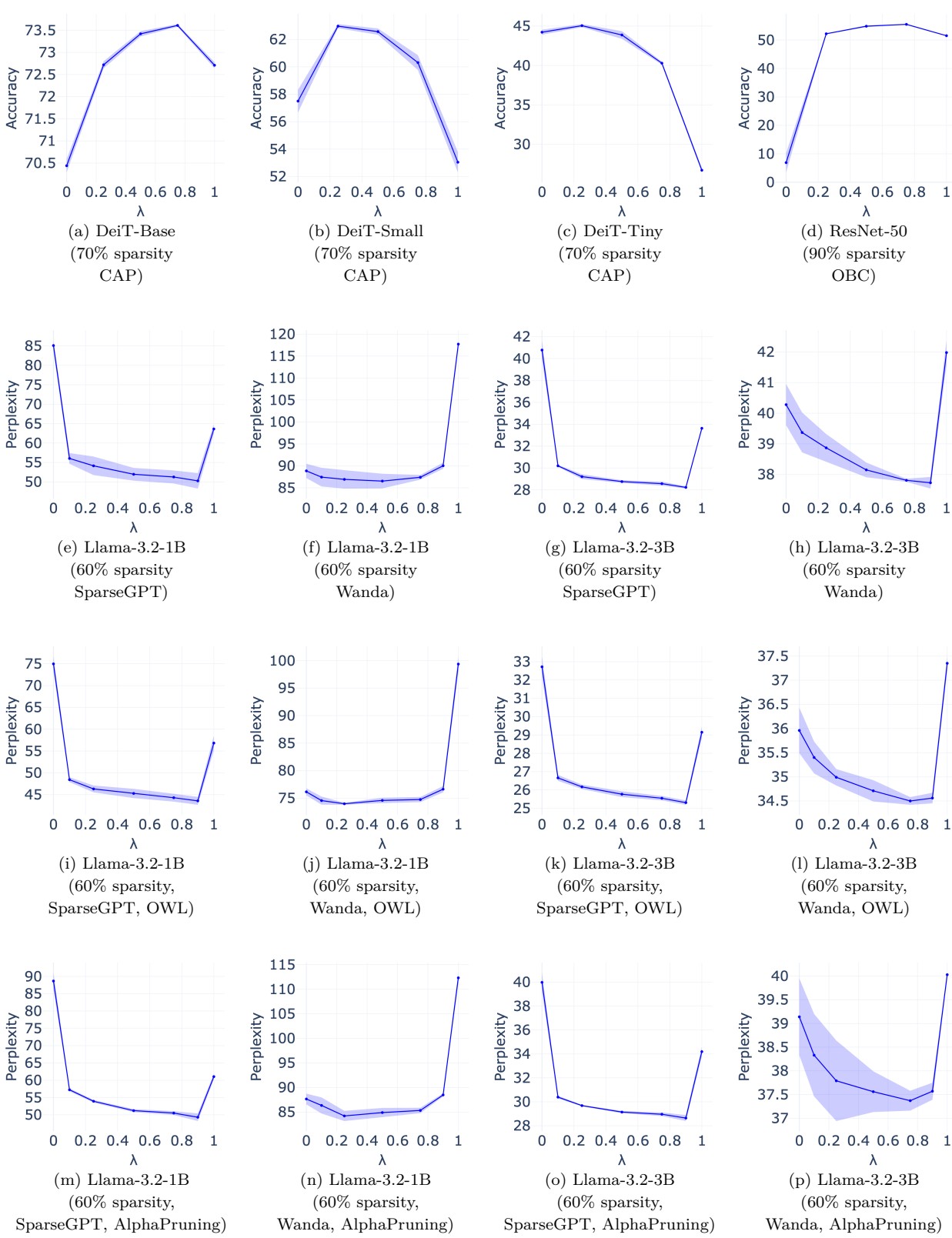

Figure 5: Performance of *MOONSHOT* across different values of $\lambda$ on the DeiT models (70% sparsity), ResNet-50 (90% sparsity), and Llama-3.2 models (60% and 2:4 sparsity), using CAP, OBC, and SparseGPT as base methods respectively. Accuracy is reported for vision models and perplexity on C4 for LLMs.

## A.11   Further Evaluation of *MOONSHOT* Across Sparsity Regimes

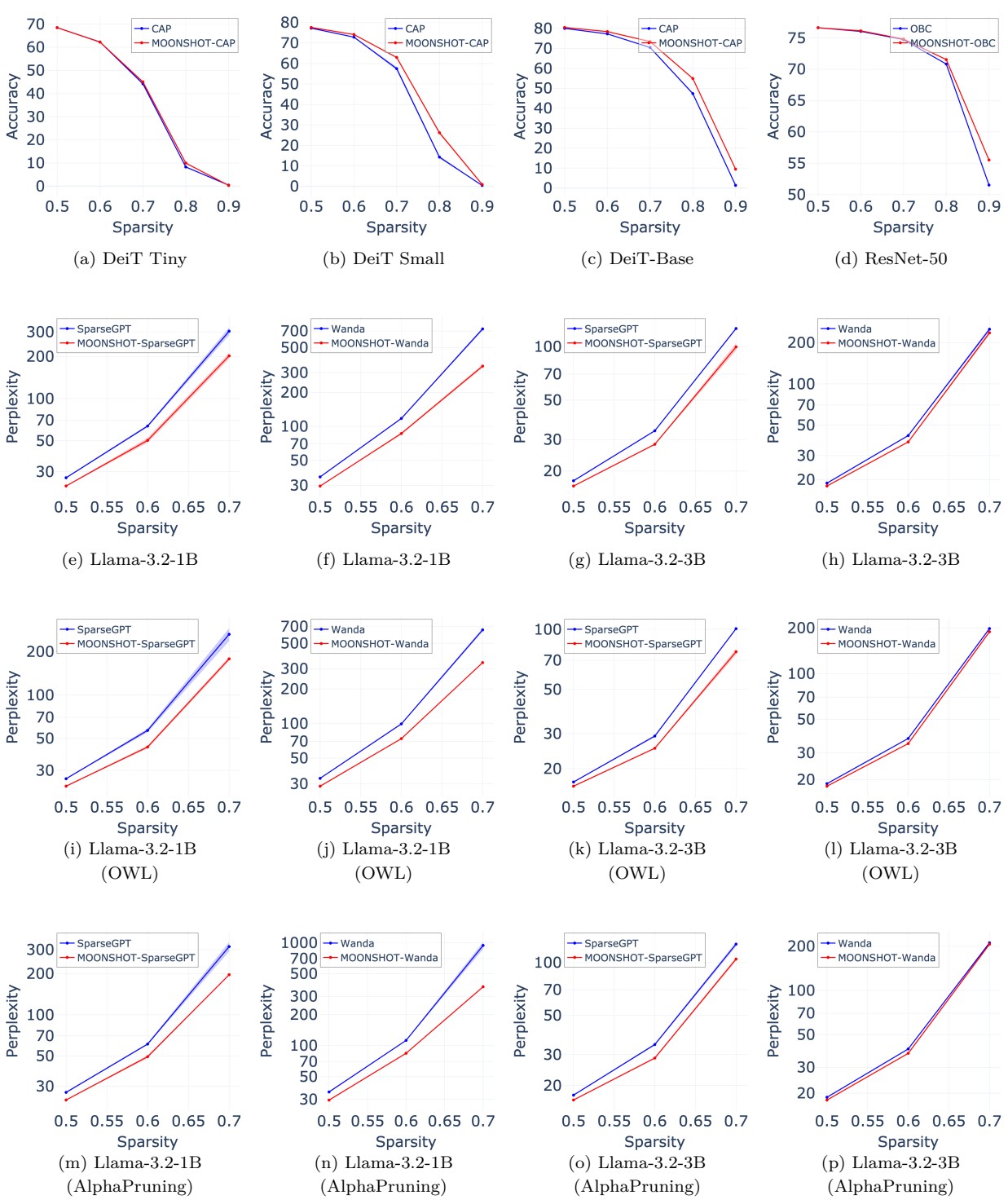

Figure 6: Impact of *MOONSHOT* across sparsity levels on CAP for the DeiT models, OBC on ResNet-50 and SparseGPT/Wanda on the Llama-3.2 models.

Figure 6 expands the sparsity sweeps for all architectures, pruning algorithms and pruning methods, and shows a consistent trend: *MOONSHOT* consistently yields better performance–sparsity tradeoff than their single-objective counterparts, with the gap widening in the high-sparsity regime where baselines degrade most. Moreover, when combined with non-uniform sparsity allocation methods (OWL and AlphaPruning), *MOONSHOT* 's gains are additive: curves shift upward at nearly all sparsity regimes, indicating that our multi-objective signal complements allocation strategies rather than replacing them.

### A.12    Comprehensive Experimental Results

Tables 2 and 3, as well as Figure 1 show the results of *MOONSHOT* for the optimal value of $\lambda$ across a selection of sparsity regimes. In this section, we provide the results of *MOONSHOT* precisely for each value of $\lambda$ that was tried, across all sparsity regimes.

Table 13: Test Accuracy for the DeiT models and ResNet-50 using *MOONSHOT*-CAP and *MOONSHOT*-OBC respectively

| Sparsity | $\lambda$ | DeiT Tiny | DeiT Small | DeiT Base | ResNet-50 |
|---|---|---|---|---|---|
| Dense | - | 72.14 | 79.83 | 81.80 | 77.11 |
| 0.5 | 0.00 | $68.49 \pm 0.1$ | $77.27 \pm 0.03$ | $80.01 \pm 0.01$ | $50.88 \pm 25.39$ |
| | 0.25 | $\mathbf{68.62} \pm 0.03$ | $77.63 \pm 0.02$ | $80.5 \pm 0.02$ | $76.56 \pm 0.03$ |
| | 0.50 | $68.35 \pm 0.05$ | $\mathbf{77.67} \pm 0.01$ | $80.56 \pm 0.02$ | $76.61 \pm 0.03$ |
| | 0.75 | $68.02 \pm 0.07$ | $77.49 \pm 0.03$ | $\mathbf{80.6} \pm 0.02$ | $\mathbf{76.63} \pm 0.04$ |
| | 1.00 | $65.49 \pm 0.02$ | $76.56 \pm 0.04$ | $80.58 \pm 0.01$ | $\mathbf{76.63} \pm 0.05$ |
| 0.6 | 0.00 | $\mathbf{62.28} \pm 0.05$ | $72.89 \pm 0.04$ | $77.27 \pm 0.1$ | $50.37 \pm 25.13$ |
| | 0.25 | $62.22 \pm 0.09$ | $\mathbf{74.16} \pm 0.04$ | $78.41 \pm 0.06$ | $76.04 \pm 0.01$ |
| | 0.50 | $61.76 \pm 0.1$ | $74.14 \pm 0.02$ | $78.62 \pm 0.04$ | $\mathbf{76.13} \pm 0.02$ |
| | 0.75 | $60.7 \pm 0.2$ | $73.76 \pm 0.09$ | $\mathbf{78.81} \pm 0.04$ | $76.11 \pm 0.0$ |
| | 1.00 | $54.18 \pm 0.15$ | $71.31 \pm 0.19$ | $78.67 \pm 0.01$ | $76.04 \pm 0.02$ |
| 0.7 | 0.00 | $44.22 \pm 0.32$ | $57.5 \pm 0.83$ | $70.44 \pm 0.15$ | $48.94 \pm 24.42$ |
| | 0.25 | $\mathbf{45.05} \pm 0.2$ | $\mathbf{62.97} \pm 0.15$ | $72.72 \pm 0.08$ | $74.7 \pm 0.04$ |
| | 0.50 | $43.87 \pm 0.49$ | $62.57 \pm 0.23$ | $73.42 \pm 0.06$ | $\mathbf{74.82} \pm 0.05$ |
| | 0.75 | $40.29 \pm 0.19$ | $60.31 \pm 0.54$ | $\mathbf{73.61} \pm 0.03$ | $\mathbf{74.82} \pm 0.04$ |
| | 1.00 | $26.71 \pm 0.27$ | $53.04 \pm 0.74$ | $72.71 \pm 0.07$ | $74.73 \pm 0.03$ |
| 0.8 | 0.00 | $8.28 \pm 0.32$ | $14.28 \pm 0.65$ | $47.32 \pm 0.48$ | $44.4 \pm 22.15$ |
| | 0.25 | $\mathbf{9.97} \pm 0.26$ | $\mathbf{26.2} \pm 0.59$ | $53.56 \pm 0.12$ | $71.02 \pm 0.08$ |
| | 0.50 | $8.58 \pm 0.26$ | $24.36 \pm 0.56$ | $54.9 \pm 0.26$ | $71.39 \pm 0.03$ |
| | 0.75 | $6.23 \pm 0.15$ | $19.27 \pm 0.88$ | $\mathbf{55.37} \pm 0.2$ | $\mathbf{71.54} \pm 0.02$ |
| | 1.00 | $2.14 \pm 0.16$ | $9.84 \pm 0.62$ | $49.2 \pm 0.18$ | $70.83 \pm 0.04$ |
| 0.9 | 0.00 | $0.43 \pm 0.06$ | $0.43 \pm 0.03$ | $1.37 \pm 0.09$ | $6.89 \pm 3.42$ |
| | 0.25 | $0.41 \pm 0.05$ | $\mathbf{0.91} \pm 0.07$ | $7.79 \pm 0.29$ | $52.2 \pm 0.06$ |
| | 0.50 | $0.43 \pm 0.09$ | $0.86 \pm 0.09$ | $\mathbf{9.49} \pm 0.37$ | $54.86 \pm 0.28$ |
| | 0.75 | $\mathbf{0.5} \pm 0.04$ | $0.72 \pm 0.08$ | $9.4 \pm 0.28$ | $\mathbf{55.52} \pm 0.09$ |
| | 1.00 | $0.3 \pm 0.03$ | $0.46 \pm 0.08$ | $3.07 \pm 0.07$ | $51.52 \pm 0.07$ |
| 2:4 | 0.00 | $52.28 \pm 0.04$ | $69.65 \pm 0.02$ | $76.21 \pm 0.07$ | $0.1 \pm 0.0$ |
| | 0.25 | $\mathbf{54.23} \pm 0.1$ | $71.1 \pm 0.07$ | $77.3 \pm 0.04$ | $75.37 \pm 0.02$ |
| | 0.50 | $54.2 \pm 0.15$ | $\mathbf{71.54} \pm 0.08$ | $77.67 \pm 0.07$ | $\mathbf{75.5} \pm 0.04$ |
| | 0.75 | $53.78 \pm 0.03$ | $71.52 \pm 0.05$ | $\mathbf{77.88} \pm 0.05$ | $\mathbf{75.5} \pm 0.03$ |
| | 1.00 | $47.65 \pm 0.11$ | $70.25 \pm 0.04$ | $77.74 \pm 0.04$ | $75.46 \pm 0.03$ |

Table 14: Test perplexity on C4, WikiText2 and PTB and zero-shot accuracy of Llama-3.2-1B using *MOON-SHOT*-OSSCAR. Zero-shot mean performance and win rate are computed over the 7 classification tasks. Results are averaged over 3 seeds with standard errors.

| Sparsity | λ | C4 ↓ | WikiText2 ↓ | PTB ↓ | BoolQ ↑ | HellaSwag ↑ | WinoGrande ↑ | ARC-e ↑ | ARC-c ↑ | PIQA ↑ | OBQA ↑ | Mean ↑ | Win Rate ↑ |
|---|---|---|---|---|---|---|---|---|---|---|---|---|---|
| Dense | - | 14.02 | 9.75 | 17.59 | 64.01 | 47.73 | 60.14 | 65.15 | 31.23 | 74.32 | 26.4 | 52.71 | - |
| 0.1 | 0.0 | $315.6 \pm 149.49$ | $242.7 \pm 114.26$ | $4120.96 \pm 2133.48$ | $51.86 \pm 1.12$ | $30.82 \pm 3.98$ | $50.51 \pm 1.02$ | $45.45 \pm 4.51$ | $22.41 \pm 2.1$ | $60.99 \pm 3.65$ | $15.27 \pm 1.17$ | $39.62 \pm 2.5$ | $28.57 \pm 28.57$ |
|  | 0.1 | $\mathbf{38.04} \pm 0.58$ | $30.93 \pm 0.92$ | $\mathbf{86.73} \pm 9.88$ | $\mathbf{56.55} \pm 0.9$ | $37.89 \pm 1.11$ | $\mathbf{55.51} \pm 0.37$ | $\mathbf{58.84} \pm 0.38$ | $28.58 \pm 0.51$ | $\mathbf{71.49} \pm 0.06$ | $\mathbf{18.93} \pm 0.48$ | $\mathbf{46.83} \pm 0.42$ | $80.95 \pm 12.6$ |
|  | 0.25 | $41.23 \pm 1.45$ | $33.54 \pm 1.31$ | $102.9 \pm 6.89$ | $56.02 \pm 1.61$ | $37.44 \pm 1.22$ | $54.12 \pm 0.66$ | $58.52 \pm 0.51$ | $\mathbf{28.64} \pm 0.21$ | $71.33 \pm 0.22$ | $17.8 \pm 0.42$ | $46.27 \pm 0.53$ | $76.19 \pm 4.76$ |
|  | 0.5 | $39.35 \pm 2.63$ | $\mathbf{30.62} \pm 1.23$ | $91.55 \pm 16.49$ | $54.31 \pm 3.38$ | $\mathbf{38.75} \pm 0.98$ | $55.22 \pm 0.61$ | $57.46 \pm 2.72$ | $27.9 \pm 0.51$ | $71.2 \pm 0.81$ | $17.67 \pm 1.77$ | $46.07 \pm 1.49$ | $80.95 \pm 12.6$ |
|  | 0.75 | $38.46 \pm 1.14$ | $33.27 \pm 0.24$ | $97.7 \pm 8.46$ | $52.1 \pm 2.0$ | $36.75 \pm 0.38$ | $53.64 \pm 0.5$ | $55.2 \pm 2.72$ | $27.05 \pm 0.53$ | $70.57 \pm 1.2$ | $17.47 \pm 0.74$ | $44.68 \pm 0.94$ | $80.95 \pm 9.52$ |
|  | 0.9 | $39.31 \pm 1.06$ | $37.59 \pm 0.66$ | $97.6 \pm 9.05$ | $52.57 \pm 1.72$ | $36.59 \pm 0.45$ | $55.2 \pm 1.05$ | $48.13 \pm 6.52$ | $24.91 \pm 1.96$ | $67.01 \pm 3.23$ | $17.4 \pm 0.42$ | $43.12 \pm 1.85$ | $\mathbf{85.71} \pm 8.25$ |
|  | 1.0 | $43.0 \pm 1.28$ | $43.91 \pm 0.6$ | $106.62 \pm 8.57$ | $52.61 \pm 1.63$ | $35.82 \pm 0.29$ | $54.17 \pm 1.11$ | $45.19 \pm 7.83$ | $24.23 \pm 2.04$ | $65.65 \pm 3.68$ | $15.93 \pm 0.33$ | $41.94 \pm 1.98$ | $0.0 \pm 0.0$ |
| 0.15 | 0.0 | $379.28 \pm 177.85$ | $302.81 \pm 141.13$ | $1378.15 \pm 633.62$ | $50.31 \pm 1.0$ | $29.86 \pm 3.4$ | $51.3 \pm 1.22$ | $42.82 \pm 5.83$ | $21.76 \pm 1.67$ | $60.95 \pm 4.05$ | $15.87 \pm 1.18$ | $38.98 \pm 2.56$ | $38.1 \pm 31.23$ |
|  | 0.1 | $148.98 \pm 3.3$ | $113.79 \pm 4.23$ | $392.84 \pm 35.98$ | $\mathbf{54.37} \pm 1.47$ | $30.91 \pm 0.41$ | $54.12 \pm 0.69$ | $51.37 \pm 0.27$ | $\mathbf{25.74} \pm 0.25$ | $64.54 \pm 0.48$ | $\mathbf{16.27} \pm 0.52$ | $42.48 \pm 0.35$ | $61.9 \pm 12.6$ |
|  | 0.25 | $87.21 \pm 20.52$ | $79.97 \pm 20.05$ | $365.34 \pm 138.38$ | $53.16 \pm 1.55$ | $32.68 \pm 1.47$ | $53.43 \pm 0.57$ | $\mathbf{52.62} \pm 1.92$ | $25.48 \pm 0.75$ | $66.54 \pm 1.46$ | $16.13 \pm 0.58$ | $\mathbf{42.86} \pm 1.09$ | $71.43 \pm 8.25$ |
|  | 0.5 | $64.68 \pm 9.44$ | $61.36 \pm 15.64$ | $282.32 \pm 156.56$ | $51.96 \pm 2.46$ | $33.42 \pm 1.79$ | $\mathbf{54.7} \pm 0.66$ | $46.83 \pm 9.01$ | $24.46 \pm 2.89$ | $64.91 \pm 5.71$ | $15.93 \pm 1.76$ | $41.74 \pm 3.34$ | $47.62 \pm 17.17$ |
|  | 0.75 | $48.07 \pm 2.62$ | $\mathbf{42.59} \pm 0.5$ | $117.8 \pm 4.46$ | $47.82 \pm 1.19$ | $\mathbf{36.05} \pm 0.43$ | $54.06 \pm 0.46$ | $49.51 \pm 5.8$ | $25.11 \pm 1.77$ | $\mathbf{68.03} \pm 2.48$ | $15.8 \pm 0.58$ | $42.34 \pm 1.68$ | $66.67 \pm 4.76$ |
|  | 0.9 | $\mathbf{47.48} \pm 1.15$ | $46.86 \pm 1.09$ | $\mathbf{102.39} \pm 3.55$ | $48.88 \pm 2.07$ | $35.65 \pm 0.5$ | $54.22 \pm 0.32$ | $46.49 \pm 6.13$ | $25.17 \pm 1.7$ | $66.74 \pm 2.73$ | $15.93 \pm 0.74$ | $41.87 \pm 1.83$ | $\mathbf{76.19} \pm 9.52$ |
|  | 1.0 | $52.65 \pm 1.85$ | $54.51 \pm 1.35$ | $132.62 \pm 9.26$ | $51.23 \pm 2.07$ | $35.22 \pm 0.32$ | $52.83 \pm 0.43$ | $45.61 \pm 6.45$ | $24.23 \pm 1.72$ | $66.12 \pm 2.69$ | $14.53 \pm 0.55$ | $41.4 \pm 1.74$ | $0.0 \pm 0.0$ |

Table 15: Test perplexity on C4, WikiText2 and PTB and zero-shot accuracy of Llama-3.2-1B using *MOON-SHOT*-SparseGPT. Zero-shot mean performance and win rate are computed over the 7 classification tasks. Results are averaged over 3 seeds with standard errors.

| Sparsity | λ | C4 ↓ | WikiText2 ↓ | PTB ↓ | BoolQ ↑ | HellaSwag ↑ | WinoGrande ↑ | ARC-e ↑ | ARC-c ↑ | PIQA ↑ | OBQA ↑ | Mean ↑ | Win Rate ↑ |
|---|---|---|---|---|---|---|---|---|---|---|---|---|---|
| Dense | - | 14.02 | 9.75 | 17.59 | 64.01 | 47.73 | 60.14 | 65.15 | 31.23 | 74.32 | 26.4 | 52.71 | - |
| 0.5 | 0.0 | 29.14±0.16 | 22.03±0.15 | 36.04±0.38 | 60.08±0.67 | 36.28±0.07 | 54.06±0.67 | 52.83±0.16 | 24.97±0.5 | 66.92±0.27 | 17.8±0.12 | 44.71±0.1 | 9.52±4.76 |
| | 0.1 | 24.77±0.09 | 18.43±0.09 | 30.72±0.35 | 62.18±0.67 | 38.25±0.03 | 55.38±0.73 | 55.39±0.49 | 26.68±0.3 | 68.12±0.05 | 19.6±0.5 | 46.51±0.16 | 33.33±4.76 |
| | 0.25 | 24.31±0.04 | 17.93±0.1 | 30.25±0.41 | 62.33±0.77 | 38.52±0.14 | 55.01±0.14 | 55.46±0.4 | 26.25±0.21 | **68.81**±0.07 | 18.87±0.47 | 46.46±0.13 | 38.1±9.52 |
| | 0.5 | 24.01±0.08 | 17.74±0.14 | 29.72±0.34 | 62.6±0.18 | 38.61±0.07 | **55.54**±0.34 | **56.0**±0.36 | 26.05±0.12 | 68.3±0.15 | 19.07±0.41 | 46.6±0.12 | 28.57±14.29 |
| | 0.75 | 23.77±0.1 | 17.53±0.1 | 29.17±0.14 | **63.16**±0.11 | 38.71±0.16 | 54.78±0.56 | 55.82±0.49 | 26.05±0.37 | 68.63±0.33 | 20.0±0.5 | 46.74±0.2 | 38.1±9.52 |
| | 0.9 | **23.7**±0.06 | **17.49**±0.12 | **29.05**±0.16 | 63.14±0.31 | 38.86±0.12 | 55.38±0.14 | 55.6±0.42 | **26.96**±0.34 | 68.59±0.42 | 20.27±0.58 | **46.97**±0.22 | **47.62**±9.52 |
| | 1.0 | 27.15±0.23 | 19.98±0.17 | 33.63±0.1 | 61.67±1.48 | **39.19**±0.08 | 55.51±0.38 | 55.16±0.53 | 26.51±0.53 | 68.59±0.18 | **21.87**±0.29 | 46.93±0.32 | - |
| 0.5 (Alpha-Pruning) | 0.0 | 29.64±0.28 | 22.33±0.14 | 36.34±0.53 | 61.12±0.36 | 36.36±0.19 | 55.96±0.57 | 51.95±0.49 | 25.63±1.05 | 66.92±0.27 | 18.67±0.37 | 45.23±0.35 | 14.29±8.25 |
| | 0.1 | 24.9±0.1 | 18.44±0.09 | 30.33±0.33 | 61.56±0.98 | 38.33±0.09 | **56.06**±0.53 | 54.8±0.4 | 26.34±0.08 | 68.73±0.26 | 19.27±0.37 | 46.44±0.13 | 28.57±8.25 |
| | 0.25 | 24.47±0.13 | 18.05±0.16 | 29.97±0.24 | 62.6±0.32 | 38.62±0.06 | 55.33±0.24 | 54.98±0.27 | **26.91**±0.28 | 68.81±0.47 | 19.87±0.94 | 46.73±0.17 | **42.86**±8.25 |
| | 0.5 | 24.17±0.11 | 17.79±0.19 | 29.57±0.35 | 62.01±1.3 | 38.79±0.08 | 55.33±0.38 | 55.12±0.12 | 26.48±0.64 | 68.66±0.51 | 19.0±0.42 | 46.48±0.1 | 23.81±12.6 |
| | 0.75 | 23.93±0.1 | 17.58±0.15 | 29.26±0.38 | **63.13**±0.22 | 38.89±0.07 | 54.49±0.74 | 55.43±0.06 | 26.54±0.5 | 68.72±0.35 | 19.8±0.5 | 46.71±0.12 | 33.33±17.17 |
| | 0.9 | **23.77**±0.06 | **17.48**±0.17 | **29.11**±0.38 | 62.81±0.4 | 39.1±0.1 | 55.59±0.27 | **55.71**±0.28 | 26.31±0.32 | 68.81±0.29 | 20.13±0.57 | 46.92±0.09 | 38.1±19.05 |
| | 1.0 | 27.03±0.29 | 19.85±0.21 | 32.85±0.24 | 62.35±1.39 | **39.12**±0.17 | 55.75±0.43 | 55.2±0.6 | 26.54±0.3 | **68.81**±0.23 | **21.67**±0.29 | **47.06**±0.14 | - |
| 0.5 (OWL) | 0.0 | 27.44±0.1 | 21.25±0.1 | 34.47±0.26 | 61.41±0.56 | 37.54±0.08 | 55.3±0.34 | 51.54±0.43 | 25.31±0.4 | 67.46±0.11 | 18.6±1.03 | 45.31±0.2 | 0.0±0.0 |
| | 0.1 | 24.13±0.07 | 18.29±0.11 | 30.68±0.46 | 62.68±0.3 | 39.11±0.07 | 56.35±0.61 | 55.39±0.82 | 26.25±0.36 | 67.92±0.15 | 20.53±0.68 | 46.89±0.29 | 23.81±17.17 |
| | 0.25 | 23.84±0.08 | 18.02±0.15 | 29.81±0.27 | 62.76±0.2 | 39.15±0.06 | 56.38±0.16 | 55.32±0.67 | 26.96±0.09 | 68.46±0.14 | 20.8±0.2 | 47.12±0.11 | 38.1±12.6 |
| | 0.5 | 23.57±0.07 | 17.77±0.1 | 29.69±0.13 | 62.77±0.45 | 39.4±0.11 | 55.99±0.47 | 55.44±0.59 | 26.93±0.49 | **68.63**±0.28 | 20.27±0.37 | 47.06±0.27 | 42.86±14.29 |
| | 0.75 | 23.41±0.05 | 17.69±0.09 | 29.38±0.19 | **63.12**±0.5 | 39.46±0.05 | 56.8±0.73 | 55.88±0.23 | **27.45**±0.28 | 68.48±0.27 | 20.67±0.47 | **47.41**±0.22 | **57.14**±8.25 |
| | 0.9 | **23.31**±0.04 | **17.58**±0.11 | **29.16**±0.31 | 62.88±0.34 | 39.56±0.1 | **56.96**±0.78 | **56.05**±0.35 | 27.08±0.12 | 68.53±0.13 | 20.27±0.35 | 47.33±0.16 | 52.38±12.6 |
| | 1.0 | 26.25±0.05 | 19.74±0.09 | 32.2±0.26 | 62.86±0.31 | **39.94**±0.08 | 56.59±0.55 | 55.36±0.32 | 26.42±0.36 | 68.32±0.22 | **22.13**±0.37 | 47.37±0.15 | - |
| 0.6 | 0.0 | 85.09±1.34 | 72.05±0.81 | 104.4±1.31 | 60.89±1.43 | 29.47±0.16 | 53.17±1.07 | 39.46±0.42 | 19.51±0.33 | 59.97±0.11 | 13.93±0.74 | 39.49±0.36 | 14.29±0.0 |
| | 0.1 | 56.05±1.38 | 44.2±0.94 | 67.85±2.09 | 61.47±0.23 | 31.68±0.11 | 52.64±0.39 | 44.99±0.81 | 21.42±0.52 | 62.44±0.37 | 15.47±0.84 | 41.44±0.18 | 38.1±12.6 |
| | 0.25 | 54.15±2.38 | 42.06±1.34 | 63.21±1.81 | 62.15±0.2 | 31.99±0.12 | 52.97±0.39 | 45.19±0.52 | 21.33±0.2 | 62.6±0.18 | 14.87±0.59 | 41.53±0.1 | 47.62±12.6 |
| | 0.5 | 51.97±1.64 | 40.32±1.13 | 61.81±2.66 | 62.19±0.07 | 32.13±0.18 | 52.91±0.45 | 46.34±0.47 | 21.3±0.23 | 62.88±0.09 | 15.87±0.24 | 41.94±0.16 | 52.38±9.52 |
| | 0.75 | 51.28±1.68 | 39.69±1.27 | 60.91±2.28 | 62.16±0.04 | 32.38±0.19 | 53.38±0.32 | 46.25±0.34 | 21.53±0.21 | **63.31**±0.13 | 15.67±0.29 | 42.1±0.13 | **57.14**±8.25 |
| | 0.9 | **50.28**±1.99 | **39.13**±1.54 | **60.14**±2.9 | **62.36**±0.12 | **32.49**±0.13 | 53.09±0.18 | **46.49**±0.38 | 21.3±0.24 | 63.22±0.17 | 15.73±0.55 | **42.1**±0.11 | **57.14**±8.25 |
| | 1.0 | 63.63±1.18 | 54.6±1.0 | 81.11±3.99 | 60.67±0.59 | 32.16±0.2 | **54.46**±0.53 | 44.94±0.11 | 21.47±0.48 | 62.21±0.2 | **17.07**±0.41 | 41.85±0.2 | - |
| 0.6 (Alpha-Pruning) | 0.0 | 88.67±2.38 | 75.35±3.09 | 109.66±5.07 | 61.52±0.51 | 29.5±0.03 | 51.64±0.47 | 38.61±0.43 | 19.48±0.45 | 59.5±0.27 | 12.67±0.47 | 38.99±0.28 | 4.76±4.76 |
| | 0.1 | 57.24±0.41 | 45.06±0.23 | 68.58±0.08 | 61.88±0.24 | 31.64±0.07 | 51.8±0.38 | 44.89±0.87 | 21.13±0.34 | 61.95±0.41 | 15.73±0.59 | 41.29±0.33 | 19.05±9.52 |
| | 0.25 | 53.93±0.37 | 42.36±0.2 | 64.63±1.4 | 62.13±0.04 | 31.89±0.07 | 52.7±0.6 | 45.23±0.44 | 21.76±0.15 | 61.82±0.19 | 15.87±0.64 | 41.63±0.22 | 28.57±0.0 |
| | 0.5 | 51.22±0.44 | 40.66±0.49 | 62.51±1.04 | 62.22±0.08 | 32.16±0.06 | 53.38±0.74 | 45.74±0.58 | 22.07±0.37 | 62.79±0.17 | 15.93±0.13 | 42.04±0.13 | 14.29±14.29 |
| | 0.75 | 50.52±0.55 | 39.44±0.3 | 61.78±0.85 | 62.06±0.22 | 32.38±0.07 | 53.33±0.39 | **46.68**±0.6 | **22.21**±0.22 | 62.66±0.16 | 16.13±0.18 | 42.21±0.21 | 61.9±4.76 |
| | 0.9 | **49.31**±1.1 | **38.44**±0.52 | **60.32**±1.2 | **62.29**±0.05 | **32.53**±0.06 | 53.7±0.55 | 46.3±0.3 | 21.99±0.15 | 62.66±0.12 | 16.6±0.12 | **42.36**±0.14 | **66.67**±9.52 |
| | 1.0 | 61.05±0.77 | 52.8±0.43 | 78.27±3.64 | 62.08±0.26 | 32.0±0.08 | **53.88**±0.25 | 45.29±0.49 | 22.01±0.59 | 62.02±0.42 | **17.6**±0.42 | 42.13±0.06 | - |
| 0.6 (OWL) | 0.0 | 74.97±1.68 | 64.7±1.38 | 94.32±3.39 | 62.11±0.11 | 30.75±0.2 | 52.07±0.57 | 40.39±0.1 | 21.05±0.37 | 61.08±0.18 | 14.2±0.5 | 40.23±0.08 | 4.76±4.76 |
| | 0.1 | 48.43±0.56 | 40.09±0.31 | 59.81±1.44 | 62.22±0.05 | 32.92±0.08 | 52.59±0.22 | 45.69±0.51 | 22.04±0.45 | 63.58±0.13 | 16.13±0.52 | 42.17±0.13 | 42.86±8.25 |
| | 0.25 | 46.31±0.78 | 38.41±0.17 | 56.18±0.88 | **62.27**±0.1 | 33.06±0.06 | 52.99±0.51 | 45.17±0.32 | 22.64±0.42 | 64.02±0.12 | 15.8±0.61 | 42.28±0.11 | 61.9±9.52 |
| | 0.5 | 45.28±1.06 | 37.36±0.54 | 54.54±1.37 | 62.21±0.02 | 33.25±0.2 | 53.62±0.38 | 45.99±0.5 | 23.07±0.79 | **64.2**±0.14 | 16.6±0.12 | 42.71±0.26 | **71.43**±12.6 |
| | 0.75 | 44.3±0.89 | 36.24±0.24 | 53.87±1.29 | **62.27**±0.09 | 33.48±0.16 | 54.06±0.44 | **46.89**±0.15 | 23.15±0.4 | 63.89±0.19 | 16.4±0.72 | **42.88**±0.23 | 66.67±12.6 |
| | 0.9 | **43.58**±0.91 | **35.72**±0.17 | **53.02**±0.91 | 62.2±0.04 | **33.66**±0.16 | 53.54±0.55 | 46.37±0.23 | 23.41±0.08 | 64.04±0.03 | 16.4±0.2 | 42.8±0.07 | 66.67±17.17 |
| | 1.0 | 56.82±1.68 | 49.54±1.22 | 68.73±3.48 | 62.2±0.11 | 32.86±0.05 | 53.67±0.33 | 44.14±0.71 | **23.46**±0.05 | 62.59±0.21 | **18.0**±0.76 | 42.42±0.07 | - |
| 0.7 | 0.0 | 553.89±45.52 | 729.08±63.3 | 855.94±159.79 | 51.74±5.13 | 26.8±0.12 | 49.67±0.3 | 29.48±0.18 | **19.34**±0.08 | 54.64±0.48 | 12.67±0.44 | 34.91±0.68 | 23.81±12.6 |
| | 0.1 | 254.69±7.12 | 246.7±5.75 | 307.55±27.07 | **58.47**±1.87 | 27.28±0.11 | 51.09±0.74 | 32.84±0.26 | 18.52±0.56 | 56.33±0.27 | 12.47±0.64 | 36.71±0.13 | 38.1±12.6 |
| | 0.25 | 234.35±6.29 | 218.38±8.22 | 276.61±20.83 | 54.34±3.3 | 27.66±0.08 | 50.64±0.61 | 33.33±0.55 | 18.63±0.08 | 56.42±0.52 | 13.07±0.44 | 36.3±0.41 | 47.62±12.6 |
| | 0.5 | 214.85±7.77 | 195.04±7.91 | 250.95±18.88 | 56.87±2.54 | 27.82±0.1 | 51.14±0.12 | 33.78±0.7 | 18.26±0.13 | 56.93±0.15 | 12.53±0.77 | 36.76±0.35 | 47.62±12.6 |
| | 0.75 | 209.63±6.29 | 182.92±2.06 | 239.21±15.09 | 56.02±3.2 | 27.77±0.07 | 50.91±1.1 | 34.18±0.56 | 18.57±0.21 | **57.15**±0.71 | 12.73±0.47 | 36.76±0.43 | 57.14±14.29 |
| | 0.9 | **202.38**±9.85 | **173.51**±0.99 | **233.86**±7.17 | 56.59±2.22 | **27.85**±0.11 | **51.25**±0.18 | **34.67**±0.66 | 18.94±0.09 | 57.13±0.22 | 13.27±0.64 | **37.1**±0.22 | **66.67**±12.6 |
| | 1.0 | 303.94±19.17 | 348.03±10.5 | 452.29±13.24 | 57.73±1.52 | 27.77±0.15 | 50.86±0.53 | 33.12±0.33 | 18.49±0.64 | 56.64±0.31 | **13.73**±0.85 | 36.9±0.28 | - |
| 0.7 (Alpha-Pruning) | 0.0 | 525.41±51.54 | 620.51±27.79 | 843.16±128.6 | 48.66±6.73 | 26.62±0.1 | 50.33±0.26 | 30.04±0.1 | **19.14**±0.45 | 54.62±0.28 | 11.8±0.12 | 34.46±0.99 | 38.1±9.52 |
| | 0.1 | 244.61±13.76 | 227.67±15.03 | 288.52±17.99 | 54.81±2.53 | 27.36±0.11 | 50.07±0.39 | 32.58±0.32 | 18.17±0.3 | 55.57±0.57 | 13.13±0.47 | 35.96±0.44 | 52.38±9.52 |
| | 0.25 | 222.33±7.61 | 202.07±10.7 | 258.32±17.98 | 55.55±3.16 | 27.6±0.01 | 49.72±0.59 | 33.33±0.3 | 19.11±0.17 | 56.29±0.36 | 12.93±0.64 | 36.36±0.43 | 66.67±12.6 |
| | 0.5 | 206.42±2.89 | 183.41±5.59 | 233.52±6.74 | 55.88±3.09 | 27.6±0.06 | 49.78±0.53 | **33.74**±0.39 | 18.57±0.41 | 56.78±0.52 | 12.67±0.52 | 36.43±0.57 | 66.67±12.6 |
| | 0.75 | 210.6±4.5 | 181.88±5.4 | 236.62±3.31 | 54.06±3.58 | 27.64±0.06 | **50.51**±0.73 | 33.7±0.28 | 18.71±0.41 | 56.84±0.38 | 13.0±0.42 | 36.35±0.59 | 66.67±9.52 |
| | 0.9 | **196.7**±0.54 | **168.34**±6.38 | **224.51**±7.57 | 56.35±2.57 | **27.86**±0.06 | 49.41±0.77 | 33.66±0.18 | 18.57±0.63 | **56.93**±0.5 | **13.6**±0.53 | **36.62**±0.52 | **71.43**±8.25 |
| | 1.0 | 316.17±25.45 | 365.32±38.53 | 439.46±61.03 | **56.43**±1.87 | 27.6±0.18 | 48.93±0.41 | 32.44±0.88 | 18.71±0.42 | 56.44±0.34 | 12.87±0.64 | 36.2±0.28 | - |
| 0.7 (OWL) | 0.0 | 454.15±42.28 | 570.39±46.9 | 853.1±184.13 | 54.44±3.84 | 26.77±0.04 | 50.46±0.71 | 30.92±0.21 | **20.14**±0.37 | 55.3±0.28 | 11.73±0.41 | 35.68±0.39 | 19.05±9.52 |
| | 0.1 | 216.91±5.44 | 217.4±7.67 | 371.76±64.08 | 55.79±4.1 | 27.76±0.09 | 50.25±0.29 | 33.35±0.42 | 19.11±0.36 | 56.82±0.18 | 12.0±0.42 | 36.44±0.63 | 38.1±4.76 |
| | 0.25 | 202.68±4.99 | 197.03±3.68 | 299.34±24.53 | 54.85±4.13 | 27.91±0.1 | 50.99±0.51 | 34.13±0.25 | 19.03±0.4 | 57.27±0.44 | 12.2±0.31 | 36.63±0.63 | 52.38±9.52 |
| | 0.5 | 186.63±4.58 | 174.83±5.25 | 264.37±13.89 | **58.64**±1.87 | 28.02±0.13 | 50.2±0.21 | 34.06±0.6 | 19.06±0.28 | 57.2±0.42 | 13.13±0.57 | 37.19±0.37 | 61.9±17.17 |
| | 0.75 | 184.44±6.26 | 172.63±7.5 | 268.01±10.51 | 56.79±2.28 | 28.03±0.07 | 51.7±0.91 | **35.02**±0.2 | 18.71±0.42 | 57.29±0.14 | 12.73±0.41 | 37.18±0.42 | **66.67**±4.76 |
| | 0.9 | **178.12**±5.92 | **167.31**±6.39 | **241.44**±6.24 | 56.39±2.51 | **28.06**±0.1 | **52.25**±1.03 | 34.88±0.2 | 18.63±0.37 | **57.98**±0.05 | 12.87±0.29 | **37.29**±0.45 | 57.14±0.0 |
| | 1.0 | 263.82±27.07 | 313.26±43.34 | 390.13±86.99 | 57.86±1.51 | 27.93±0.11 | 49.88±0.71 | 33.16±0.36 | 19.37±0.2 | 56.29±0.36 | **13.67**±0.07 | 36.88±0.31 | - |
| 2:4 | 0.0 | 77.52±0.12 | 63.66±0.35 | 88.79±1.7 | 60.73±0.32 | 29.52±0.06 | 51.72±0.41 | 40.04±0.19 | 19.08±0.44 | 60.07±0.32 | 13.33±0.18 | 39.21±0.23 | 0.0±0.0 |
| | 0.1 | 55.22±0.36 | 41.5±0.92 | 63.2±1.74 | 61.8±0.13 | 31.12±0.13 | 53.59±0.37 | 44.11±0.23 | 20.05±0.0 | 61.75±0.36 | 14.27±0.24 | 40.96±0.11 | 23.81±0.0 |
| | 0.25 | 53.63±0.46 | 40.17±0.78 | 63.09±1.21 | 61.69±0.28 | 31.29±0.18 | 54.09±0.22 | 44.75±0.23 | 20.08±0.3 | 61.72±0.26 | 14.53±0.48 | 41.17±0.09 | 42.86±8.25 |
| | 0.5 | 51.89±0.33 | 39.22±0.59 | 60.78±1.41 | 61.45±0.27 | 31.42±0.15 | **54.38**±0.6 | 44.96±0.06 | 20.36±0.41 | 62.02±0.14 | 14.13±0.27 | 41.25±0.1 | 52.38±12.6 |
| | 0.75 | **50.99**±0.47 | **38.0**±0.58 | 59.32±1.55 | **61.98**±0.29 | 31.47±0.21 | 53.09±0.27 | **45.37**±0.5 | 20.28±0.58 | 62.13±0.19 | 14.33±0.53 | 41.24±0.2 | 42.86±14.29 |
| | 0.9 | 51.02±0.33 | 38.22±0.48 | **59.09**±1.07 | 61.77±0.35 | **31.69**±0.18 | 53.51±0.51 | 45.31±0.08 | 20.51±0.32 | **62.53**±0.18 | 14.73±0.35 | **41.44**±0.16 | **57.14**±16.5 |
| | 1.0 | 53.59±0.35 | 42.56±0.37 | 63.79±0.19 | 61.42±0.21 | 31.68±0.05 | 53.83±0.37 | 44.04±0.23 | **21.47**±0.44 | 61.79±0.4 | **15.0**±0.2 | 41.32±0.08 | - |

Table 16: Test perplexity on C4, WikiText2 and PTB and zero-shot accuracy of Llama-3.2-1B using *MOON-SHOT*-Wanda. Zero-shot mean performance and win rate are computed over the 7 classification tasks. Results are averaged over 3 seeds with standard errors.

| Sparsity | λ | C4 ↓ | WikiText2 ↓ | PTB ↓ | BoolQ ↑ | HellaSwag ↑ | WinoGrande ↑ | ARC-e ↑ | ARC-c ↑ | PIQA ↑ | OBQA ↑ | Mean ↑ | Win Rate ↑ |
|---|---|---|---|---|---|---|---|---|---|---|---|---|---|
| Dense | - | 14.02 | 9.75 | 17.59 | 64.01 | 47.73 | 60.14 | 65.15 | 31.23 | 74.32 | 26.4 | 52.71 | - |
| 0.5 | 0.0 | 30.48 ± 0.14 | 21.48 ± 0.08 | 39.41 ± 0.22 | 61.59 ± 0.37 | 35.57 ± 0.08 | 54.14 ± 0.6 | 54.32 ± 0.28 | 24.94 ± 0.12 | 65.96 ± 0.21 | 17.8 ± 0.7 | 44.9 ± 0.21 | 80.95 ± 4.76 |
| | 0.1 | 30.26 ± 0.17 | 21.32 ± 0.12 | 38.76 ± 0.18 | 61.85 ± 0.22 | 35.59 ± 0.13 | 54.51 ± 0.11 | 54.66 ± 0.28 | 24.94 ± 0.1 | 66.36 ± 0.02 | 17.93 ± 0.7 | 45.12 ± 0.17 | 80.95 ± 4.76 |
| | 0.25 | 30.08 ± 0.09 | 21.1 ± 0.07 | 38.19 ± 0.04 | 61.98 ± 0.19 | 35.68 ± 0.13 | 53.96 ± 0.3 | **54.7** ± 0.44 | 25.2 ± 0.15 | 66.29 ± 0.1 | 18.0 ± 1.01 | 45.11 ± 0.14 | 80.95 ± 4.76 |
| | 0.5 | 29.82 ± 0.04 | 20.81 ± 0.04 | 37.68 ± 0.25 | 61.99 ± 0.27 | 35.91 ± 0.11 | 54.14 ± 0.6 | 54.69 ± 0.18 | 25.2 ± 0.41 | **66.56** ± 0.13 | **18.27** ± 0.44 | **45.25** ± 0.19 | **85.71** ± 8.25 |
| | 0.75 | **29.63** ± 0.04 | **20.69** ± 0.08 | 37.32 ± 0.16 | 62.06 ± 0.18 | 35.95 ± 0.06 | 54.46 ± 0.05 | 54.45 ± 0.21 | **25.31** ± 0.12 | 66.2 ± 0.19 | 17.8 ± 0.23 | 45.18 ± 0.08 | 71.43 ± 0.0 |
| | 0.9 | 29.7 ± 0.08 | **20.69** ± 0.05 | **37.23** ± 0.1 | **62.13** ± 0.13 | **36.16** ± 0.06 | **54.59** ± 0.21 | 54.31 ± 0.15 | 25.26 ± 0.31 | 66.34 ± 0.11 | 17.4 ± 0.42 | 45.17 ± 0.1 | 80.95 ± 4.76 |
| | 1.0 | 35.71 ± 0.21 | 24.43 ± 0.04 | 43.15 ± 0.26 | 60.23 ± 0.49 | 35.26 ± 0.04 | 54.56 ± 0.03 | 51.59 ± 0.29 | 24.69 ± 0.21 | 65.51 ± 0.13 | 18.2 ± 0.2 | 44.29 ± 0.07 | - |
| 0.5 (Alpha-Pruning) | 0.0 | 30.46 ± 0.16 | 21.25 ± 0.11 | 38.88 ± 0.19 | 61.15 ± 0.09 | 35.64 ± 0.07 | 54.91 ± 0.39 | 53.62 ± 0.09 | 24.32 ± 0.13 | 66.29 ± 0.23 | 17.73 ± 0.24 | 44.81 ± 0.09 | 80.95 ± 4.76 |
| | 0.1 | 30.28 ± 0.13 | 21.1 ± 0.08 | 38.24 ± 0.19 | **61.78** ± 0.08 | 35.67 ± 0.09 | 54.33 ± 0.32 | 53.87 ± 0.17 | 24.12 ± 0.25 | 66.3 ± 0.07 | 17.8 ± 0.72 | 44.84 ± 0.18 | 71.43 ± 8.25 |
| | 0.25 | 30.24 ± 0.16 | 21.01 ± 0.05 | 38.06 ± 0.08 | 61.67 ± 0.15 | 35.74 ± 0.11 | 54.35 ± 0.26 | **54.12** ± 0.22 | 24.32 ± 0.13 | 66.23 ± 0.07 | **18.0** ± 0.7 | 44.92 ± 0.15 | 80.95 ± 4.76 |
| | 0.5 | 29.93 ± 0.18 | 20.74 ± 0.09 | 37.38 ± 0.08 | 61.58 ± 0.2 | 35.79 ± 0.08 | 54.75 ± 0.13 | 54.05 ± 0.09 | 24.26 ± 0.16 | 66.61 ± 0.19 | **18.0** ± 0.61 | 45.01 ± 0.09 | 80.95 ± 4.76 |
| | 0.75 | **29.57** ± 0.03 | 20.5 ± 0.04 | **37.03** ± 0.1 | 61.64 ± 0.13 | 35.92 ± 0.01 | **55.46** ± 0.48 | 53.98 ± 0.15 | 24.63 ± 0.21 | 66.65 ± 0.11 | **18.0** ± 0.31 | **45.18** ± 0.11 | 85.71 ± 8.25 |
| | 0.9 | 29.64 ± 0.06 | **20.48** ± 0.07 | 37.13 ± 0.14 | 61.41 ± 0.13 | **36.08** ± 0.03 | 55.38 ± 0.35 | 53.7 ± 0.08 | **24.66** ± 0.21 | **66.83** ± 0.04 | 17.93 ± 0.13 | 45.14 ± 0.03 | **90.48** ± 4.76 |
| | 1.0 | 35.47 ± 0.21 | 23.99 ± 0.11 | 42.65 ± 0.3 | 58.79 ± 0.37 | 35.31 ± 0.13 | 55.09 ± 0.32 | 51.07 ± 0.08 | 24.57 ± 0.31 | 65.14 ± 0.19 | 17.27 ± 0.24 | 43.89 ± 0.12 | - |
| 0.5 (OWL) | 0.0 | 28.88 ± 0.09 | 21.1 ± 0.1 | 38.01 ± 0.37 | 61.68 ± 0.12 | 36.48 ± 0.04 | 54.54 ± 0.51 | 52.95 ± 0.43 | 24.32 ± 0.13 | 66.16 ± 0.05 | 19.2 ± 0.2 | 45.05 ± 0.14 | 57.14 ± 8.25 |
| | 0.1 | 28.8 ± 0.09 | 20.98 ± 0.09 | 37.79 ± 0.41 | 61.82 ± 0.2 | 36.52 ± 0.11 | 55.12 ± 0.43 | 52.92 ± 0.56 | 24.43 ± 0.32 | 66.23 ± 0.29 | 19.27 ± 0.24 | 45.19 ± 0.17 | 76.19 ± 4.76 |
| | 0.25 | 28.82 ± 0.12 | 20.9 ± 0.1 | 37.48 ± 0.36 | **62.08** ± 0.09 | 36.66 ± 0.06 | 54.75 ± 0.07 | **53.3** ± 0.56 | 24.74 ± 0.09 | **66.52** ± 0.16 | 19.53 ± 0.29 | 45.37 ± 0.09 | 76.19 ± 12.6 |
| | 0.5 | 28.7 ± 0.13 | 20.71 ± 0.13 | 36.85 ± 0.42 | 61.96 ± 0.2 | 36.79 ± 0.05 | 54.78 ± 0.36 | 53.04 ± 0.21 | **25.11** ± 0.21 | 66.38 ± 0.25 | 19.93 ± 0.07 | 45.43 ± 0.06 | 85.71 ± 8.25 |
| | 0.75 | **28.51** ± 0.04 | **20.58** ± 0.07 | 36.22 ± 0.29 | 61.98 ± 0.02 | 36.76 ± 0.04 | 55.25 ± 0.44 | 53.24 ± 0.13 | 24.69 ± 0.38 | 66.49 ± 0.14 | 19.87 ± 0.07 | 45.47 ± 0.11 | 85.71 ± 8.25 |
| | 0.9 | 28.52 ± 0.03 | 20.59 ± 0.02 | **36.18** ± 0.12 | 61.82 ± 0.32 | **37.0** ± 0.06 | **55.56** ± 0.45 | 52.89 ± 0.18 | 24.89 ± 0.29 | 66.36 ± 0.2 | **20.27** ± 0.29 | **45.54** ± 0.15 | **95.24** ± 4.76 |
| | 1.0 | 33.38 ± 0.16 | 23.16 ± 0.06 | 40.06 ± 0.08 | 60.01 ± 0.77 | 36.63 ± 0.05 | 54.83 ± 0.4 | 50.53 ± 0.26 | 24.66 ± 0.38 | 65.42 ± 0.1 | 19.47 ± 0.29 | 44.51 ± 0.27 | - |
| 0.6 | 0.0 | 88.87 ± 1.62 | 66.78 ± 1.37 | 108.78 ± 4.4 | **61.99** ± 0.11 | 29.39 ± 0.07 | 51.14 ± 0.3 | 39.86 ± 0.39 | 19.0 ± 0.42 | 60.52 ± 0.14 | 13.27 ± 0.35 | 39.31 ± 0.12 | 80.95 ± 12.6 |
| | 0.1 | 87.45 ± 2.13 | 65.41 ± 1.82 | 106.65 ± 4.21 | 61.78 ± 0.22 | 29.47 ± 0.03 | **52.35** ± 0.53 | 40.31 ± 0.35 | 19.2 ± 0.15 | 60.23 ± 0.33 | 13.67 ± 0.24 | 39.57 ± 0.06 | 80.95 ± 12.6 |
| | 0.25 | 86.94 ± 2.1 | 64.92 ± 2.09 | 103.22 ± 3.32 | 61.9 ± 0.11 | 29.56 ± 0.04 | 51.72 ± 0.59 | 40.4 ± 0.06 | 19.37 ± 0.13 | 60.66 ± 0.16 | 13.73 ± 0.37 | 39.59 ± 0.06 | **85.71** ± 8.25 |
| | 0.5 | **86.55** ± 1.67 | **63.57** ± 1.61 | 98.44 ± 3.98 | 61.56 ± 0.29 | 29.53 ± 0.06 | 51.64 ± 0.23 | 40.4 ± 0.17 | **19.6** ± 0.06 | **61.12** ± 0.08 | 13.4 ± 0.2 | 39.61 ± 0.07 | 80.95 ± 4.76 |
| | 0.75 | 87.41 ± 0.53 | 63.83 ± 0.69 | **96.72** ± 2.62 | 61.57 ± 0.04 | 29.57 ± 0.04 | 52.33 ± 0.76 | 40.95 ± 0.15 | 19.03 ± 0.09 | 60.9 ± 0.07 | 13.4 ± 0.23 | 39.67 ± 0.12 | 76.19 ± 9.52 |
| | 0.9 | 90.04 ± 0.69 | 64.72 ± 0.13 | 97.73 ± 0.94 | 61.64 ± 0.21 | **29.64** ± 0.04 | 52.17 ± 0.44 | **41.11** ± 0.38 | 19.06 ± 0.1 | 61.1 ± 0.11 | 13.33 ± 0.37 | **39.72** ± 0.07 | 80.95 ± 12.6 |
| | 1.0 | 117.71 ± 0.87 | 84.73 ± 0.73 | 119.64 ± 1.0 | 58.96 ± 1.39 | 28.86 ± 0.03 | 51.35 ± 0.49 | 38.82 ± 0.32 | 18.94 ± 0.26 | 59.05 ± 0.18 | **13.93** ± 0.24 | 38.56 ± 0.12 | - |
| 0.6 (Alpha-Pruning) | 0.0 | 87.66 ± 1.09 | 66.36 ± 0.7 | 107.17 ± 2.93 | **61.94** ± 0.15 | 29.34 ± 0.1 | 51.49 ± 0.39 | 39.06 ± 0.38 | 19.62 ± 0.31 | 59.99 ± 0.07 | 12.4 ± 0.53 | 39.12 ± 0.08 | 71.43 ± 0.0 |
| | 0.1 | 86.35 ± 1.64 | 65.13 ± 1.22 | 104.51 ± 3.58 | 61.79 ± 0.29 | 29.49 ± 0.12 | 51.7 ± 0.34 | 39.16 ± 0.7 | 19.74 ± 0.08 | 60.28 ± 0.06 | 12.4 ± 0.5 | 39.22 ± 0.13 | **85.71** ± 8.25 |
| | 0.25 | **84.21** ± 1.04 | 63.3 ± 0.92 | 101.96 ± 2.98 | 61.69 ± 0.24 | 29.56 ± 0.08 | **52.33** ± 0.37 | 39.28 ± 0.29 | 19.65 ± 0.06 | 60.32 ± 0.13 | 12.07 ± 0.58 | 39.27 ± 0.09 | 80.95 ± 4.76 |
| | 0.5 | 84.9 ± 0.94 | 62.75 ± 1.05 | 98.54 ± 2.09 | 61.86 ± 0.21 | **29.67** ± 0.09 | 51.93 ± 0.18 | 39.96 ± 0.26 | 19.74 ± 0.4 | 60.39 ± 0.27 | 12.13 ± 0.37 | 39.38 ± 0.04 | 76.19 ± 4.76 |
| | 0.75 | 85.33 ± 0.55 | **62.02** ± 0.85 | **97.49** ± 1.57 | 61.46 ± 0.25 | 29.58 ± 0.1 | 51.83 ± 0.21 | 40.07 ± 0.49 | 19.43 ± 0.23 | 60.83 ± 0.05 | 12.47 ± 0.13 | 39.38 ± 0.02 | 71.43 ± 0.0 |
| | 0.9 | 88.49 ± 0.32 | 63.13 ± 0.36 | 100.53 ± 0.5 | 61.51 ± 0.27 | **29.67** ± 0.1 | 51.75 ± 0.09 | **40.32** ± 0.42 | **19.88** ± 0.05 | **60.95** ± 0.12 | 12.27 ± 0.13 | **39.48** ± 0.06 | 80.95 ± 4.76 |
| | 1.0 | 112.33 ± 1.03 | 80.75 ± 1.03 | 120.89 ± 0.73 | 58.01 ± 1.1 | 28.92 ± 0.09 | 51.22 ± 0.47 | 37.56 ± 0.16 | 19.51 ± 0.21 | 59.32 ± 0.04 | **13.4** ± 0.4 | 38.28 ± 0.1 | - |
| 0.6 (OWL) | 0.0 | 76.14 ± 0.64 | 62.0 ± 0.52 | 104.62 ± 1.6 | 61.64 ± 0.17 | 30.36 ± 0.1 | 51.49 ± 0.54 | 40.32 ± 0.19 | 20.53 ± 0.3 | 60.23 ± 0.16 | 13.2 ± 0.46 | 39.68 ± 0.06 | 57.14 ± 8.25 |
| | 0.1 | 74.55 ± 0.73 | 60.11 ± 0.58 | 101.96 ± 0.17 | 61.72 ± 0.09 | 30.42 ± 0.11 | 52.25 ± 0.16 | 40.7 ± 0.23 | 20.34 ± 0.31 | 60.5 ± 0.03 | 13.6 ± 0.31 | 39.93 ± 0.13 | 61.9 ± 9.52 |
| | 0.25 | **73.97** ± 0.19 | 58.81 ± 0.28 | 100.07 ± 1.96 | **61.81** ± 0.08 | 30.47 ± 0.07 | 51.51 ± 0.37 | 40.92 ± 0.22 | 20.71 ± 0.15 | 60.36 ± 0.19 | 13.27 ± 0.35 | 39.86 ± 0.15 | 66.67 ± 4.76 |
| | 0.5 | 74.58 ± 0.49 | 58.75 ± 0.37 | 97.11 ± 0.77 | 61.72 ± 0.04 | 30.58 ± 0.11 | 52.01 ± 0.12 | 40.67 ± 0.28 | 20.31 ± 0.09 | 60.39 ± 0.08 | 13.8 ± 0.42 | 39.93 ± 0.13 | 52.38 ± 9.52 |
| | 0.75 | 74.74 ± 0.47 | **57.56** ± 0.17 | **94.12** ± 1.37 | 61.75 ± 0.16 | 30.58 ± 0.11 | 52.8 ± 0.28 | 40.81 ± 0.23 | 20.19 ± 0.17 | 60.68 ± 0.15 | 13.93 ± 0.48 | 40.11 ± 0.11 | **71.43** ± 8.25 |
| | 0.9 | 76.62 ± 0.57 | 58.53 ± 0.42 | 94.61 ± 0.92 | 61.73 ± 0.14 | **30.66** ± 0.09 | **52.99** ± 0.39 | **41.18** ± 0.25 | 20.59 ± 0.27 | **61.08** ± 0.24 | 13.87 ± 0.35 | **40.3** ± 0.19 | **71.43** ± 8.25 |
| | 1.0 | 99.38 ± 1.37 | 73.0 ± 0.37 | 111.24 ± 1.83 | 61.28 ± 0.28 | 29.88 ± 0.11 | 52.07 ± 0.82 | 40.22 ± 0.09 | **20.82** ± 0.05 | 60.03 ± 0.2 | **14.8** ± 0.12 | 39.87 ± 0.17 | - |
| 0.7 | 0.0 | 363.68 ± 3.74 | 393.51 ± 5.96 | 459.34 ± 12.4 | 38.81 ± 0.41 | 26.85 ± 0.04 | 49.14 ± 0.37 | 29.69 ± 0.24 | 18.86 ± 0.1 | 55.39 ± 0.25 | 12.4 ± 0.42 | 33.02 ± 0.07 | 52.38 ± 9.52 |
| | 0.1 | 354.91 ± 6.53 | 404.15 ± 1.97 | 438.68 ± 18.66 | 38.99 ± 0.61 | 26.9 ± 0.03 | 50.22 ± 0.62 | 29.87 ± 0.24 | 18.66 ± 0.25 | 55.6 ± 0.26 | 12.53 ± 0.07 | 33.25 ± 0.16 | 52.38 ± 17.17 |
| | 0.25 | **342.24** ± 11.1 | 370.67 ± 5.15 | 419.8 ± 37.86 | 39.01 ± 0.83 | 26.89 ± 0.09 | 50.38 ± 0.94 | 29.92 ± 0.13 | 18.57 ± 0.3 | **55.79** ± 0.24 | 12.2 ± 0.23 | 33.25 ± 0.29 | 57.14 ± 8.25 |
| | 0.5 | 356.93 ± 11.89 | 377.15 ± 1.18 | 415.21 ± 15.26 | 38.83 ± 0.59 | 26.95 ± 0.01 | 50.57 ± 0.39 | **29.99** ± 0.17 | **19.14** ± 0.08 | 55.64 ± 0.12 | 12.27 ± 0.18 | 33.34 ± 0.06 | **61.9** ± 4.76 |
| | 0.75 | 357.83 ± 11.36 | **358.07** ± 2.4 | **384.48** ± 10.82 | 38.84 ± 0.17 | **26.98** ± 0.05 | 51.62 ± 0.46 | 29.55 ± 0.13 | 18.91 ± 0.03 | 55.73 ± 0.21 | 12.53 ± 0.55 | **33.45** ± 0.1 | 57.14 ± 8.25 |
| | 0.9 | 383.24 ± 9.61 | 380.41 ± 5.38 | 420.72 ± 13.07 | 38.13 ± 0.06 | 26.93 ± 0.05 | **51.85** ± 0.34 | 29.81 ± 0.27 | 19.03 ± 0.21 | 55.46 ± 0.13 | 12.07 ± 0.52 | 33.33 ± 0.06 | **61.9** ± 4.76 |
| | 1.0 | 730.58 ± 21.78 | 777.79 ± 20.03 | 1130.38 ± 57.74 | **39.91** ± 0.83 | 26.52 ± 0.05 | 50.86 ± 0.41 | 29.32 ± 0.17 | 18.86 ± 0.32 | 55.28 ± 0.28 | **12.93** ± 0.44 | 33.38 ± 0.15 | - |
| 0.7 (Alpha-Pruning) | 0.0 | **372.76** ± 4.51 | 399.6 ± 23.33 | 448.03 ± 16.78 | 40.22 ± 0.43 | 26.76 ± 0.04 | 48.93 ± 0.32 | **29.81** ± 0.17 | 18.6 ± 0.09 | 55.28 ± 0.19 | 11.93 ± 0.18 | 33.08 ± 0.07 | 47.62 ± 4.76 |
| | 0.1 | 380.49 ± 8.11 | 410.81 ± 13.71 | 467.38 ± 22.23 | 40.04 ± 1.0 | 26.78 ± 0.06 | 49.43 ± 0.63 | 29.66 ± 0.04 | 18.77 ± 0.05 | 55.08 ± 0.07 | 11.67 ± 0.18 | 33.06 ± 0.11 | 52.38 ± 9.52 |
| | 0.25 | 378.98 ± 10.46 | 395.64 ± 3.8 | 456.96 ± 32.25 | 39.91 ± 0.87 | 26.76 ± 0.08 | 48.91 ± 0.3 | 29.67 ± 0.21 | **19.17** ± 0.25 | 55.19 ± 0.27 | 12.2 ± 0.6 | 33.12 ± 0.19 | **61.9** ± 4.76 |
| | 0.5 | 388.09 ± 11.97 | 405.85 ± 6.13 | 426.49 ± 25.19 | 39.11 ± 0.19 | 26.8 ± 0.07 | 49.51 ± 0.5 | 29.71 ± 0.21 | 18.63 ± 0.06 | 55.15 ± 0.1 | 11.87 ± 0.27 | 32.97 ± 0.08 | 52.38 ± 9.52 |
| | 0.75 | 396.58 ± 6.47 | 375.31 ± 10.65 | 403.93 ± 21.1 | 38.87 ± 0.49 | **26.88** ± 0.05 | 49.72 ± 0.23 | 29.49 ± 0.19 | 18.94 ± 0.23 | **55.4** ± 0.17 | 11.0 ± 0.12 | 32.9 ± 0.13 | 52.38 ± 4.76 |
| | 0.9 | 393.62 ± 8.05 | **367.34** ± 19.52 | **390.17** ± 15.0 | 39.99 ± 0.42 | 26.83 ± 0.02 | 49.57 ± 0.16 | 29.73 ± 0.16 | 18.66 ± 0.08 | 55.26 ± 0.32 | 11.47 ± 0.57 | 33.07 ± 0.16 | 57.14 ± 14.29 |
| | 1.0 | 939.78 ± 74.72 | 1333.27 ± 241.11 | 1873.76 ± 464.53 | **54.71** ± 1.38 | 26.4 ± 0.02 | **50.46** ± 0.41 | 28.1 ± 0.38 | 18.83 ± 0.21 | 54.41 ± 0.05 | **13.0** ± 0.35 | **35.13** ± 0.29 | - |
| 0.7 (OWL) | 0.0 | 357.81 ± 8.65 | 428.9 ± 9.65 | 561.51 ± 34.46 | 40.73 ± 0.95 | 26.92 ± 0.05 | 49.7 ± 0.41 | 30.13 ± 0.07 | 18.34 ± 0.32 | 55.3 ± 0.16 | **13.6** ± 0.5 | 33.53 ± 0.2 | **57.14** ± 8.25 |
| | 0.1 | 352.35 ± 4.06 | 405.89 ± 10.17 | 528.95 ± 24.52 | 40.31 ± 1.09 | 26.95 ± 0.06 | 50.04 ± 0.45 | 30.18 ± 0.08 | 18.34 ± 0.15 | 54.88 ± 0.28 | 13.33 ± 0.35 | 33.43 ± 0.19 | 52.38 ± 4.76 |
| | 0.25 | 344.15 ± 8.77 | 393.93 ± 20.15 | 525.98 ± 16.66 | 39.83 ± 0.56 | 26.98 ± 0.07 | 49.25 ± 0.39 | 30.15 ± 0.11 | 18.46 ± 0.06 | 54.99 ± 0.3 | 13.4 ± 0.46 | 33.29 ± 0.12 | 47.62 ± 4.76 |
| | 0.5 | 343.72 ± 11.99 | 376.14 ± 19.99 | 512.21 ± 17.46 | 41.07 ± 0.16 | 27.04 ± 0.05 | 49.51 ± 0.35 | **30.7** ± 0.12 | 18.09 ± 0.15 | 55.1 ± 0.2 | 13.53 ± 0.64 | 33.58 ± 0.06 | **57.14** ± 8.25 |
| | 0.75 | **339.11** ± 4.0 | **353.82** ± 7.97 | **485.37** ± 9.25 | 41.44 ± 0.83 | **27.05** ± 0.01 | 49.7 ± 0.72 | 30.13 ± 0.09 | 18.03 ± 0.15 | **55.37** ± 0.1 | 12.67 ± 0.37 | 33.48 ± 0.13 | 52.38 ± 12.6 |
| | 0.9 | 347.72 ± 3.14 | 358.32 ± 10.71 | 521.4 ± 26.14 | 43.01 ± 1.7 | 27.03 ± 0.06 | 50.78 ± 0.61 | 30.37 ± 0.09 | 17.86 ± 0.21 | **55.37** ± 0.17 | 11.93 ± 0.24 | 33.76 ± 0.31 | 52.38 ± 12.6 |
| | 1.0 | 653.63 ± 22.43 | 765.73 ± 71.72 | 1164.96 ± 72.63 | **46.24** ± 0.82 | 26.69 ± 0.04 | **50.88** ± 0.44 | 28.86 ± 0.52 | **19.88** ± 0.13 | 55.06 ± 0.48 | 12.33 ± 0.24 | **34.28** ± 0.24 | - |
| 2:4 | 0.0 | 113.94 ± 2.42 | 80.67 ± 2.12 | **125.89** ± 4.63 | 61.37 ± 0.26 | 28.36 ± 0.06 | 50.51 ± 0.36 | 37.88 ± 0.42 | 18.86 ± 0.34 | 59.18 ± 0.25 | 12.27 ± 0.35 | 38.35 ± 0.09 | 61.9 ± 9.52 |
| | 0.1 | 113.39 ± 2.38 | 80.47 ± 1.4 | 127.79 ± 3.53 | **61.59** ± 0.23 | 28.47 ± 0.1 | 50.57 ± 0.5 | 37.89 ± 0.17 | 18.89 ± 0.27 | 58.9 ± 0.16 | 11.93 ± 0.77 | 38.32 ± 0.11 | 76.19 ± 4.76 |
| | 0.25 | 111.95 ± 2.69 | 80.06 ± 1.98 | 128.38 ± 3.09 | 61.14 ± 0.51 | 28.59 ± 0.02 | 50.75 ± 0.2 | 38.06 ± 0.32 | **19.51** ± 0.27 | 59.25 ± 0.13 | 12.47 ± 0.47 | 38.54 ± 0.11 | 71.43 ± 0.0 |
| | 0.5 | **110.74** ± 1.17 | **78.55** ± 1.14 | 126.91 ± 1.66 | 61.08 ± 0.57 | 28.51 ± 0.05 | 50.62 ± 0.25 | 38.41 ± 0.26 | 19.43 ± 0.08 | **59.36** ± 0.33 | 12.67 ± 0.37 | 38.58 ± 0.14 | 71.43 ± 0.0 |
| | 0.75 | 115.64 ± 1.47 | 80.31 ± 1.48 | 131.07 ± 1.05 | 60.55 ± 0.63 | 28.54 ± 0.04 | **51.67** ± 0.59 | **39.0** ± 0.09 | 19.28 ± 0.47 | 59.25 ± 0.19 | 12.33 ± 0.29 | 38.66 ± 0.07 | **80.95** ± 4.76 |
| | 0.9 | 116.92 ± 1.49 | 82.07 ± 1.16 | 130.21 ± 0.66 | 59.68 ± 1.09 | **28.62** ± 0.07 | **51.67** ± 0.07 | 38.89 ± 0.15 | 19.4 ± 0.35 | 59.05 ± 0.42 | 13.4 ± 0.12 | **38.67** ± 0.25 | **80.95** ± 4.76 |
| | 1.0 | 164.32 ± 2.37 | 114.73 ± 2.32 | 190.58 ± 1.5 | 57.28 ± 1.0 | 28.32 ± 0.03 | 51.51 ± 0.15 | 35.76 ± 0.26 | 18.34 ± 0.09 | 58.41 ± 0.35 | **13.67** ± 0.07 | 37.61 ± 0.1 | - |

Table 17: Test perplexity on C4, WikiText2 and PTB and zero-shot accuracy of Llama-3.2-3B using *MOON-SHOT*-OSSCAR. Zero-shot mean performance and win rate are computed over the 7 classification tasks. Results are averaged over 3 seeds with standard errors.

| Sparsity | λ | C4 ↓ | WikiText2 ↓ | PTB ↓ | BoolQ ↑ | HellaSwag ↑ | WinoGrande ↑ | ARC-e ↑ | ARC-c ↑ | PIQA ↑ | OBQA ↑ | Mean ↑ | Win Rate ↑ |
|---|---|---|---|---|---|---|---|---|---|---|---|---|---|
| Dense | - | 11.34 | 7.81 | 13.54 | 72.72 | 55.30 | 69.22 | 74.37 | 42.41 | 76.71 | 31.20 | 60.28 | - |
| | 0.0 | 19.73 ± 0.91 | 15.94 ± 0.42 | 51.85 ± 5.02 | 60.2 ± 2.06 | 45.85 ± 0.78 | 60.59 ± 0.97 | 64.8 ± 1.09 | 31.85 ± 0.62 | 73.94 ± 0.67 | 25.0 ± 0.42 | 51.75 ± 0.7 | 9.52 ± 4.76 |
| | 0.1 | 18.86 ± 1.07 | 14.93 ± 0.6 | 50.43 ± 9.15 | **65.4** ± 1.45 | **51.07** ± 0.51 | **62.4** ± 1.71 | **67.86** ± 0.62 | 34.7 ± 0.53 | 74.86 ± 0.37 | 24.13 ± 1.43 | **54.35** ± 0.84 | 57.14 ± 21.82 |
| | 0.25 | 18.5 ± 0.24 | 13.47 ± 0.73 | 27.07 ± 0.13 | 64.37 ± 0.98 | 50.83 ± 0.28 | 60.77 ± 1.78 | 66.89 ± 0.4 | **35.13** ± 0.38 | 75.66 ± 0.52 | 22.33 ± 2.1 | 53.71 ± 0.74 | 61.9 ± 4.76 |
| 0.1 | 0.5 | **16.42** ± 0.22 | **12.63** ± 0.33 | 23.22 ± 0.42 | 65.06 ± 1.18 | 50.36 ± 0.3 | 60.41 ± 1.79 | 66.23 ± 0.94 | 34.67 ± 0.78 | 75.55 ± 0.67 | 25.67 ± 0.74 | 53.99 ± 0.83 | **66.67** ± 20.76 |
| | 0.75 | 16.56 ± 0.38 | 13.28 ± 0.55 | **22.1** ± 0.28 | 64.91 ± 1.0 | 49.5 ± 0.62 | 61.43 ± 0.47 | 66.39 ± 0.83 | 34.41 ± 0.33 | 75.54 ± 0.45 | 26.8 ± 0.76 | 54.14 ± 0.2 | **66.67** ± 4.76 |
| | 0.9 | 16.57 ± 0.41 | 13.58 ± 0.8 | 22.16 ± 0.24 | 64.36 ± 1.07 | 49.21 ± 0.9 | 61.09 ± 0.4 | 66.47 ± 0.93 | 34.36 ± 0.52 | 75.5 ± 0.56 | 27.07 ± 0.7 | 54.01 ± 0.23 | 61.9 ± 4.76 |
| | 1.0 | 16.8 ± 0.46 | 13.76 ± 0.85 | 22.34 ± 0.43 | 64.38 ± 0.86 | 49.03 ± 0.86 | 61.3 ± 0.74 | 65.82 ± 0.64 | 34.1 ± 0.53 | **75.7** ± 0.78 | **27.33** ± 0.41 | 53.95 ± 0.08 | 0.0 ± 0.0 |
| | 0.0 | 25.74 ± 1.05 | 23.46 ± 0.84 | 48.04 ± 3.18 | 50.73 ± 1.87 | 38.44 ± 1.87 | 56.3 ± 0.71 | 61.53 ± 0.87 | 29.01 ± 0.67 | 72.25 ± 0.52 | 20.4 ± 1.55 | 46.95 ± 0.55 | 14.29 ± 8.25 |
| | 0.1 | 22.26 ± 1.04 | 18.81 ± 2.21 | 55.04 ± 19.76 | **64.46** ± 1.75 | **48.69** ± 0.24 | **60.69** ± 1.21 | **65.75** ± 0.59 | 32.65 ± 0.92 | 73.81 ± 0.57 | 21.87 ± 1.38 | 52.56 ± 0.85 | 76.19 ± 9.52 |
| | 0.25 | **21.5** ± 0.22 | **18.28** ± 0.41 | **27.99** ± 1.03 | 64.34 ± 2.28 | 48.37 ± 0.21 | 58.77 ± 1.44 | 65.4 ± 0.71 | **33.16** ± 0.67 | **74.77** ± 0.55 | **25.0** ± 1.15 | **52.83** ± 0.84 | **95.24** ± 4.76 |
| 0.15 | 0.5 | 22.17 ± 0.48 | 20.19 ± 1.46 | 30.22 ± 0.2 | 63.65 ± 2.03 | 47.74 ± 0.53 | 57.98 ± 1.77 | 64.41 ± 1.03 | 32.54 ± 0.69 | 74.52 ± 0.68 | 23.93 ± 1.16 | 52.11 ± 1.02 | 90.48 ± 4.76 |
| | 0.75 | 22.26 ± 0.65 | 21.1 ± 1.9 | 30.73 ± 0.23 | 63.72 ± 2.37 | 47.42 ± 0.78 | 58.43 ± 1.76 | 64.28 ± 1.04 | 31.68 ± 0.85 | 73.76 ± 0.65 | 24.33 ± 0.94 | 51.95 ± 1.07 | 85.71 ± 0.0 |
| | 0.9 | 22.32 ± 0.65 | 21.32 ± 1.79 | 31.24 ± 0.3 | 63.48 ± 2.25 | 47.22 ± 0.84 | 58.14 ± 1.6 | 63.57 ± 1.28 | 31.48 ± 0.98 | 74.16 ± 0.56 | 24.13 ± 0.85 | 51.74 ± 1.09 | 80.95 ± 9.52 |
| | 1.0 | 22.52 ± 0.58 | 21.44 ± 1.93 | 31.7 ± 0.68 | 63.39 ± 2.33 | 47.11 ± 0.78 | 57.64 ± 2.02 | 61.94 ± 1.86 | 30.94 ± 1.25 | 73.83 ± 0.68 | 23.27 ± 1.38 | 51.16 ± 1.31 | 0.0 ± 0.0 |

Table 18: Test perplexity on C4, WikiText2 and PTB and zero-shot accuracy of Llama-3.2-3B using *MOON-SHOT*-SparseGPT. Zero-shot mean performance and win rate are computed over the 7 classification tasks. Results are averaged over 3 seeds with standard errors.

| Sparsity | λ | C4 ↓ | WikiText2 ↓ | PTB ↓ | BoolQ ↑ | HellaSwag ↑ | WinoGrande ↑ | ARC-e ↑ | ARC-c ↑ | PIQA ↑ | OBQA ↑ | Mean ↑ | Win Rate ↑ |
|---|---|---|---|---|---|---|---|---|---|---|---|---|---|
| Dense | - | 11.34 | 7.81 | 13.54 | 72.72 | 55.30 | 69.22 | 74.37 | 42.41 | 76.71 | 31.20 | 60.28 | - |
| 0.5 | 0.0 | 18.12 ± 0.06 | 13.0 ± 0.11 | 21.83 ± 0.18 | 69.19 ± 1.07 | 45.96 ± 0.21 | 63.67 ± 0.8 | 61.52 ± 1.45 | 30.69 ± 0.93 | 73.38 ± 0.37 | 24.27 ± 0.66 | 52.67 ± 0.43 | 9.52 ± 4.76 |
| | 0.1 | 16.72 ± 0.04 | 11.93 ± 0.06 | 19.62 ± 0.17 | 70.62 ± 0.65 | 47.2 ± 0.05 | 64.69 ± 0.39 | 64.72 ± 0.69 | 32.51 ± 0.27 | **73.72** ± 0.25 | 25.8 ± 0.31 | 54.18 ± 0.18 | 33.33 ± 12.6 |
| | 0.25 | 16.63 ± 0.02 | 11.86 ± 0.06 | 19.49 ± 0.19 | 70.59 ± 0.59 | 47.36 ± 0.08 | 64.09 ± 0.3 | 64.86 ± 0.89 | 32.51 ± 0.67 | 73.16 ± 0.19 | 26.0 ± 0.12 | 54.08 ± 0.21 | 33.33 ± 4.76 |
| | 0.5 | 16.54 ± 0.02 | 11.83 ± 0.06 | 19.34 ± 0.23 | 71.12 ± 0.63 | 47.47 ± 0.05 | 64.56 ± 0.16 | 65.22 ± 0.63 | 32.65 ± 0.28 | 73.27 ± 0.28 | 25.73 ± 0.37 | 54.29 ± 0.17 | 33.33 ± 4.76 |
| | 0.75 | 16.49 ± 0.04 | 11.78 ± 0.06 | 19.18 ± 0.19 | 71.0 ± 0.74 | 47.64 ± 0.05 | 64.33 ± 0.41 | **65.4** ± 0.69 | 32.94 ± 0.39 | 73.3 ± 0.36 | 26.13 ± 0.24 | 54.39 ± 0.28 | **47.62** ± 4.76 |
| | 0.9 | **16.45** ± 0.02 | **11.76** ± 0.05 | **19.05** ± 0.11 | 71.15 ± 0.76 | **47.65** ± 0.08 | 65.09 ± 0.23 | 65.01 ± 0.88 | 33.3 ± 0.57 | 73.36 ± 0.18 | 26.27 ± 0.37 | 54.55 ± 0.28 | **47.62** ± 4.76 |
| | 1.0 | 17.61 ± 0.08 | 12.58 ± 0.06 | 20.34 ± 0.13 | **72.41** ± 0.77 | 47.32 ± 0.1 | **65.93** ± 0.25 | 64.27 ± 0.53 | **33.36** ± 0.37 | 72.98 ± 0.31 | **26.8** ± 0.31 | **54.72** ± 0.24 | - |
| 0.5 (Alpha-Pruning) | 0.0 | 18.22 ± 0.02 | 13.07 ± 0.1 | 21.67 ± 0.21 | 68.6 ± 0.75 | 46.08 ± 0.12 | 63.98 ± 0.82 | 63.45 ± 0.22 | 30.94 ± 0.57 | 72.96 ± 0.23 | 23.87 ± 1.05 | 52.84 ± 0.28 | 4.76 ± 4.76 |
| | 0.1 | 16.8 ± 0.03 | 11.99 ± 0.08 | 19.59 ± 0.26 | 71.05 ± 0.34 | 47.37 ± 0.05 | 65.69 ± 0.15 | 65.7 ± 0.18 | 32.82 ± 0.1 | 73.16 ± 0.16 | 25.47 ± 0.64 | 54.47 ± 0.14 | 42.86 ± 8.25 |
| | 0.25 | 16.72 ± 0.02 | 11.91 ± 0.07 | 19.47 ± 0.26 | 71.35 ± 0.56 | 47.51 ± 0.05 | 65.25 ± 0.37 | 65.21 ± 0.22 | 32.99 ± 0.28 | 73.2 ± 0.42 | 25.27 ± 0.18 | 54.39 ± 0.11 | 42.86 ± 0.0 |
| | 0.5 | 16.66 ± 0.03 | 11.86 ± 0.06 | 19.36 ± 0.24 | 71.85 ± 0.4 | 47.69 ± 0.03 | 65.17 ± 0.27 | 65.47 ± 0.33 | 32.59 ± 0.81 | 73.49 ± 0.41 | 25.13 ± 0.29 | 54.48 ± 0.14 | 42.86 ± 0.0 |
| | 0.75 | 16.6 ± 0.03 | 11.83 ± 0.06 | 19.26 ± 0.24 | 71.64 ± 0.43 | 47.73 ± 0.06 | 65.22 ± 0.64 | 65.77 ± 0.16 | 33.02 ± 0.58 | **73.7** ± 0.49 | 26.07 ± 0.24 | 54.74 ± 0.18 | 42.86 ± 0.0 |
| | 0.9 | **16.56** ± 0.03 | **11.79** ± 0.05 | **19.01** ± 0.24 | 71.96 ± 0.28 | **47.85** ± 0.03 | 65.19 ± 0.66 | **66.12** ± 0.18 | 33.42 ± 0.86 | 73.45 ± 0.33 | 26.27 ± 0.07 | 54.89 ± 0.18 | **52.38** ± 4.76 |
| | 1.0 | 17.65 ± 0.06 | 12.59 ± 0.05 | 20.31 ± 0.17 | **72.93** ± 0.22 | 47.31 ± 0.04 | **66.09** ± 0.65 | 64.63 ± 0.28 | **34.24** ± 0.55 | 72.98 ± 0.24 | **26.8** ± 0.3 | **55.0** ± 0.18 | - |
| 0.5 (OWL) | 0.0 | 17.62 ± 0.02 | 12.92 ± 0.09 | 21.0 ± 0.15 | 69.13 ± 0.85 | 46.64 ± 0.12 | 64.88 ± 0.83 | 62.63 ± 0.49 | 31.11 ± 0.41 | 72.89 ± 0.33 | 24.0 ± 0.2 | 53.04 ± 0.1 | 9.52 ± 4.76 |
| | 0.1 | 16.54 ± 0.02 | 11.99 ± 0.06 | 19.46 ± 0.24 | 70.65 ± 0.61 | 47.82 ± 0.1 | 65.93 ± 0.42 | 66.13 ± 0.74 | 33.16 ± 0.37 | 72.98 ± 0.13 | 24.8 ± 0.31 | 54.5 ± 0.08 | 38.1 ± 12.6 |
| | 0.25 | 16.45 ± 0.01 | 11.95 ± 0.07 | 19.08 ± 0.21 | 70.83 ± 0.07 | 48.01 ± 0.04 | 66.04 ± 0.37 | **66.61** ± 0.32 | 33.42 ± 0.28 | 73.07 ± 0.41 | 25.27 ± 0.24 | 54.75 ± 0.08 | 42.86 ± 8.25 |
| | 0.5 | 16.4 ± 0.0 | 11.89 ± 0.06 | 19.06 ± 0.3 | 71.04 ± 0.1 | 48.11 ± 0.05 | 65.77 ± 0.66 | 66.32 ± 0.46 | 32.94 ± 0.3 | 73.25 ± 0.2 | 24.93 ± 0.37 | 54.62 ± 0.15 | 42.86 ± 8.25 |
| | 0.75 | 16.36 ± 0.02 | 11.86 ± 0.06 | 18.96 ± 0.26 | 71.3 ± 0.18 | 48.15 ± 0.08 | 66.14 ± 0.36 | 66.41 ± 0.35 | 33.02 ± 0.6 | 73.25 ± 0.18 | 25.13 ± 0.18 | 54.77 ± 0.12 | **52.38** ± 4.76 |
| | 0.9 | **16.33** ± 0.02 | **11.83** ± 0.07 | **18.85** ± 0.29 | 71.49 ± 0.2 | **48.21** ± 0.06 | 66.32 ± 0.07 | 65.99 ± 0.56 | 33.59 ± 0.51 | **73.39** ± 0.19 | 25.27 ± 0.18 | 54.9 ± 0.11 | 42.86 ± 8.25 |
| | 1.0 | 17.14 ± 0.06 | 12.35 ± 0.03 | 19.81 ± 0.25 | **72.73** ± 0.17 | 47.86 ± 0.08 | **66.4** ± 0.17 | 65.25 ± 0.24 | **34.36** ± 0.77 | 72.69 ± 0.09 | **26.2** ± 0.31 | **55.07** ± 0.16 | - |
| 0.6 | 0.0 | 40.78 ± 0.89 | 33.01 ± 1.0 | 58.58 ± 2.31 | 62.36 ± 1.25 | 35.6 ± 0.06 | 56.54 ± 0.47 | 49.93 ± 0.51 | 23.78 ± 0.58 | 66.87 ± 0.17 | 16.33 ± 0.66 | 44.49 ± 0.25 | 0.0 ± 0.0 |
| | 0.1 | 30.2 ± 0.09 | 23.74 ± 0.25 | 39.7 ± 1.07 | 66.47 ± 0.92 | 38.47 ± 0.06 | 60.43 ± 0.54 | 56.5 ± 0.39 | 26.96 ± 0.13 | 68.68 ± 0.13 | 19.93 ± 0.71 | 48.21 ± 0.13 | 80.95 ± 4.76 |
| | 0.25 | 29.21 ± 0.22 | 22.9 ± 0.23 | 37.44 ± 0.78 | 67.42 ± 0.06 | 38.88 ± 0.0 | 59.72 ± 0.4 | 56.66 ± 0.58 | 26.71 ± 0.21 | 69.24 ± 0.08 | 19.93 ± 0.77 | 48.37 ± 0.03 | 76.19 ± 4.76 |
| | 0.5 | 28.76 ± 0.13 | 22.81 ± 0.17 | 36.8 ± 0.62 | 67.67 ± 0.3 | 38.82 ± 0.1 | 60.38 ± 0.57 | 57.55 ± 0.86 | 27.1 ± 0.44 | 68.99 ± 0.25 | 19.47 ± 0.85 | 48.57 ± 0.1 | 85.71 ± 0.0 |
| | 0.75 | 28.57 ± 0.18 | 22.65 ± 0.21 | 36.17 ± 0.5 | 67.6 ± 0.13 | 38.96 ± 0.14 | **61.27** ± 0.19 | 57.37 ± 0.83 | 27.39 ± 0.31 | 69.04 ± 0.57 | 19.73 ± 0.82 | 48.77 ± 0.12 | 90.48 ± 4.76 |
| | 0.9 | **28.23** ± 0.11 | **22.46** ± 0.17 | **35.63** ± 0.68 | **67.76** ± 0.35 | **39.13** ± 0.07 | 61.01 ± 0.23 | **57.59** ± 0.92 | **27.79** ± 0.71 | **69.44** ± 0.13 | **20.0** ± 0.53 | **48.96** ± 0.12 | **95.24** ± 4.76 |
| | 1.0 | 33.63 ± 0.14 | 26.12 ± 0.23 | 42.69 ± 0.73 | 66.82 ± 0.6 | 38.14 ± 0.14 | 60.91 ± 0.65 | 53.89 ± 0.11 | 26.28 ± 0.18 | 67.75 ± 0.34 | 18.47 ± 0.44 | 47.47 ± 0.08 | - |
| 0.6 (Alpha-Pruning) | 0.0 | 39.98 ± 0.8 | 32.14 ± 1.25 | 57.63 ± 0.55 | 64.43 ± 0.69 | 36.11 ± 0.05 | 57.27 ± 0.35 | 49.96 ± 0.18 | 23.83 ± 0.9 | 66.7 ± 0.39 | 17.47 ± 0.35 | 45.11 ± 0.14 | 0.0 ± 0.0 |
| | 0.1 | 30.39 ± 0.11 | 23.23 ± 0.16 | 37.83 ± 1.2 | 67.9 ± 0.49 | 38.53 ± 0.03 | 61.51 ± 0.27 | 55.19 ± 0.05 | 26.42 ± 0.29 | 68.39 ± 0.51 | 19.33 ± 0.18 | 48.18 ± 0.08 | 71.43 ± 8.25 |
| | 0.25 | 29.68 ± 0.05 | 23.01 ± 0.3 | 36.99 ± 0.82 | 68.92 ± 0.39 | 38.89 ± 0.09 | 61.72 ± 0.39 | 55.27 ± 0.27 | 26.02 ± 0.23 | 68.52 ± 0.35 | 19.07 ± 0.44 | 48.34 ± 0.08 | 80.95 ± 12.6 |
| | 0.5 | 29.14 ± 0.1 | 22.56 ± 0.07 | 36.56 ± 1.25 | 68.5 ± 0.45 | 39.19 ± 0.08 | 61.56 ± 0.37 | 56.41 ± 0.37 | **26.99** ± 0.46 | 68.92 ± 0.34 | 19.87 ± 0.24 | 48.63 ± 0.07 | 71.43 ± 14.29 |
| | 0.75 | 28.96 ± 0.15 | 22.44 ± 0.04 | 35.89 ± 1.52 | **68.93** ± 0.35 | 39.15 ± 0.14 | 61.3 ± 0.27 | 56.45 ± 0.46 | 26.82 ± 0.54 | 68.59 ± 0.14 | 20.0 ± 0.83 | 48.75 ± 0.05 | 80.95 ± 4.76 |
| | 0.9 | **28.64** ± 0.25 | **22.17** ± 0.19 | **35.48** ± 1.52 | 68.73 ± 0.11 | **39.34** ± 0.18 | **62.27** ± 0.52 | **56.52** ± 0.25 | 26.69 ± 0.29 | **69.13** ± 0.32 | **20.33** ± 0.24 | **49.04** ± 0.06 | **85.71** ± 8.25 |
| | 1.0 | 34.19 ± 0.4 | 26.06 ± 0.64 | 42.82 ± 0.13 | 68.17 ± 0.43 | 38.62 ± 0.16 | 61.98 ± 0.81 | 53.37 ± 0.7 | 26.0 ± 0.23 | 68.06 ± 0.38 | 19.8 ± 0.9 | 48.0 ± 0.38 | - |
| 0.6 (OWL) | 0.0 | 32.72 ± 0.56 | 27.69 ± 0.6 | 45.87 ± 0.76 | 64.46 ± 0.51 | 38.09 ± 0.1 | 58.01 ± 0.25 | 53.38 ± 0.31 | 25.14 ± 0.33 | 67.94 ± 0.1 | 18.0 ± 0.4 | 46.43 ± 0.14 | 0.0 ± 0.0 |
| | 0.1 | 26.66 ± 0.16 | 21.77 ± 0.33 | 33.89 ± 0.66 | 66.67 ± 0.14 | 40.01 ± 0.11 | 61.93 ± 0.66 | 56.8 ± 0.29 | 27.79 ± 0.54 | 69.17 ± 0.4 | 20.67 ± 0.18 | 49.01 ± 0.08 | 61.9 ± 4.76 |
| | 0.25 | 26.17 ± 0.12 | 21.56 ± 0.35 | 33.74 ± 0.66 | 66.99 ± 0.37 | 40.38 ± 0.07 | 61.14 ± 0.49 | 56.93 ± 0.78 | 27.9 ± 0.69 | 69.48 ± 0.45 | **21.4** ± 0.35 | 49.17 ± 0.16 | **71.43** ± 14.29 |
| | 0.5 | 25.77 ± 0.16 | 21.16 ± 0.23 | 32.36 ± 0.55 | **67.41** ± 0.49 | 40.52 ± 0.12 | 61.48 ± 0.2 | 57.15 ± 0.86 | 27.87 ± 0.72 | 69.6 ± 0.44 | 21.07 ± 0.07 | 49.31 ± 0.2 | 66.67 ± 12.6 |
| | 0.75 | 25.55 ± 0.12 | 21.0 ± 0.21 | 31.7 ± 0.41 | 67.35 ± 0.33 | 40.61 ± 0.13 | 61.48 ± 0.84 | 57.06 ± 1.27 | 27.7 ± 0.46 | **69.99** ± 0.61 | 21.2 ± 0.12 | 49.34 ± 0.19 | 66.67 ± 12.6 |
| | 0.9 | **25.31** ± 0.12 | **20.85** ± 0.11 | **31.39** ± 0.69 | **67.41** ± 0.47 | **40.64** ± 0.13 | 61.62 ± 0.09 | **57.39** ± 0.78 | **28.16** ± 0.62 | 69.73 ± 0.29 | 20.6 ± 0.5 | **49.36** ± 0.17 | **71.43** ± 8.25 |
| | 1.0 | 29.15 ± 0.31 | 23.58 ± 0.4 | 36.58 ± 0.78 | 66.64 ± 0.77 | 39.89 ± 0.16 | **61.96** ± 0.4 | 55.57 ± 0.41 | 27.53 ± 0.12 | 68.21 ± 0.46 | 21.2 ± 0.35 | 48.79 ± 0.11 | - |
| 0.7 | 0.0 | 290.72 ± 16.15 | 466.55 ± 68.13 | 728.59 ± 27.38 | 56.11 ± 0.86 | 27.29 ± 0.08 | 48.54 ± 0.09 | 32.0 ± 0.37 | 17.21 ± 0.12 | 56.82 ± 0.2 | 11.13 ± 0.41 | 35.58 ± 0.2 | 0.0 ± 0.0 |
| | 0.1 | 118.35 ± 3.46 | 118.46 ± 4.92 | 217.69 ± 6.26 | 61.44 ± 0.36 | 28.87 ± 0.05 | 50.25 ± 0.32 | 36.0 ± 0.48 | 17.78 ± 0.15 | 59.05 ± 0.31 | 13.0 ± 0.42 | 38.05 ± 0.07 | 57.14 ± 16.5 |
| | 0.25 | 109.6 ± 3.01 | 108.4 ± 6.08 | 197.45 ± 3.95 | 61.56 ± 0.57 | 29.1 ± 0.15 | 50.8 ± 0.59 | 36.24 ± 0.36 | 17.46 ± 0.28 | 59.29 ± 0.23 | 12.4 ± 0.58 | 38.07 ± 0.18 | 71.43 ± 8.25 |
| | 0.5 | 104.09 ± 3.16 | 101.36 ± 4.66 | 182.24 ± 11.37 | 61.99 ± 0.32 | 29.32 ± 0.14 | **51.62** ± 0.36 | 36.63 ± 0.43 | 17.61 ± 0.27 | 59.63 ± 0.21 | **13.13** ± 0.67 | **38.56** ± 0.1 | **76.19** ± 4.76 |
| | 0.75 | 101.67 ± 3.13 | 98.35 ± 3.75 | 167.03 ± 3.36 | 61.54 ± 0.66 | 29.42 ± 0.16 | 50.3 ± 0.3 | 37.14 ± 0.55 | 17.97 ± 0.13 | 59.38 ± 0.16 | 12.27 ± 0.64 | 38.34 ± 0.24 | 71.43 ± 8.25 |
| | 0.9 | **99.51** ± 4.73 | **93.39** ± 5.34 | **158.49** ± 11.74 | 61.58 ± 0.22 | **29.51** ± 0.18 | 51.17 ± 0.27 | **37.22** ± 0.59 | **18.15** ± 0.19 | **59.74** ± 0.22 | 12.4 ± 0.53 | 38.54 ± 0.16 | 71.43 ± 0.0 |
| | 1.0 | 126.28 ± 1.69 | 132.44 ± 6.04 | 197.02 ± 7.93 | **62.16** ± 0.04 | 28.81 ± 0.14 | 50.3 ± 0.3 | 34.05 ± 0.37 | 17.75 ± 0.26 | 58.87 ± 0.47 | 12.47 ± 0.18 | 37.77 ± 0.13 | - |
| 0.7 (Alpha-Pruning) | 0.0 | 261.75 ± 10.92 | 340.21 ± 40.8 | 477.68 ± 7.48 | 56.3 ± 3.25 | 27.04 ± 0.09 | 49.3 ± 0.62 | 31.06 ± 0.3 | 17.06 ± 0.23 | 56.87 ± 0.36 | 12.07 ± 0.57 | 35.67 ± 0.48 | 14.29 ± 8.25 |
| | 0.1 | 119.79 ± 3.08 | 118.4 ± 5.46 | 195.98 ± 2.28 | 61.6 ± 0.28 | 29.07 ± 0.09 | 50.07 ± 0.42 | 35.7 ± 0.66 | 17.63 ± 0.31 | 58.83 ± 0.07 | 12.27 ± 0.93 | 37.88 ± 0.3 | 52.38 ± 23.81 |
| | 0.25 | 113.92 ± 1.83 | 110.78 ± 5.71 | 183.24 ± 2.56 | 62.12 ± 0.11 | 29.28 ± 0.05 | 50.8 ± 0.59 | 36.17 ± 0.89 | 18.15 ± 0.28 | 58.85 ± 0.11 | 12.67 ± 0.52 | 38.29 ± 0.26 | 66.67 ± 17.17 |
| | 0.5 | 110.54 ± 1.35 | 104.32 ± 3.38 | 169.78 ± 6.71 | 61.83 ± 0.14 | 29.43 ± 0.07 | 50.62 ± 0.73 | 36.46 ± 0.39 | 17.75 ± 0.13 | 59.01 ± 0.29 | 12.2 ± 0.64 | 38.19 ± 0.29 | 57.14 ± 14.29 |
| | 0.75 | **104.33** ± 1.89 | **99.31** ± 4.75 | **160.87** ± 5.87 | 62.29 ± 0.38 | 29.53 ± 0.22 | 51.33 ± 0.27 | **37.3** ± 0.52 | 18.03 ± 0.47 | 59.14 ± 0.6 | 12.67 ± 0.98 | **38.61** ± 0.34 | **85.71** ± 8.25 |
| | 0.9 | 104.48 ± 2.34 | 99.49 ± 4.65 | 164.97 ± 0.97 | **62.42** ± 0.41 | **29.55** ± 0.13 | **51.35** ± 0.23 | 36.67 ± 0.39 | 17.97 ± 0.37 | **59.32** ± 0.37 | **12.87** ± 0.33 | 38.59 ± 0.15 | 85.71 ± 8.25 |
| | 1.0 | 127.0 ± 3.62 | 130.27 ± 5.8 | 192.86 ± 7.65 | 62.01 ± 0.16 | 28.88 ± 0.07 | 50.75 ± 0.42 | 34.03 ± 0.24 | **18.26** ± 0.09 | 57.94 ± 0.6 | 12.67 ± 0.33 | 37.79 ± 0.11 | - |
| 0.7 (OWL) | 0.0 | 180.86 ± 5.16 | 245.13 ± 34.76 | 411.51 ± 26.3 | 60.96 ± 1.0 | 28.07 ± 0.12 | 49.72 ± 0.78 | 33.54 ± 0.69 | 17.29 ± 0.43 | 58.81 ± 0.25 | 10.93 ± 0.27 | 37.05 ± 0.29 | 4.76 ± 4.76 |
| | 0.1 | 92.71 ± 1.59 | 95.0 ± 5.23 | 177.06 ± 7.1 | 62.38 ± 0.26 | 30.19 ± 0.14 | 52.49 ± 0.5 | 38.01 ± 0.68 | 18.26 ± 0.34 | 60.14 ± 0.43 | 13.07 ± 0.47 | 39.22 ± 0.2 | 57.14 ± 8.25 |
| | 0.25 | 82.1 ± 0.76 | 82.37 ± 2.67 | 154.35 ± 5.83 | **62.92** ± 0.55 | 30.53 ± 0.18 | 52.28 ± 0.39 | 38.86 ± 0.47 | 19.25 ± 0.65 | 60.54 ± 0.16 | 12.4 ± 0.42 | 39.54 ± 0.17 | 66.67 ± 19.05 |
| | 0.5 | 81.71 ± 1.61 | 82.28 ± 2.45 | 139.98 ± 5.0 | 62.73 ± 0.01 | 30.63 ± 0.2 | **52.83** ± 0.34 | 38.78 ± 0.2 | 19.28 ± 0.36 | 60.72 ± 0.06 | 13.27 ± 0.35 | 39.75 ± 0.05 | **95.24** ± 4.76 |
| | 0.75 | 79.17 ± 1.96 | 78.8 ± 2.88 | **131.69** ± 10.61 | 62.72 ± 0.23 | 30.76 ± 0.14 | 52.43 ± 0.33 | 39.34 ± 0.27 | 19.0 ± 0.27 | 61.01 ± 0.15 | 12.47 ± 0.13 | 39.68 ± 0.06 | 85.71 ± 8.25 |
| | 0.9 | **77.2** ± 2.57 | **73.75** ± 4.0 | 131.93 ± 7.78 | 62.71 ± 0.25 | **30.8** ± 0.2 | 52.59 ± 0.35 | **39.35** ± 0.63 | **19.62** ± 0.37 | **61.28** ± 0.02 | 12.6 ± 0.6 | **39.85** ± 0.19 | 85.71 ± 0.0 |
| | 1.0 | 100.8 ± 0.6 | 105.99 ± 3.33 | 175.39 ± 2.06 | 62.3 ± 0.13 | 30.22 ± 0.12 | 51.67 ± 0.58 | 36.5 ± 0.69 | 18.46 ± 0.11 | 60.12 ± 0.14 | **13.47** ± 0.52 | 38.96 ± 0.17 | - |
| 2:4 | 0.0 | 37.73 ± 0.41 | 31.27 ± 0.32 | 52.93 ± 2.08 | 62.99 ± 0.16 | 35.06 ± 0.11 | 55.09 ± 0.6 | 52.24 ± 0.49 | 24.32 ± 0.6 | 67.03 ± 0.49 | 17.4 ± 0.7 | 44.88 ± 0.3 | 0.0 ± 0.0 |
| | 0.1 | 30.25 ± 0.14 | 24.33 ± 0.24 | 37.75 ± 0.96 | 65.71 ± 0.15 | 37.51 ± 0.21 | 58.22 ± 0.8 | 55.78 ± 0.36 | 26.05 ± 0.16 | 68.01 ± 0.27 | 19.6 ± 0.5 | 47.27 ± 0.16 | 33.33 ± 4.76 |
| | 0.25 | 29.61 ± 0.37 | 23.87 ± 0.43 | 37.21 ± 1.24 | 65.22 ± 0.57 | 37.72 ± 0.13 | 57.77 ± 1.01 | 55.99 ± 0.16 | 26.39 ± 0.28 | 67.75 ± 0.15 | 19.87 ± 0.24 | 47.25 ± 0.2 | 23.81 ± 9.52 |
| | 0.5 | 29.17 ± 0.15 | 23.48 ± 0.29 | 36.88 ± 0.72 | 65.03 ± 0.34 | 37.99 ± 0.12 | 57.98 ± 0.72 | 55.96 ± 0.34 | 26.17 ± 0.16 | 68.14 ± 0.15 | 20.73 ± 0.58 | 47.43 ± 0.13 | 28.57 ± 0.0 |
| | 0.75 | **28.79** ± 0.29 | 23.21 ± 0.32 | 35.94 ± 0.73 | 65.58 ± 0.08 | 38.06 ± 0.13 | 59.3 ± 0.41 | 55.99 ± 0.63 | 26.56 ± 0.37 | 67.94 ± 0.21 | 20.93 ± 0.55 | 47.77 ± 0.18 | **47.62** ± 9.52 |
| | 0.9 | 28.83 ± 0.12 | **23.2** ± 0.16 | **35.46** ± 0.46 | **65.82** ± 0.39 | 38.14 ± 0.14 | 58.56 ± 0.48 | **56.65** ± 0.45 | **26.73** ± 0.08 | 68.01 ± 0.28 | **21.33** ± 0.35 | **47.89** ± 0.09 | 42.86 ± 8.25 |
| | 1.0 | 30.0 ± 0.29 | 24.4 ± 0.23 | 38.32 ± 0.74 | 65.64 ± 0.7 | **38.31** ± 0.08 | **59.93** ± 0.13 | 55.63 ± 1.1 | 26.14 ± 0.42 | **68.34** ± 0.33 | 20.87 ± 0.35 | 47.83 ± 0.18 | - |

Table 19: Test perplexity on C4, WikiText2 and PTB and zero-shot accuracy of Llama-3.2-3B using *MOON-SHOT*-Wanda. Zero-shot mean performance and win rate are computed over the 7 classification tasks. Results are averaged over 3 seeds with standard errors.

| Sparsity | λ | C4 ↓ | WikiText2 ↓ | PTB ↓ | BoolQ ↑ | HellaSwag ↑ | WinoGrande ↑ | ARC-e ↑ | ARC-c ↑ | PIQA ↑ | OBQA ↑ | Mean ↑ | Win Rate ↑ |
|---|---|---|---|---|---|---|---|---|---|---|---|---|---|
| Dense | - | 11.34 | 7.81 | 13.54 | 72.72 | 55.30 | 69.22 | 74.37 | 42.41 | 76.71 | 31.20 | 60.28 | - |
| 0.5 | 0.0 | 18.2±0.04 | 12.52±0.0 | 21.25±0.03 | 64.89±0.39 | 45.57±0.04 | 63.04±0.41 | 65.8±0.33 | 32.71±0.57 | 73.25±0.15 | 25.53±0.52 | 52.97±0.19 | 42.86±14.29 |
| | 0.1 | 18.16±0.03 | 12.48±0.0 | 21.17±0.05 | 65.24±0.16 | 45.6±0.07 | 62.9±0.16 | 65.98±0.27 | 32.57±0.5 | 73.47±0.1 | 25.53±0.41 | 53.04±0.21 | 52.38±12.6 |
| | 0.25 | 18.08±0.03 | 12.42±0.01 | 21.04±0.07 | 64.22±0.47 | 45.61±0.07 | 62.75±0.61 | 65.77±0.35 | 32.48±0.44 | 73.36±0.35 | 25.6±0.35 | 52.83±0.09 | 52.38±12.6 |
| | 0.5 | 18.0±0.04 | 12.36±0.01 | 20.87±0.04 | 64.98±0.31 | 45.73±0.03 | **63.11±0.3** | 65.71±0.59 | 32.11±0.16 | 73.36±0.15 | 25.67±0.07 | 52.95±0.1 | 52.38±12.6 |
| | 0.75 | **17.99±0.04** | **12.35±0.02** | 20.76±0.05 | 64.59±0.63 | **45.84±0.04** | 62.56±0.42 | 66.16±0.35 | 32.65±0.15 | **73.61±0.21** | 25.67±0.66 | 53.01±0.15 | 61.9±12.6 |
| | 0.9 | 18.02±0.02 | 12.37±0.02 | **20.74±0.05** | 63.91±0.83 | 45.81±0.08 | 62.64±0.41 | **66.36±0.34** | **32.99±0.34** | 73.32±0.07 | 25.6±0.42 | 52.95±0.04 | **66.67±4.76** |
| | 1.0 | 18.88±0.03 | 13.0±0.01 | 21.73±0.07 | **66.41±0.39** | 45.64±0.08 | 62.96±0.14 | 65.84±0.17 | 32.68±0.27 | 72.67±0.17 | **25.8±0.61** | **53.14±0.08** | - |
| 0.5 (Alpha-Pruning) | 0.0 | 18.19±0.03 | 12.48±0.02 | 21.31±0.03 | 66.27±0.68 | 45.69±0.08 | 63.67±0.25 | **66.89±0.3** | 33.42±0.41 | 73.45±0.2 | 25.33±0.77 | 53.53±0.2 | 52.38±17.17 |
| | 0.1 | 18.14±0.02 | 12.42±0.02 | 21.17±0.05 | 65.96±0.73 | 45.7±0.12 | 63.46±0.33 | 66.72±0.38 | 33.3±0.1 | 73.12±0.24 | 25.47±0.68 | 53.39±0.12 | 42.86±16.5 |
| | 0.25 | 18.05±0.04 | 12.37±0.02 | 21.08±0.08 | 66.55±0.42 | 45.84±0.03 | 63.64±0.38 | 66.37±0.27 | 32.79±0.38 | 73.3±0.17 | 25.27±0.47 | 53.4±0.06 | 42.86±16.5 |
| | 0.5 | 17.97±0.03 | 12.29±0.01 | 20.87±0.01 | 66.68±0.58 | 45.9±0.08 | **63.83±0.3** | 66.71±0.14 | 33.13±0.44 | 73.38±0.08 | 25.53±0.27 | 53.59±0.12 | 52.38±9.52 |
| | 0.75 | **17.94±0.03** | **12.28±0.02** | 20.75±0.03 | 66.18±0.8 | 46.05±0.05 | 63.4±0.19 | 66.62±0.18 | 33.53±0.18 | **73.54±0.16** | **25.67±0.44** | 53.57±0.1 | **57.14±8.25** |
| | 0.9 | 17.97±0.02 | 12.3±0.04 | **20.69±0.01** | 66.25±0.73 | **46.08±0.01** | 62.83±0.41 | **66.89±0.11** | 33.5±0.16 | 73.3±0.24 | 25.6±0.35 | 53.49±0.1 | 52.38±4.76 |
| | 1.0 | 18.78±0.01 | 12.83±0.03 | 21.42±0.04 | **67.72±0.78** | 45.9±0.04 | 63.38±0.4 | 66.69±0.1 | **34.04±0.17** | 72.65±0.13 | 25.07±0.29 | **53.64±0.19** | - |
| 0.5 (OWL) | 0.0 | 18.24±0.05 | 12.58±0.02 | 20.7±0.04 | 67.37±0.34 | 46.05±0.06 | 64.11±0.23 | 65.88±0.23 | 32.76±0.32 | 73.09±0.17 | 24.87±0.35 | 53.45±0.16 | 42.86±8.25 |
| | 0.1 | 18.2±0.04 | 12.55±0.02 | 20.69±0.06 | 67.12±0.3 | 46.09±0.01 | 63.48±0.53 | 66.13±0.23 | 32.88±0.24 | 73.16±0.13 | 24.67±0.27 | 53.36±0.17 | 33.33±4.76 |
| | 0.25 | 18.16±0.04 | 12.51±0.01 | 20.6±0.04 | 66.95±0.22 | 46.13±0.06 | 63.77±0.24 | 66.25±0.08 | 32.82±0.23 | 73.25±0.2 | **25.2±0.4** | 53.48±0.1 | 42.86±8.25 |
| | 0.5 | 18.13±0.04 | 12.49±0.01 | 20.5±0.02 | 66.37±0.44 | 46.26±0.02 | 64.14±0.23 | 65.91±0.39 | 32.99±0.27 | **73.39±0.32** | 25.0±0.35 | 53.44±0.13 | 47.62±9.52 |
| | 0.75 | **18.11±0.03** | **12.47±0.01** | 20.35±0.02 | 66.52±0.63 | 46.25±0.05 | 64.06±0.07 | 66.37±0.21 | 33.36±0.27 | 73.14±0.14 | 24.8±0.31 | 53.5±0.07 | 52.38±4.76 |
| | 0.9 | **18.11±0.03** | **12.47±0.03** | **20.31±0.02** | 66.3±0.92 | **46.39±0.04** | 63.98±0.35 | **66.4±0.22** | 33.59±0.24 | 73.14±0.1 | 24.93±0.35 | 53.53±0.17 | **52.38±4.76** |
| | 1.0 | 18.82±0.03 | 12.98±0.01 | 21.06±0.04 | **68.98±0.49** | 46.24±0.05 | **64.46±0.27** | 66.05±0.32 | **34.22±0.05** | 72.45±0.04 | 24.73±0.37 | **53.88±0.05** | - |
| 0.6 | 0.0 | 40.28±0.67 | 30.39±0.33 | 52.83±1.47 | 60.46±0.73 | 35.1±0.12 | 51.57±0.31 | 53.81±0.44 | 23.81±0.44 | 66.78±0.13 | 16.87±0.35 | 44.24±0.27 | 57.14±8.25 |
| | 0.1 | 39.37±0.65 | 29.45±0.33 | 50.79±1.44 | 60.4±0.72 | 35.14±0.13 | 54.96±0.16 | 51.77±0.23 | 23.95±0.27 | 67.01±0.09 | 17.0±0.42 | 44.32±0.21 | 57.14±8.25 |
| | 0.25 | 38.87±0.45 | 28.74±0.3 | 49.29±1.24 | 60.68±0.55 | 35.25±0.12 | 55.09±0.66 | 52.3±0.29 | 24.06±0.15 | 67.14±0.11 | 16.27±0.29 | 44.4±0.11 | 57.14±0.0 |
| | 0.5 | 38.15±0.24 | 28.04±0.14 | 47.61±0.7 | 60.45±1.05 | 35.53±0.07 | 55.17±0.09 | 52.31±0.2 | 24.69±0.22 | **67.23±0.05** | **17.07±0.24** | 44.53±0.22 | **61.9±4.76** |
| | 0.75 | 37.81±0.05 | 27.75±0.16 | 46.75±0.16 | 61.86±1.24 | **35.54±0.02** | 55.56±0.6 | **52.69±0.22** | 24.49±0.49 | 66.49±0.14 | 16.53±0.18 | 44.74±0.26 | **61.9±4.76** |
| | 0.9 | **37.73±0.19** | **27.71±0.26** | **46.47±0.08** | 61.33±0.91 | 35.53±0.08 | 54.83±0.14 | 52.53±0.29 | **24.69±0.21** | 66.81±0.03 | 16.6±0.23 | 44.62±0.12 | **61.9±4.76** |
| | 1.0 | 41.98±0.4 | 30.56±0.32 | 51.0±0.45 | **64.82±0.35** | 35.12±0.07 | **56.56±0.46** | 50.58±0.41 | 23.83±0.12 | 65.58±0.19 | 16.93±0.07 | **44.77±0.1** | - |
| 0.6 (Alpha-Pruning) | 0.0 | 39.14±0.81 | 29.06±0.27 | 52.35±1.36 | 60.69±2.09 | 35.6±0.16 | 56.49±0.52 | 53.45±0.25 | 24.43±0.08 | 66.32±0.22 | 17.0±0.5 | 44.86±0.4 | 47.62±4.76 |
| | 0.1 | 38.33±0.87 | 28.37±0.37 | 50.78±1.45 | 61.15±1.57 | 35.66±0.11 | 57.04±0.34 | 53.48±0.14 | 24.66±0.3 | 66.59±0.46 | **17.2±0.4** | 45.11±0.26 | 61.9±4.76 |
| | 0.25 | 37.79±0.85 | 27.72±0.35 | 49.1±1.42 | 61.74±2.05 | 35.76±0.08 | 57.22±0.39 | 53.82±0.41 | 24.8±0.1 | 66.67±0.35 | 17.13±0.57 | 45.31±0.35 | 57.14±0.0 |
| | 0.5 | 37.56±0.43 | 27.26±0.13 | 47.87±0.88 | 63.15±1.22 | 35.95±0.08 | **57.51±0.56** | 53.94±0.33 | 24.15±0.27 | 66.76±0.3 | 16.13±0.29 | 45.39±0.17 | 47.62±4.76 |
| | 0.75 | **37.37±0.21** | **27.08±0.16** | **47.01±0.29** | 63.38±0.65 | 36.05±0.1 | **57.51±0.56** | **54.15±0.27** | **25.43±0.18** | **66.96±0.36** | 16.27±0.18 | **45.68±0.1** | 61.9±4.76 |
| | 0.9 | 37.57±0.18 | 27.31±0.1 | 47.45±0.29 | 63.07±0.61 | **36.18±0.11** | 57.2±0.17 | 53.86±0.3 | 25.23±0.19 | 66.94±0.31 | 16.73±0.35 | 45.6±0.07 | **66.67±4.76** |
| | 1.0 | 40.03±0.04 | 29.19±0.2 | 50.24±0.27 | **65.93±0.24** | 35.95±0.01 | **57.51±0.09** | 51.09±0.16 | 24.32±0.36 | 66.03±0.15 | 16.87±0.24 | 45.39±0.03 | - |
| 0.6 (OWL) | 0.0 | 35.96±0.47 | 27.31±0.4 | 45.24±0.7 | 61.81±0.96 | 37.06±0.16 | 57.51±0.39 | 53.72±0.64 | 26.11±0.13 | 67.21±0.13 | **17.6±0.5** | 45.86±0.14 | 57.14±0.0 |
| | 0.1 | 35.4±0.33 | 26.77±0.33 | 43.81±0.77 | 61.4±1.55 | 37.16±0.2 | 57.35±0.34 | 54.0±0.67 | 25.8±0.22 | 67.27±0.15 | 17.27±0.41 | 45.75±0.25 | 47.62±4.76 |
| | 0.25 | 34.99±0.17 | 26.44±0.23 | 42.75±0.89 | 61.95±1.53 | 37.22±0.12 | 57.56±0.16 | 54.22±0.44 | 26.08±0.27 | **67.3±0.23** | 17.13±0.18 | 45.92±0.24 | 61.9±4.76 |
| | 0.5 | 34.71±0.22 | 25.94±0.15 | 40.98±0.64 | 63.11±1.86 | 37.28±0.14 | 57.91±0.43 | 54.32±0.44 | **26.31±0.03** | **67.3±0.06** | 17.2±0.23 | 46.2±0.29 | **66.67±4.76** |
| | 0.75 | **34.5±0.08** | 25.64±0.08 | **40.35±0.63** | 64.06±1.0 | **37.41±0.12** | 58.33±0.25 | **54.41±0.39** | 25.85±0.18 | 67.03±0.17 | 17.4±0.12 | 46.35±0.17 | **66.67±4.76** |
| | 0.9 | 34.56±0.11 | **25.59±0.06** | 40.41±0.78 | 63.73±0.92 | 37.4±0.12 | 58.93±0.35 | 54.31±0.07 | 25.68±0.13 | 67.05±0.25 | 17.0±0.23 | 46.3±0.13 | **66.67±9.52** |
| | 1.0 | 37.35±0.14 | 27.93±0.23 | 44.25±0.74 | **67.26±0.07** | 37.18±0.13 | **59.06±0.73** | 51.89±0.34 | 25.54±0.23 | 66.47±0.1 | 17.2±0.12 | **46.37±0.18** | - |
| 0.7 | 0.0 | 246.47±5.88 | 232.57±8.94 | 323.16±2.66 | 40.2±1.09 | 27.32±0.04 | **49.33±0.62** | 33.52±0.2 | 17.83±0.17 | 56.67±0.14 | 12.0±0.42 | 33.84±0.28 | **52.38±4.76** |
| | 0.1 | 240.18±3.7 | 224.64±6.04 | 318.96±7.59 | 40.63±0.96 | 27.39±0.05 | 48.38±0.16 | 33.33±0.21 | 17.26±0.15 | 57.02±0.19 | 12.0±0.23 | 33.72±0.21 | 42.86±0.0 |
| | 0.25 | 240.29±4.53 | 225.62±7.31 | 307.06±10.91 | 40.39±1.24 | 27.43±0.03 | 48.17±0.7 | 33.49±0.16 | 17.12±0.17 | 57.05±0.25 | 11.8±0.0 | 33.64±0.27 | 47.62±4.76 |
| | 0.5 | 247.47±3.63 | 222.78±6.1 | 310.9±14.25 | 40.31±1.26 | 27.55±0.02 | 48.51±0.88 | 33.54±0.19 | 17.12±0.12 | 57.16±0.17 | **12.13±0.13** | 33.76±0.34 | 47.62±4.76 |
| | 0.75 | 239.91±9.44 | 206.01±6.04 | 298.6±4.91 | 38.66±0.18 | 27.67±0.04 | 48.46±0.6 | **33.87±0.47** | 17.21±0.42 | 57.25±0.28 | 11.87±0.07 | 33.57±0.15 | 47.62±4.76 |
| | 0.9 | **235.73±6.87** | **195.34±4.93** | **297.54±7.2** | 38.85±0.35 | 27.73±0.02 | 47.91±0.32 | 33.54±0.25 | 17.52±0.32 | **57.56±0.08** | 12.07±0.41 | 33.6±0.16 | 47.62±4.76 |
| | 1.0 | 250.62±7.51 | 230.39±4.31 | 332.57±5.15 | **50.06±0.28** | **27.79±0.06** | 48.57±0.18 | 32.74±0.28 | **17.89±0.2** | 56.6±0.24 | 10.73±0.24 | **34.91±0.13** | - |
| 0.7 (Alpha-Pruning) | 0.0 | 229.42±11.77 | 202.94±9.43 | 282.4±14.97 | 49.63±2.65 | 27.27±0.03 | 48.46±0.48 | 32.49±0.47 | 17.72±0.3 | 57.04±0.02 | 12.27±0.13 | 34.98±0.42 | 38.1±12.6 |
| | 0.1 | 222.38±9.21 | 190.27±6.69 | 261.37±14.19 | 50.94±2.29 | 27.29±0.04 | **49.57±0.55** | 32.72±0.42 | **17.78±0.23** | 57.29±0.34 | 12.27±0.18 | 35.41±0.34 | 42.86±14.29 |
| | 0.25 | 216.49±6.86 | 183.02±8.68 | 250.18±20.32 | 50.7±1.28 | 27.31±0.1 | 48.43±0.25 | **33.07±0.36** | 17.61±0.06 | 57.09±0.29 | 11.87±0.33 | 35.15±0.13 | 38.1±4.76 |
| | 0.5 | 215.37±6.76 | 174.76±7.2 | **243.8±8.94** | 47.46±1.4 | 27.5±0.08 | 48.49±0.77 | 33.02±0.19 | 17.09±0.16 | 57.45±0.22 | 11.93±0.35 | 34.71±0.21 | 38.1±4.76 |
| | 0.75 | 214.23±5.13 | 169.94±7.34 | 253.34±17.49 | 48.66±2.8 | 27.62±0.06 | 47.62±0.33 | 32.84±0.11 | 17.18±0.08 | 57.69±0.12 | 12.0±0.2 | 34.8±0.39 | 42.86±0.0 |
| | 0.9 | **206.62±3.64** | **158.5±1.88** | 259.24±10.32 | 50.83±2.47 | **27.78±0.03** | 47.57±0.57 | 32.66±0.23 | 17.21±0.23 | **57.78±0.05** | 11.67±0.24 | 35.07±0.27 | **52.38±4.76** |
| | 1.0 | 211.14±2.72 | 181.22±1.14 | 274.72±9.43 | **58.26±1.0** | 27.68±0.02 | 47.96±0.42 | 32.24±0.18 | 17.61±0.12 | 57.39±0.12 | **12.53±0.13** | **36.21±0.14** | - |
| 0.7 (OWL) | 0.0 | 216.6±5.92 | 208.7±2.72 | 334.96±11.98 | **53.08±2.46** | 27.82±0.07 | 47.75±0.21 | 34.72±0.3 | **18.17±0.1** | 57.47±0.15 | 12.8±0.12 | **35.97±0.29** | 61.9±4.76 |
| | 0.1 | 209.37±3.5 | 197.24±0.41 | 321.69±7.22 | 51.68±2.37 | 27.82±0.05 | 48.01±0.33 | 34.78±0.27 | 17.92±0.09 | 57.54±0.05 | 12.87±0.27 | 35.8±0.3 | 52.38±4.76 |
| | 0.25 | 205.23±1.92 | 186.48±3.0 | 305.69±7.81 | 50.68±2.57 | 27.89±0.06 | 47.75±0.68 | 34.71±0.27 | 18.0±0.05 | 57.58±0.12 | 12.93±0.24 | 35.65±0.22 | 61.9±12.6 |
| | 0.5 | 196.93±3.28 | 171.2±5.77 | 287.82±7.34 | 49.94±2.05 | 28.05±0.04 | 47.51±0.65 | 34.81±0.29 | 18.15±0.08 | 57.89±0.19 | 13.2±0.2 | 35.65±0.2 | **76.19±12.6** |
| | 0.75 | 191.45±4.4 | 165.12±6.12 | 273.98±4.7 | 48.71±2.75 | 28.13±0.04 | **48.96±0.11** | 34.93±0.06 | 17.83±0.13 | 57.76±0.15 | 12.87±0.29 | 35.6±0.42 | 61.9±9.52 |
| | 0.9 | **188.59±2.32** | **157.61±4.94** | 271.39±6.26 | 48.47±2.83 | **28.21±0.0** | 48.28±0.51 | **35.06±0.11** | 17.78±0.32 | **57.91±0.39** | 13.2±0.12 | 35.56±0.5 | 71.43±0.0 |
| | 1.0 | 198.03±1.2 | 177.82±2.02 | **269.86±2.53** | 52.94±2.73 | 28.09±0.06 | 48.75±0.42 | 33.26±0.24 | 17.61±0.12 | 56.89±0.24 | 12.4±0.2 | 35.7±0.44 | - |
| 2:4 | 0.0 | 45.64±0.54 | 33.05±0.68 | 62.99±0.54 | 61.76±1.34 | 34.26±0.09 | 54.04±0.43 | 52.15±0.17 | 25.77±0.3 | 65.49±0.1 | 17.0±0.4 | 44.35±0.11 | 57.14±8.25 |
| | 0.1 | 45.64±0.47 | 33.01±0.45 | 62.91±0.69 | 61.72±0.95 | 34.25±0.13 | 54.56±0.29 | 52.27±0.38 | 25.65±0.16 | 65.52±0.27 | 16.67±0.37 | 44.38±0.08 | 52.38±12.6 |
| | 0.25 | 45.28±0.45 | 32.59±0.36 | 61.75±0.73 | 61.57±0.81 | 34.35±0.15 | 54.31±0.53 | 52.36±0.3 | **26.05±0.04** | 64.63±0.04 | 17.0±0.53 | 44.63±0.04 | 57.14±8.25 |
| | 0.5 | **45.1±0.26** | 32.47±0.5 | **61.19±0.23** | 61.6±0.86 | 34.41±0.16 | 55.51±0.66 | 52.53±0.18 | 25.6±0.38 | 65.4±0.17 | 17.2±0.61 | 44.61±0.09 | 61.9±12.6 |
| | 0.75 | 45.12±0.36 | **32.43±0.39** | 61.78±0.93 | 61.73±0.91 | 34.42±0.01 | 55.2±0.46 | **52.54±0.18** | 25.97±0.23 | 65.29±0.3 | 17.13±0.18 | 44.61±0.06 | **66.67±4.76** |
| | 0.9 | 45.39±0.18 | 32.59±0.22 | 61.41±0.16 | 61.94±0.56 | **34.49±0.07** | 55.33±0.43 | 52.22±0.22 | 26.02±0.23 | 65.58±0.22 | **17.33±0.33** | **44.7±0.08** | 61.9±4.76 |
| | 1.0 | 49.79±0.26 | 35.9±0.37 | 68.16±0.29 | **64.29±0.13** | 34.16±0.06 | **55.88±0.36** | 50.88±0.23 | 25.28±0.03 | 65.25±0.1 | 17.13±0.24 | **44.7±0.06** | - |

