# OpenReview forum: "MOONSHOT: A Framework for Multi-Objective Pruning of Vision and Large Language Models"
_TMLR — Accepted by TMLR_

### Review · Reviewer_EUyu · 2025-11-19

**Summary Of Contributions:**

This paper proposes Moonshot, a framework for pruning large neural models that optimises a multi-objective criterion combining 1) a layer-wise reconstruction loss, and 2) a second-order Taylor approximation of the pre-training (next-token prediction) loss; the idea is that Moonshot is a plug-and-play component that can be used jointly with existing pruning strategies, such as SparseGPT and Wanda (that rely on single-objective criteria). To make Moonshot practical, the authors propose some modelling choices that make computing their objective (which includes Hessian matrices) computationally tractable with models composed of several billion parameters. In their experiments, authors show that neither obj (1) nor (2) is better than the other, and weighting these two objectives using a hyper-parameter $\lambda$ can yield better performance-pruning trade-offs across several models and sparsity regimes. Authors experiment with Llama-3.2-{1,3}B (base) on C4, WikiText2/PTB, and several zero-shot classification benchmarks, DeiT models and ResNet-40 under unstructured and semi-structured sparsity regimes; their experiments show that Moonshot (used within SparseGPT and Wanda) can improve over the single-objective baselines without introducing prohibitive computational overhead (then why capping the LLM size to 3B?).

**Audience:**

Yes

**Audience Explanation:**

LLMs and VLMs have a significant computational footprint, and reducing, e.g. their parameter count can make it possible to deploy them on a wider variety of hardware. Identifying better pruning methods can have a significant real-world impact.

**Broader Impact Concerns:**

There's no ethical concern with this work. Finding better pruning strategies can reduce the energy and environmental costs of modern AI system.

**Claims And Evidence:**

Yes

**Claims Explanation:**

The main empirical claims are well supported.

**Requested Changes:**

- (Important) In this paper, you only consider tiny-sized ({1,3}B) base LLM -- why? Can the proposed method be applied to e.g. 8B+ models? Why the focus on base models? Can the approach be used on instruction fine-tuned version of such models? If you use base LLMs rather than instruction fine-tuned ones, aren't results on BoolQ, PIQA, HellaSWag, etc. significantly sub-par, since base models struggle to generalise in instruction following tasks?
- (Important) Is the comparison to the single-objective baselines fair? The multi-objective ones receive the extra advantage of having the extra hyper-parameter $\lambda$ to be tuned to maximise their performance -- isn't a comparison with "untuned" baselines problematic?
- Results in Tab. 3 seem very mixed -- e.g. we can often see that the untuned single-objective baselines often have an higher accuracy than their Moonshot version (BoolQ, HellaSwag, WinoGrande, ARC-c, OBQA); can you please expand on this? Shouldn't the Moonshot downstream performance be lower-bounded by the performance of the corresponding baseline pruner, since at worst $\lambda$ can recover that baseline? ($\lambda \in \{ 0, 1\}$)

---

> ### Author Response · Authors · 2026-01-09
> **MOONSHOT scales beyond 1 and 3B parameters**
>
> We thank the reviewer for their careful reading of the paper and comments.
>
> ## Results on Llama-2-13B-chat-hf
>
> The reviewer is correct that our main experiments focus on 1B and 3B models. However, MOONSHOT can be used for larger architectures. To demonstrate this, we apply MOONSHOT to Llama-2-13B-chat-hf, the instruction-tuned version of Llama-2-13B-hf, at 70\% sparsity using a single A100 (80GB). In this experiment, we fix $\lambda = 0.9$ for MOONSHOT. We report mean and standard error over 3 seeds.
> We obtain the following results:
>
> **Results for 13b-chat-hf**
>
> |Sparsity|Method|C4 ↓|WikiText2 ↓|PTB ↓|BoolQ ↑|HellaSwag ↑|WinoGrande ↑|ARC-e ↑|ARC-c ↑|PIQA ↑|OBQA ↑|Median ↑|
> |-|-|-|-|-|-|-|-|-|-|-|-|-|
> |Dense|–|8.49±0.00|6.11±0.00|50.36±0.00|81.65±0.00|60.71±0.00|71.11±0.00|77.53±0.00|46.16±0.00|77.91±0.00|35.20±0.00|71.11±0.00|
> ||
> |0.7|SparseGPT|27.62±0.31|24.87±0.41|460.75±16.25|71.86±1.18|38.59±0.10|59.93±0.27|53.65±1.11|**28.21±0.16**|67.16±0.54|**22.73±0.07**|53.65±1.11|
> |0.7|MOONSHOT-SparseGPT (0.9)|**23.67±0.06**|**20.95±0.59**|**368.57±5.75**|**75.71±0.50**|**40.01±0.27**|**61.12±0.42**|**57.41±0.72**|28.16±0.23|**68.64±0.28**|21.73±0.77|**57.41±0.72**|
> ||
> |0.7|Wanda|46.15±0.16|47.85±1.00|629.11±9.28|64.05±0.04|32.17±0.13|54.22±0.28|43.95±0.27|20.71±0.17|61.35±0.30|**17.00±0.12**|43.95±0.27|
> |0.7|MOONSHOT-Wanda (0.9)|**41.00±0.25**|**38.38±1.31**|**607.22±8.48**|**66.68±0.13**|**34.11±0.08**|**54.75±0.38**|**48.22±0.27**|**21.16±0.09**|**64.40±0.10**|14.53±0.52|**48.22±0.27**|
>
> At this scale we must adjust the implementation details for SparseGPT: when $\lambda \neq 1.0$ the Hessian becomes row-dependent, and storing the Hessian for all rows simultaneously is infeasible (for instance, it would require $\sim$ 536 GB in float32 for each of the $K$, $Q$, and $V$ matrices). For MOONSHOT, we therefore prune $K_p = 512$ rows at a time (Algorithm 2). As a consequence, the sparsity constraint is imposed on blocks of 512 rows.
>
> At 70\% sparsity, MOONSHOT reduces perplexity by up to 20\% on PTB and WikiText2 and 14\% on C4, and improves the zero-shot median accuracy on the seven classification benchmarks by up to 4.27 points (Wanda) and 3.76 points (SparseGPT).
>
> These experiments confirm that MOONSHOT is applicable to significantly larger models with standard hardware, and continues to provide consistent and significant improvements over strong pruning baselines.

---

> ### Author Response · Authors · 2026-01-09
> **MOONSHOT applies to instruction-tuned models (Part 1/2)**
>
> ## Results on Llama-3.2-1B-Instruct and Llama-3.2-3B-Instruct
>
> We now address the question of base vs. instruction-tuned models. In the main paper we follow the standard practices of prior pruning work: SparseGPT [1] prunes the OPT [2] models, OWL [3] considers the OPT [2] and LLaMA [4] models; Wanda [5] and AlphaPruning [6] focus on LLaMA and LLaMA-2 [7]. In all these cases, the evaluations are performed on base models.
>
> We agree with the reviewer that instruction-tuned models are highly relevant in practice. For this reason, beyond the Llama-2-13B-chat-hf results above, we ran new experiments on Llama-3.2-1B-Instruct and Llama-3.2-3B-Instruct. Again, we fix $\lambda = 0.9$ and evaluate at multiple sparsity levels:
>
> **Results for 1B-Instruct**
>
> |Sparsity|Method|C4 ↓|WikiText2 ↓|PTB ↓|BoolQ ↑|HellaSwag ↑|WinoGrande ↑|ARC-e ↑|ARC-c ↑|PIQA ↑|OBQA ↑|Median ↑|
> |-|-|-|-|-|-|-|-|-|-|-|-|-|
> |Dense|–|21.31±0.00|13.16±0.00|25.69±0.00|69.36±0.00|45.11±0.00|59.67±0.00|68.31±0.00|35.75±0.00|74.16±0.00|24.80±0.00|59.67±0.00|
> ||
> |0.5|SparseGPT|33.94±0.18|24.56±0.15|44.76±0.34|**63.98±0.35**|38.39±0.06|**55.93±0.13**|57.48±0.14|27.53±0.48|68.35±0.43|**22.27±0.13**|**55.93±0.13**|
> |0.5|MOONSHOT-SparseGPT (0.9)|**30.97±0.03**|**22.26±0.09**|**40.20±0.38**|63.74±0.15|**38.45±0.04**|55.12±0.78|**59.15±0.11**|**28.58±0.52**|**68.97±0.11**|20.93±0.27|55.12±0.78|
> ||
> |0.5|Wanda|46.50±0.11|33.63±0.04|59.43±0.11|62.62±0.18|35.77±0.08|**55.99±0.52**|54.05±0.24|25.97±0.34|65.89±0.06|**18.07±0.44**|54.05±0.24|
> |0.5|MOONSHOT-Wanda (0.9)|**40.16±0.17**|**28.28±0.03**|**52.16±0.18**|**63.06±0.27**|**36.34±0.00**|54.72±0.48|**56.96±0.05**|**26.28±0.20**|**66.74±0.13**|17.87±0.07|**54.72±0.48**|
> ||
> |0.6|SparseGPT|61.34±0.64|51.01±1.18|77.49±1.38|**62.57±0.05**|32.81±0.09|51.96±0.44|47.98±0.53|22.75±0.23|63.22±0.65|16.27±0.41|47.98±0.53|
> |0.6|MOONSHOT-SparseGPT (0.9)|**50.87±0.22**|**40.47±0.43**|**66.03±0.58**|62.46±0.13|**33.29±0.11**|**52.07±0.03**|**49.54±0.80**|**23.04±0.34**|**64.15±0.30**|**16.67±0.07**|**49.54±0.80**|
> ||
> |0.6|Wanda|154.55±0.94|129.18±1.52|141.38±1.56|61.73±0.09|29.62±0.02|**52.28±0.29**|39.69±0.29|19.80±0.64|59.16±0.26|**15.00±0.12**|39.69±0.29|
> |0.6|MOONSHOT-Wanda (0.9)|**99.91±1.39**|**77.19±1.06**|**105.98±1.66**|**62.31±0.06**|**30.29±0.07**|52.20±0.38|**43.74±0.21**|**20.42±0.21**|**60.57±0.07**|14.20±0.12|**43.74±0.21**|
> ||
> |0.7|SparseGPT|220.95±7.06|278.28±18.56|393.47±37.16|60.43±0.48|27.92±0.10|50.51±0.83|33.26±0.48|**19.11±0.26**|56.67±0.07|13.67±0.18|33.26±0.48|
> |0.7|MOONSHOT-SparseGPT (0.9)|**152.47±1.60**|**159.61±7.77**|**235.26±9.45**|**61.53±0.04**|**28.49±0.03**|**51.38±0.24**|**37.85±0.66**|19.08±0.16|**58.23±0.31**|**14.20±0.12**|**37.85±0.66**|
> ||
> |0.7|Wanda|1797.15±176.66|2120.64±117.11|2225.24±246.18|52.10±3.48|26.62±0.06|49.25±0.33|28.30±0.20|**18.26±0.20**|54.39±0.15|**13.00±0.23**|28.30±0.20|
> |0.7|MOONSHOT-Wanda (0.9)|**583.01±11.37**|**559.07±16.93**|**513.80±22.06**|**56.83±0.84**|**27.15±0.04**|**51.91±0.25**|**31.00±0.13**|17.72±0.11|**55.89±0.52**|12.73±0.33|**31.00±0.13**|

---

> ### Author Response · Authors · 2026-01-09
> **MOONSHOT applies to instruction-tuned models (Part 2/2)**
>
> **Results for 3B-Instruct**
>
> |Sparsity|Method|C4 ↓|WikiText2 ↓|PTB ↓|BoolQ ↑|HellaSwag ↑|WinoGrande ↑|ARC-e ↑|ARC-c ↑|PIQA ↑|OBQA ↑|Median ↑|
> |-|-|-|-|-|-|-|-|-|-|-|-|-|
> |Dense|–|16.49±0.00|11.04±0.00|20.42±0.00|78.47±0.00|52.24±0.00|67.40±0.00|73.99±0.00|43.43±0.00|75.79±0.00|27.40±0.00|67.40±0.00|
> ||
> |0.5|SparseGPT|23.11±0.05|17.73±0.26|30.01±0.16|**76.06±0.34**|**45.08±0.10**|**64.59±0.35**|64.52±0.21|34.22±0.60|71.16±0.27|**23.33±0.44**|**64.46±0.26**|
> |0.5|MOONSHOT-SparseGPT (0.9)|**22.37±0.02**|**17.04±0.14**|**29.24±0.27**|75.81±0.13|44.89±0.01|63.35±0.13|**66.16±0.06**|**34.33±0.14**|**71.65±0.05**|22.33±0.24|63.35±0.13|
> ||
> |0.5|Wanda|24.76±0.01|18.70±0.04|32.27±0.13|**74.28±0.34**|**43.43±0.06**|**63.85±0.16**|64.66±0.12|34.50±0.22|70.26±0.16|21.27±0.35|**63.85±0.16**|
> |0.5|MOONSHOT-Wanda (0.9)|**24.35±0.08**|**18.46±0.01**|**31.62±0.17**|73.86±0.26|43.20±0.01|63.59±0.09|**66.40±0.10**|**35.32±0.21**|**71.25±0.02**|**21.60±0.23**|63.59±0.09|
> ||
> |0.6|SparseGPT|41.20±0.10|38.64±0.82|61.84±1.86|69.73±0.98|36.61±0.18|59.30±0.54|53.27±0.16|26.22±0.32|65.16±0.23|17.27±0.57|53.27±0.16|
> |0.6|MOONSHOT-SparseGPT (0.9)|**37.31±0.18**|**33.40±0.50**|**53.89±1.03**|**70.37±1.18**|**36.90±0.12**|**59.67±0.75**|**55.57±0.16**|**27.90±0.47**|**65.96±0.40**|**17.33±0.52**|**55.57±0.16**|
> ||
> |0.6|Wanda|57.74±0.74|50.82±0.81|82.33±0.32|65.54±0.49|34.33±0.07|57.14±0.24|48.72±0.28|25.06±0.24|63.98±0.23|**16.67±0.07**|48.72±0.28|
> |0.6|MOONSHOT-Wanda (0.9)|**56.47±0.27**|**49.09±0.69**|**80.34±0.05**|**66.41±0.40**|**34.49±0.06**|**57.51±0.27**|**51.46±0.09**|**25.65±0.19**|**64.54±0.21**|16.20±0.35|**51.46±0.09**|
> ||
> |0.7|SparseGPT|136.30±0.13|156.05±2.00|193.42±6.92|**62.92±0.51**|28.22±0.14|52.12±0.61|34.69±0.09|18.86±0.13|57.29±0.35|**12.73±0.48**|34.69±0.09|
> |0.7|MOONSHOT-SparseGPT (0.9)|**123.54±1.30**|**131.66±1.76**|**175.18±11.33**|62.63±0.17|**28.83±0.05**|**52.33±0.83**|**35.69±0.46**|**19.20±0.44**|**58.14±0.20**|12.20±0.12|**35.69±0.46**|
> ||
> |0.7|Wanda|**234.03±3.39**|271.85±7.51|**264.91±5.09**|**55.83±0.19**|**27.42±0.04**|48.80±0.18|31.00±0.18|**17.83±0.25**|56.84±0.18|**12.20±0.20**|31.00±0.18|
> |0.7|MOONSHOT-Wanda (0.9)|255.35±1.85|**258.39±3.53**|268.83±4.87|45.51±1.63|27.38±0.04|**50.30±0.15**|**31.80±0.20**|17.78±0.19|**57.05±0.30**|11.47±0.13|**31.80±0.20**|
>
> In nearly all these experiments, MOONSHOT again achieves the best perplexity and provides significant gains in zero-shot median accuracy on the seven classification benchmarks: up to 4 points for Llama-3.2-1B-Instruct and nearly 3 points for Llama-3.2-3B-Instruct.
>
> However, we note that the instruction-tuned models, which are optimized to interact with human users, do not systematically outperform their base counterparts on these seven classification benchmarks.
>
> [1] E. Frantar and D. Alistarh. SparseGPT: Massive language models can be accurately pruned in one shot. arXiv:2301.00774, 2023.
>
> [2] S. Zhang et al. OPT: Open pre-trained transformer language models. arXiv:2205.01068, 2022.
>
> [3] L. Yin et al. Outlier Weighted Layerwise Sparsity (OWL): A missing secret sauce for pruning LLMs to high sparsity. ICML, 2024.
>
> [4] H. Touvron et al. LLaMA: Open and efficient foundation language models. arXiv:2302.13971, 2023.
>
> [5] M. Sun, Z. Liu, A. Bair, and J. Z. Kolter. A simple and effective pruning approach for large language models. arXiv:2306.11695, 2024.
>
> [6] H. Lu et al. AlphaPruning: Using heavy-tailed self-regularization theory for improved layer-wise pruning of large language models. NeurIPS, 2024.
>
> [7] H. Touvron et al. LLaMA 2: Open foundation and fine-tuned chat models. arXiv:2307.09288, 2023.

---

> ### Author Response · Authors · 2026-01-09
> **Protocol for selecting λ and its impact on results**
>
> ## MOONSHOT benefits from hyperparameter tuning, but does not rely on it
>
> We thank the Reviewer for raising the question of fairness regarding the additional hyperparameter $\lambda$. For the single-objective baselines, we use the hyperparameter settings recommended by the original authors, without any extra tuning on our side. For MOONSHOT, we additionally sweep over 7 values of $\lambda$ and report the best result. We view this as a realistic scenario: in practice, pruning is performed once and the resulting sparse model is reused across many downstream applications, so spending a small additional budget to select a good $\lambda$ is a reasonable choice for users. Accordingly, we report the best result over this modest 7-value sweep, reflecting a realistic setting where a small calibration budget is available.
>
> Importantly, our conclusions do not rely on heavy tuning of $\lambda$. As discussed at the end of Section 4.1 and illustrated in Figure 5 (Appendix A.7), any $\lambda \neq 0$ often yields substantial or near-optimal gains over the underlying single-objective method.
>
> We also observe in Figure 3 and 5 that simple default values work well in practice: $\lambda = 0.5$ is a strong choice, especially for vision models, and $\lambda = 0.9$ is frequently close to optimal for LLMs (and is the value we use in the experiments reported in this rebuttal). Thus, even if one fixes $\lambda$ to a default value and performs no tuning, MOONSHOT still provides consistent gains over the single-objective baselines. Tables 7--11 in the Appendix, which report MOONSHOT’s performance for each tested value of $\lambda$, show the effect of MOONSHOT in the regime where the user chooses to fix $\lambda$ rather than tune it.
>
> ## The best $\lambda$ is chosen based on training perplexity on C4 for the LLMs
>
> The Reviewer is correct that, by construction, MOONSHOT is a generalization of the underlying single-objective pruners: for layer-wise reconstruction methods such as SparseGPT and Wanda, setting $\lambda = 1.0$ in MOONSHOT recovers the original baseline. In that sense, for any model, there exists a value of $\lambda$ (namely $\lambda = 1$) whose performance at least matches the corresponding baseline.
>
> However, for vision models, we choose $\lambda$ based on training loss on ImageNet-1k, and for LLMs, we choose $\lambda$ based on training perplexity on C4. For the LLMs, the same $\lambda$ is then used for all downstream evaluations. Consequently, there is no guarantee that the $\lambda$ that optimizes in-sample C4 perplexity also maximizes accuracy on each downstream benchmark.
>
> In practice, however, we find that perplexity improvements obtained with MOONSHOT generally translate into better downstream performance, and that MOONSHOT improves the overall downstream performance.

---

### Review · Reviewer_PmGe · 2026-01-25

**Summary Of Contributions:**

This paper proposes MOONSHOT, a general framework for post-training one-shot pruning that extends existing pruning methods to a multi-objective formulation. Instead of optimizing a single criterion, MOONSHOT jointly minimizes the layer-wise reconstruction error and a second-order (Fisher/Taylor) approximation of the training loss, motivated by the empirical observation that neither objective is consistently optimal across models, sparsity levels, or domains.

MOONSHOT acts as a lightweight wrapper around state-of-the-art pruning methods (e.g., SparseGPT, Wanda, CAP, OBC) and preserves their original block-diagonal assumptions. To ensure scalability to large language models, the paper introduces an efficient inverse-Hessian computation using structured decompositions and the Woodbury identity. Extensive experiments on vision models and LLMs demonstrate consistent improvements in accuracy and perplexity across unstructured and semi-structured sparsity regimes, including when combined with non-uniform sparsity allocation strategies.

**Key Strengths**
- Clear motivation and empirical justification: The paper clearly motivates the one-shot pruning setting and convincingly argues for combining multiple pruning objectives.
- General and modular framework: MOONSHOT is not tied to a specific pruning algorithm and can be applied to multiple state-of-the-art methods with minimal changes.
- Empirical evaluation: The experimental results span vision and language domains, multiple architectures, sparsity regimes, and downstream tasks, and consistently demonstrate performance gains.
- Scalability-aware design: The proposed inverse Hessian computation is technically sound and explicitly designed to make the approach practical for large-scale LLMs.

**Key Weaknesses**
- Limited scale of evaluated LLMs: While MOONSHOT is designed to scale to large language models, the experimental evaluation is limited to relatively small LLMs (1B and 3B parameters). Although evaluating larger models is computationally demanding, this limits the strength of the claim that the framework is suitable for LLMs at larger scales.
- Additional hyperparameter tuning: The multi-objective formulation introduces a weighting parameter (λ), which requires tuning and adds complexity compared to single-objective baselines.
- Partial application in LLMs: For efficiency reasons, the multi-objective formulation is applied only to attention layers in large language models, which slightly weakens the conceptual completeness of the framework.

**Audience:**

Yes

**Audience Explanation:**

The paper addresses a well-established and practically relevant problem, post-training one-shot pruning of large models, and proposes a general framework that improves several widely used state-of-the-art methods. Its focus on scalability, applicability to both vision models and LLMs, and compatibility with existing pruning pipelines makes the findings relevant to researchers working on model compression, efficient inference, and deployment of large models, which aligns well with the interests of a substantial portion of the TMLR audience.

**Broader Impact Concerns:**

The paper does not raise significant ethical concerns beyond those commonly associated with model compression and efficiency research. The proposed method improves post-training pruning efficiency and can reduce the computational and environmental costs of deploying large models, which has a generally positive impact. At the same time, as with most advances in model efficiency, the technique could indirectly lower the barrier to deploying large-scale models in settings where misuse is possible. However, this risk is not specific to the proposed method and does not introduce new or unique ethical challenges beyond those already covered in the existing literature. Overall, no additional Broader Impact Statement appears necessary.

**Claims And Evidence:**

Yes

**Claims Explanation:**

The paper’s claims are supported by clear and generally convincing empirical evidence. The authors substantiate the complementarity of the two pruning objectives through controlled comparisons and demonstrate consistent improvements over strong baselines across vision and language models, multiple sparsity regimes, and evaluation metrics. Results are reported with multiple seeds and appropriate statistics. While scalability claims would be stronger with experiments on larger LLMs, the evidence is sufficient to support the conclusions within the evaluated setting.

**Requested Changes:**

1) **Empirical validation on larger LLMs (Critical)**:
The paper’s central claim is that MOONSHOT scales to modern large language models, yet the empirical results are limited to 1B- and 3B-parameter models. While the algorithmic design is sound, validating the approach on at least one larger-scale model (e.g., >=7B parameters) would significantly strengthen the credibility of the scalability claim.
2) **Clarify scope of multi-objective application in LLMs (Critical)**:
In the current implementation, the multi-objective formulation is applied only to attention layers, while projection layers are pruned using a single objective for efficiency reasons. This design choice should be more explicitly highlighted and discussed in the main text, including its implications for both conceptual completeness and empirical gains.
3) **Broader discussion of limitations and failure cases (Non-critical)**:
A brief discussion of scenarios in which MOONSHOT provides limited gains or in which one objective may dominate the other would improve transparency and help practitioners better understand when the framework is most beneficial.

---

> ### Author Response · Authors · 2026-02-08
> **MOONSHOT scales to models larger than 1–3B parameters**
>
> We thank the reviewer for their careful reading of the paper and comments.
>
> ## Results on Llama-2-13B-chat-hf
>
> We agree that experiments on larger models would strengthen the scalability claim. As indicated in our response to reviewer EUyu, although our experiments focus on Llama-3.2 models with 1 and 3 billion parameters, MOONSHOT can also be used for larger architectures. To demonstrate this, we apply MOONSHOT to Llama-2-13B-chat-hf, the instruct version of Llama-2-13B-hf. We prune this model to 70\% sparsity using a single A100 (80GB). We set $\lambda=0.9$ for MOONSHOT and report below the average performance with standard errors over 3 seeds:
>
> **Results for Llama-2-13b-chat-hf**
>
> |Sparsity|Method|C4↓|WikiText2↓|PTB↓|BoolQ↑|HellaSwag↑|WinoGrande↑|ARC-e↑|ARC-c↑|PIQA↑|OBQA↑|Median↑|Mean↑|
> |-|-|-:|-:|-:|-:|-:|-:|-:|-:|-:|-:|-:|-:|
> |Dense|-|8.49|6.11|50.36|81.65|60.71|71.11|77.53|46.16|77.91|35.20|71.11|64.32|
> |0.7|SparseGPT|27.62±0.31|24.87±0.41|460.75±16.25|71.86±1.18|38.59±0.10|59.93±0.27|53.65±1.11|**28.21±0.16**|67.16±0.54|**22.73±0.07**|53.65±1.11|48.88±0.28|
> |0.7|MOONSHOT-SparseGPT(0.9)|**23.67±0.06**|**20.95±0.59**|**368.57±5.75**|**75.71±0.50**|**40.01±0.27**|**61.12±0.42**|**57.41±0.72**|28.16±0.23|**68.64±0.28**|21.73±0.77|**57.41±0.72**|**50.40±0.30**|
> |0.7|Wanda|46.15±0.16|47.85±1.00|629.11±9.28|64.05±0.04|32.17±0.13|54.22±0.28|43.95±0.27|20.71±0.17|61.35±0.30|**17.00±0.12**|43.95±0.27|41.92±0.07|
> |0.7|MOONSHOT-Wanda(0.9)|**41.00±0.25**|**38.38±1.31**|**607.22±8.48**|**66.68±0.13**|**34.11±0.08**|**54.75±0.38**|**48.22±0.27**|**21.16±0.09**|**64.40±0.10**|14.53±0.52|**48.22±0.27**|**43.41±0.10**|
>
> At this scale, some implementation adjustments are necessary for SparseGPT for memory constraints.
>
> As mentioned in our response to Reviewer EUyu, when $\lambda \neq 1.0$ the Hessian becomes row-dependent, and storing the Hessian for all rows simultaneously is infeasible (e.g., $\sim$536GB in float32 for each of the $K$, $Q$, and $V$ matrices). For MOONSHOT, we therefore prune blocks of rows of size $K_p=512$ (Algorithm 2).
>
> Importantly, MOONSHOT continues to provide substantial benefits at 13B: up to 20\% lower perplexity on PTB and WikiText-2, and up to +4.27 points in median zero-shot accuracy.

---

> ### Author Response · Authors · 2026-02-08
> **Extending MOONSHOT beyond attention layers (Part 1/2)**
>
> ## MOONSHOT focuses on the attention layers for efficiency reasons
>
> We agree that this point should be made more explicit in the main paper. We will update the main text.
>
> Projection matrices are substantially larger in practice. For instance, the MLP down-projection is at least $4\times$ larger than the attention matrices in Llama-3.2-1B and $2.7\times$ larger in Llama-3.2-3B. Moreover, unlike the single-objective setting where the Hessian is block-diagonal with identical blocks repeated, $H=\mathrm{Diag}(XX^{\top},\ldots,XX^{\top})$, MOONSHOT with $\lambda \neq 1$ yields row-dependent blocks, $H=\mathrm{Diag}(F_1,\ldots,F_K)$. This  makes it infeasible to store all blocks simultaneously under GPU memory constraints, so we prune blocks of rows of size $K_p$ (Algorithm~2). For the down-projection layers, the Hessian blocks scale quadratically with the number of columns of $W$ and are therefore much larger than in the attention layers (e.g., $4^2=16\times$ larger for Llama-3.2-1B and $(2.7)^2\approx7.29\times$ larger for Llama-3.2-3B relative to $Q$). Consequently, $K_p$ must be reduced to fit memory, which increases runtime due to reduced parallelism. In particular, we use the following $K_p$:
>
> **Number $K_p$ of rows (Algorithm 2) pruned at the same time for MOONSHOT-SparseGPT ($\lambda \neq 1$).**
>
> |Layer|$K_p$ (Llama-3.2-1B)|$n_{\text{blocks}}$ (Llama-3.2-1B)|$K_p$ (Llama-3.2-3B)|$n_{\text{blocks}}$(Llama-3.2-3B)|
> |-|-:|-:|-:|-:|
> |`q_proj`|2048|2048|1536|3072|
> |`k_proj`|512|512|512|512|
> |`v_proj`|512|512|512|512|
> |`o_proj`|2048|2048|1536|3072|
> |`gate_proj`|1024|8192|1024|8192|
> |`up_proj`|1024|8192|1024|8192|
> |`down_proj`|64|2048|192|3072|
>
> We ran additional experiments pruning both the attention and projection layers with MOONSHOT. The results are provided below:
>
> **Results for Llama-3.2-1B**
>
> |Sparsity|Method|C4↓|WikiText2↓|PTB↓|BoolQ↑|HellaSwag↑|WinoGrande↑|ARC-e↑|ARC-c↑|PIQA↑|OBQA↑|Median↑|Mean↑|
> |-:|-|-:|-:|-:|-:|-:|-:|-:|-:|-:|-:|-:|-:|
> |0.6|SparseGPT|63.63±1.18|54.60±1.00|81.11±3.99|60.67±0.59|32.16±0.20|**54.46±0.53**|44.94±0.11|21.47±0.48|62.21±0.20|**17.07±0.41**|44.94±0.11|41.85±0.20|
> |0.6|MOONSHOT-SparseGPT|50.28±1.99|39.13±1.54|60.14±2.90|**62.36±0.12**|32.49±0.13|53.09±0.18|46.49±0.38|21.30±0.24|63.22±0.17|15.73±0.55|46.49±0.38|42.10±0.11|
> |0.6|MOONSHOT(all)-SparseGPT|**42.96±0.27**|**34.29±0.46**|**54.92±1.22**|62.09±0.13|**32.88±0.08**|53.14±0.78|**46.87±0.14**|**21.50±0.57**|**63.73±0.43**|16.93±0.44|**46.87±0.14**|**42.45±0.22**|
> |0.6|Wanda|117.71±0.87|84.73±0.73|119.64±1.00|58.96±1.39|28.86±0.03|51.35±0.49|38.82±0.32|18.94±0.26|59.05±0.18|**13.93±0.24**|38.82±0.32|38.56±0.12|
> |0.6|MOONSHOT-Wanda|86.55±1.67|63.57±1.61|98.44±3.98|61.56±0.29|29.53±0.06|51.64±0.23|40.40±0.17|**19.60±0.06**|61.12±0.08|13.40±0.20|40.40±0.17|39.61±0.07|
> |0.6|MOONSHOT(all)-Wanda|**85.66±0.92**|**63.12±0.29**|**94.39±1.78**|**61.81±0.18**|**29.62±0.05**|**51.88±0.38**|**41.27±0.22**|19.11±0.13|**61.19±0.17**|12.73±0.47|**41.27±0.22**|**39.66±0.06**|

---

> ### Author Response · Authors · 2026-02-08
> **Extending MOONSHOT beyond attention layers (Part 2/2)**
>
> **Results for Llama-3.2-3B**
>
> |Sparsity|Method|C4↓|WikiText2↓|PTB↓|BoolQ↑|HellaSwag↑|WinoGrande↑|ARC-e↑|ARC-c↑|PIQA↑|OBQA↑|Median↑|Mean↑|
> |-:|-|-:|-:|-:|-:|-:|-:|-:|-:|-:|-:|-:|-:|
> |0.6|SparseGPT|33.63±0.14|26.12±0.23|42.69±0.73|66.82±0.60|38.14±0.14|60.91±0.65|53.89±0.11|26.28±0.18|67.75±0.34|18.47±0.44|53.89±0.11|47.47±0.08|
> |0.6|MOONSHOT-SparseGPT|28.23±0.11|22.46±0.17|35.63±0.68|**67.76±0.35**|39.13±0.07|61.01±0.23|57.59±0.92|**27.79±0.71**|69.44±0.13|**20.00±0.53**|57.59±0.92|**48.96±0.12**|
> |0.6|MOONSHOT(all)-SparseGPT|**26.26±0.12**|**20.67±0.16**|**33.01±0.86**|65.66±1.76|**39.60±0.06**|**61.19±0.43**|**58.25±0.79**|27.45±1.13|**69.57±0.07**|**20.00±0.31**|**58.25±0.79**|48.82±0.38|
> |0.6|Wanda|41.98±0.40|30.56±0.32|51.00±0.45|**64.82±0.35**|35.12±0.07|**56.56±0.46**|50.58±0.41|23.83±0.12|65.58±0.19|16.93±0.07|50.58±0.41|**44.77±0.10**|
> |0.6|MOONSHOT-Wanda|**37.73±0.19**|**27.71±0.26**|**46.47±0.08**|61.33±0.91|35.53±0.08|54.83±0.14|**52.53±0.29**|24.69±0.21|66.81±0.03|16.60±0.23|**52.53±0.29**|44.62±0.12|
> |0.6|MOONSHOT(all)-Wanda|37.83±0.04|27.84±0.18|47.37±0.57|60.40±0.51|**35.54±0.07**|56.30±0.21|52.24±0.17|**24.72±0.16**|**66.85±0.35**|**17.20±0.23**|52.24±0.17|44.75±0.14|
>
> A single L40 (40GB) was used for Llama-3.2-1B and a single A100 (80GB) for Llama-3.2-3B.
>
> Empirically, applying MOONSHOT to projection layers too (`gate_proj`, `up_proj`, `down_proj`) yield additional gains.
> With the exception of Wanda on Llama-3.2-3B, we observe consistent additional improvements when extending MOONSHOT to projection layers, indicating that the multi-objective formulation is beneficial beyond the attention blocks. However, pruning time increases by $8 \times$ for Wanda and $12\times$ for SparseGPT on Llama-3.2-1B. For Llama-3.2-3B, pruning time increases by $8 \times$ for Wanda and $6\times$ for SparseGPT. This increase is due to the smaller feasible $K_p$ and the increased computations with the larger Hessian.
>
> Pruning projection layers with MOONSHOT is thus a practical compute/memory trade-off: depending on available resources, users may choose to apply MOONSHOT only to attention layers for efficiency, or extend it to projection layers for additional performance gains. Since pruning is typically performed once as an offline step, the extra runtime can be justified when resources permit, given the consistent improvements we observe.

---

> ### Author Response · Authors · 2026-02-08
> **The multi-objective formulation is a stronger choice than either single objective.**
>
> ## Broader discussion: when MOONSHOT helps most / least
>
> We agree that it is useful to clarify when MOONSHOT provides the largest improvements and whether either objective can dominate.
> Empirically, across all settings we tested, the multi-objective formulation always improved perplexity for LLMs and accuracy for vision, and we did not observe qualitative failure modes. While it is not possible to predict in advance the impact of improved perplexity on downstream classification tasks for LLMs, we observe in practice a positive correlation and an improved performance with MOONSHOT. In addition, we observe that MOONSHOT is most beneficial at higher sparsity and in regimes where single-objective baselines incur the largest drops (Section 4.2).
>
> It is difficult to determine a priori which single objective will be most effective for a given model, sparsity level, and pruning algorithm; this motivates using the multi-objective formulation as a robust and stronger alternative.

---

> > ### Comment · Reviewer_PmGe · 2026-02-15
> >
> > Thank you for the detailed and constructive rebuttal and for providing additional experiments and clarifications.
> >
> > My main concern about the limited scale of evaluated LLMs has been substantially addressed by additional results on Llama-2-13B, which strengthen the empirical support for scalability claims. I also appreciate the clarification and the additional experiments on applying MOONSHOT beyond attention layers, which address my concerns about the partial use of the framework in LLMs. The added discussion on when MOONSHOT is most beneficial further improves the presentation of the work.
> >
> > While I still see some room for improvement in parts of the paper (e.g., evaluation of models from a different model family), the rebuttal has meaningfully strengthened the submission and addressed my main concerns. Based on these revisions, I am happy to recommend acceptance of the paper.

---

### Review · Reviewer_488S · 2026-01-26

**Summary Of Contributions:**

This paper makes an interesting observation that depending on the architecture, pruning method, and sparsity level, either the layer-wise reconstruction error or second-order Taylor approximation of the training-loss objective may yield better results. This motivates the proposal of "Multi-objective pruning", which conbines the two objective in an unified optimization framework.

Strength:
1. The observation of the performance difference between two major post-training reconstruction objective is novel and significant
2. The proposed method is technically solid. The derivation is clear and correct.
3. Results show consistent improvement compared to baseline.

Weakness:
The main issue needs to be revised is the reporting of zero-shot downstream performance. See requested changes for details

**Audience:**

Yes

**Audience Explanation:**

The paper works on LLM efficiency improvement via post-training pruning. This work is interesting to researchers in LLM application and deployment domains and efficient deep learning domain in general.

**Broader Impact Concerns:**

No broader impact concerns

**Claims And Evidence:**

Yes

**Claims Explanation:**

This work provides detailed theoreitical derivations, ablation studies, and main results to support the idea of multi-objective pruning. The work's claims are well-supported.

**Requested Changes:**

1. As the author identified 3 types of pruning, the paper only addresses unstructured and semi-structured pruning. Meanwhile, I see no reason why the proposed method cannot generalize to structured pruning. It would be interesting to see how the proposed method work on structured pruning, even at a lower pruning ratio.
2. Table 3 uses median to highlight zeroshot performance, which is not a meaningful number. Given that different tasks have significantly different accurayc range, the reported medians are mostly the performance of a single task, which is not meaningful to report seperatly since task-specific performance are also in the table. Reporting averaged accuracy is a more common practice in previous work. Also, it would be interesting to add some analysis if the performance of a specific down stream task sees a significant gain or loss when using the proposed method.

---

> ### Author Response · Authors · 2026-02-08
> **Using MOONSHOT for structured pruning**
>
> We thank the reviewer for their careful reading of the paper and comments.
>
> ## Adapting OSSCAR with MOONSHOT.
>
> We agree with the reviewer: although we focus on unstructured and semi-structured sparsity in the paper, MOONSHOT can be used for structured pruning as well.
>
> To illustrate this, we adapt OSSCAR [1], a state-of-the-art structured pruning method that greedily selects columns to remove using a layer-wise reconstruction objective (with an optional local-search refinement, which we do not use in this experiment).
> We replace the original layer-wise reconstruction objective with the MOONSHOT convex combination, and follow the same Hessian construction protocol described in Figure 2.
> As in our SparseGPT implementation (Algorithm 2), we prune rows by blocks of size $K_p$ for memory reasons. We chose $K_p$ so that 50\% of the rows are pruned at a time for Llama-3.2-1B and 25\% for Llama-3.2-3B.
>
> OSSCAR prunes only the `down_proj` and `o_proj` matrices. For efficiency reasons (since `down_proj` is substantially larger), we evaluate the impact of the convex combination on `o_proj`  only and use the single-objective for `down_proj`.
>
> Strong gains are similarly observed in the case of structured-pruning:
>
> **Results for Llama-3.2-1B**
>
> |Sparsity|Method|C4↓|WikiText2↓|PTB↓|BoolQ↑|HellaSwag↑|WinoGrande↑|ARC-e↑|ARC-c↑|PIQA↑|OBQA↑|Median↑|Mean↑|
> |-:|-|-:|-:|-:|-:|-:|-:|-:|-:|-:|-:|-:|-:|
> |0.1|OSSCAR|43.00±1.28|43.91±0.60|106.62±8.57|52.61±1.63|35.82±0.29|54.17±1.11|45.19±7.83|24.23±2.04|65.65±3.68|15.93±0.33|42.95±4.71|41.94±1.98|
> |0.1|MOONSHOT-OSSCAR|**38.04±0.58**|**30.93±0.92**|**86.73±9.88**|**56.55±0.90**|**37.89±1.11**|**55.51±0.37**|**58.84±0.38**|**28.58±0.51**|**71.49±0.06**|**18.93±0.48**|**55.46±0.41**|**46.83±0.42**|
> |0.15|OSSCAR|52.65±1.85|54.51±1.35|132.62±9.26|**51.23±2.07**|35.22±0.32|52.83±0.43|45.61±6.45|24.23±1.72|66.12±2.69|14.53±0.55|43.62±4.52|41.40±1.74|
> |0.15|MOONSHOT-OSSCAR|**47.48±1.15**|**46.86±1.09**|**102.39±3.55**|48.88±2.07|**35.65±0.50**|**54.22±0.32**|**46.49±6.13**|**25.17±1.70**|**66.74±2.73**|**15.93±0.74**|**44.62±4.40**|**41.87±1.83**|
>
>
> **Results for Llama-3.2-3B**
>
> |Sparsity|Method|C4↓|WikiText2↓|PTB↓|BoolQ↑|HellaSwag↑|WinoGrande↑|ARC-e↑|ARC-c↑|PIQA↑|OBQA↑|Median↑|Mean↑|
> |-:|-|-:|-:|-:|-:|-:|-:|-:|-:|-:|-:|-:|-:|
> |0.1|OSSCAR|16.80±0.46|13.76±0.85|22.34±0.43|64.38±0.86|49.03±0.86|61.30±0.74|65.82±0.64|34.10±0.53|**75.70±0.78**|**27.33±0.41**|61.30±0.74|53.95±0.08|
> |0.1|MOONSHOT-OSSCAR|**16.56±0.38**|**13.28±0.55**|**22.10±0.28**|**64.91±1.00**|**49.50±0.62**|**61.43±0.47**|**66.39±0.83**|**34.41±0.33**|75.54±0.45|26.80±0.76|**61.43±0.47**|**54.14±0.20**|
> |0.15|OSSCAR|22.52±0.58|21.44±1.93|31.70±0.68|63.39±2.33|47.11±0.78|57.64±2.02|61.94±1.86|30.94±1.25|73.83±0.68|23.27±1.38|57.64±2.02|51.16±1.31|
> |0.15|MOONSHOT-OSSCAR|**21.50±0.22**|**18.28±0.41**|**27.99±1.03**|**64.34±2.28**|**48.37±0.21**|**58.77±1.44**|**65.40±0.71**|**33.16±0.67**|**74.77±0.55**|**25.00±1.15**|**58.77±1.44**|**52.83±0.84**|
>
> We observe up to 30\% lower perplexity on WikiText2, 22\% on PTB, and 11\% on C4, along with +4.9 points in mean accuracy, supporting that the benefits from the convex-combination extends to structured pruning as well.
>
> Adapting OSSCAR to the multi-objective formulation requires some algorithmic adaptations, which we describe in Appendix A.4 of the revised paper.
>
> [1] X. Meng, S. Ibrahim, K. Behdin, H. Hazimeh, N. Ponomareva, and R. Mazumder. OSSCAR: One-shot structured pruning in vision and language models with combinatorial optimization. ICML, 2024.

---

> ### Author Response · Authors · 2026-02-08
> **Replacing median accuracy with mean accuracy (Part 1/2)**
>
> ## MOONSHOT shows substantial improvements in mean accuracy
>
> We thank the reviewer for pointing this out. We agree that reporting median accuracy is non-standard, and we acknowledge the issues the reviewer highlighted.
>
> We will replace the median with the mean accuracy (the common metric in prior work).
> In addition to reporting mean accuracy, we also report a per-task win rate: the fraction of tasks on which one method outperforms the other. We compute win rate for each seed and report the mean and standard error over three seeds.
>
> Below, we provide the updated main tables with both mean accuracy and win rate.
>
> **Results for Llama-3.2-1B. The mean accuracy and win rate are calculated across all seven classification datasets.**
>
> |Sparsity|Method|C4↓|WikiText2↓|PTB↓|BoolQ↑|HellaSwag↑|WinoGrande↑|ARC-e↑|ARC-c↑|PIQA↑|OBQA↑|Mean↑|WinRate↑|
> |-:|-|-:|-:|-:|-:|-:|-:|-:|-:|-:|-:|-:|-:|
> |Dense|-|14.02|9.75|17.59|64.01|47.73|60.14|65.15|31.23|74.32|26.4|52.71|-|
> |0.6|Sparsegpt|63.63±1.18|54.60±1.00|81.11±3.99|60.67±0.59|32.16±0.20|**54.46±0.53**|44.94±0.11|**21.47±0.48**|62.21±0.20|**17.07±0.41**|41.85±0.20|42.86±8.25|
> |0.6|MOONSHOT-Sparsegpt|**50.28±1.99**|**39.13±1.54**|**60.14±2.90**|**62.36±0.12**|**32.49±0.13**|53.09±0.18|**46.49±0.38**|21.30±0.24|**63.22±0.17**|15.73±0.55|**42.10±0.11**|**57.14±8.25**|
> |0.6 (Alpha-Pruning)|Sparsegpt|61.05±0.77|52.80±0.43|78.27±3.64|62.08±0.26|32.00±0.08|**53.88±0.25**|45.29±0.49|**22.01±0.59**|62.02±0.42|**17.60±0.42**|42.13±0.06|33.33±9.52|
> |0.6 (Alpha-Pruning)|MOONSHOT-Sparsegpt|**49.31±1.10**|**38.44±0.52**|**60.32±1.20**|**62.29±0.05**|**32.53±0.06**|53.70±0.55|**46.30±0.30**|21.99±0.15|**63.13±0.10**|16.60±0.12|**42.36±0.14**|**66.67±9.52**|
> |0.6 (OWL)|Sparsegpt|56.82±1.68|49.54±1.22|68.73±3.48|**62.20±0.11**|32.86±0.05|**53.67±0.33**|44.14±0.71|**23.46±0.05**|62.59±0.21|**18.00±0.76**|42.42±0.07|33.33±17.17|
> |0.6 (OWL)|MOONSHOT-Sparsegpt|**43.58±0.91**|**35.72±0.17**|**53.02±0.91**|62.20±0.04|**33.66±0.16**|53.54±0.55|**46.37±0.23**|23.41±0.08|**64.04±0.03**|16.40±0.20|**42.80±0.07**|**66.67±17.17**|
> |2:4|Sparsegpt|53.59±0.35|42.56±0.37|63.79±0.19|61.42±0.21|**31.68±0.05**|**53.83±0.37**|44.04±0.23|**21.47±0.44**|61.79±0.40|**15.00±0.20**|**41.32±0.08**|**57.14±14.29**|
> |2:4|MOONSHOT-Sparsegpt|**50.99±0.47**|**38.00±0.58**|**59.32±1.55**|**61.98±0.29**|31.47±0.21|53.09±0.27|**45.37±0.50**|20.28±0.58|**62.13±0.19**|14.33±0.35|41.24±0.20|42.86±14.29|
> |0.6|Wanda|117.71±0.87|84.73±0.73|119.64±1.00|58.96±1.39|28.86±0.03|51.35±0.49|38.82±0.32|18.94±0.26|59.05±0.18|**13.93±0.24**|38.56±0.12|19.05±4.76|
> |0.6|MOONSHOT-Wanda|**86.55±1.67**|**63.57±1.61**|**98.44±3.98**|**61.56±0.29**|**29.53±0.06**|**51.64±0.23**|**40.40±0.17**|**19.60±0.06**|**61.12±0.08**|13.40±0.20|**39.61±0.07**|**80.95±4.76**|
> |0.6 (Alpha-Pruning)|Wanda|112.33±1.03|80.75±1.03|120.89±0.73|58.01±1.10|28.92±0.09|51.22±0.47|37.56±0.16|19.51±0.21|59.32±0.04|**13.40±0.40**|38.28±0.10|14.29±8.25|
> |0.6 (Alpha-Pruning)|MOONSHOT-Wanda|**84.21±1.04**|**63.30±0.92**|**101.96±2.98**|**61.69±0.24**|**29.56±0.08**|**52.33±0.37**|**39.28±0.29**|**19.65±0.06**|**60.32±0.13**|12.07±0.58|**39.27±0.09**|**85.71±8.25**|
> |0.6 (OWL)|Wanda|99.38±1.37|73.00±0.37|111.24±1.83|61.28±0.28|29.88±0.11|**52.07±0.82**|40.22±0.09|**20.82±0.05**|60.03±0.20|**14.80±0.12**|**39.87±0.17**|33.33±4.76|
> |0.6 (OWL)|MOONSHOT-Wanda|**73.97±0.19**|**58.81±0.28**|**100.07±1.96**|**61.81±0.08**|**30.47±0.07**|51.51±0.37|**40.92±0.22**|20.71±0.15|**60.36±0.19**|13.27±0.35|39.86±0.15|**66.67±4.76**|
> |2:4|Wanda|164.32±2.37|114.73±2.32|190.58±1.50|57.28±1.00|28.32±0.03|**51.51±0.15**|35.76±0.26|18.34±0.09|58.41±0.35|**13.67±0.07**|37.61±0.10|28.57±0.00|
> |2:4|MOONSHOT-Wanda|**110.74±1.17**|**78.55±1.14**|**126.91±1.66**|**61.08±0.57**|**28.51±0.05**|50.62±0.25|**38.41±0.26**|**19.43±0.08**|**59.36±0.33**|12.67±0.37|**38.58±0.14**|**71.43±0.00**|

---

> ### Author Response · Authors · 2026-02-08
> **Replacing median accuracy with mean accuracy (Part 2/2)**
>
> **Results for Llama-3.2-3B. The mean accuracy and win rate are calculated across all seven classification datasets.**
>
> |Sparsity|Method|C4↓|WikiText2↓|PTB↓|BoolQ↑|HellaSwag↑|WinoGrande↑|ARC-e↑|ARC-c↑|PIQA↑|OBQA↑|Mean↑|WinRate↑|
> |-:|-|-:|-:|-:|-:|-:|-:|-:|-:|-:|-:|-:|-:|
> |Dense|-|11.34|7.81|13.54|72.72|55.30|69.22|74.37|42.41|76.71|31.20|60.28|-|
> |0.6|Sparsegpt|33.63±0.14|26.12±0.23|42.69±0.73|66.82±0.60|38.14±0.14|60.91±0.65|53.89±0.11|26.28±0.18|67.75±0.34|18.47±0.44|47.47±0.08|4.76±4.76|
> |0.6|MOONSHOT-Sparsegpt|**28.23±0.11**|**22.46±0.17**|**35.63±0.68**|**67.76±0.35**|**39.13±0.07**|**61.01±0.23**|**57.59±0.92**|**27.79±0.71**|**69.44±0.13**|**20.00±0.53**|**48.96±0.12**|**95.24±4.76**|
> |0.6 (Alpha-Pruning)|Sparsegpt|34.19±0.40|26.06±0.64|42.82±0.13|68.17±0.43|38.62±0.16|61.98±0.81|53.37±0.70|26.00±0.23|68.06±0.38|19.80±0.90|48.00±0.38|14.29±8.25|
> |0.6 (Alpha-Pruning)|MOONSHOT-Sparsegpt|**28.64±0.25**|**22.17±0.19**|**35.48±1.52**|**68.73±0.11**|**39.34±0.18**|**62.27±0.52**|**56.52±0.25**|**26.93±0.37**|**69.13±0.32**|**20.33±0.24**|**49.04±0.06**|**85.71±8.25**|
> |0.6 (OWL)|Sparsegpt|29.15±0.31|23.58±0.40|36.58±0.78|66.64±0.77|39.89±0.16|**61.96±0.40**|55.57±0.41|27.53±0.12|68.72±0.14|**21.20±0.35**|48.79±0.11|28.57±8.25|
> |0.6 (OWL)|MOONSHOT-Sparsegpt|**25.31±0.12**|**20.85±0.11**|**31.39±0.69**|**67.41±0.47**|**40.64±0.13**|61.62±0.09|**57.39±0.78**|**28.16±0.62**|**69.73±0.29**|20.60±0.50|**49.36±0.17**|**71.43±8.25**|
> |2:4|Sparsegpt|30.00±0.29|24.40±0.23|38.32±0.74|**65.64±0.70**|**38.31±0.08**|**59.93±0.13**|55.63±1.10|26.14±0.42|**68.34±0.33**|20.87±0.35|**47.83±0.18**|**52.38±9.52**|
> |2:4|MOONSHOT-Sparsegpt|**28.79±0.29**|**23.21±0.32**|**35.94±0.73**|65.58±0.08|38.06±0.13|59.30±0.41|**55.99±0.63**|**26.56±0.37**|67.94±0.21|**20.93±0.55**|47.77±0.18|47.62±9.52|
> |0.6|Wanda|41.98±0.40|30.56±0.32|51.00±0.45|**64.82±0.35**|35.12±0.07|**56.56±0.46**|50.58±0.41|23.83±0.12|65.58±0.19|**16.93±0.07**|**44.77±0.10**|38.10±4.76|
> |0.6|MOONSHOT-Wanda|**37.73±0.19**|**27.71±0.26**|**46.47±0.08**|61.33±0.91|**35.53±0.08**|54.83±0.14|**52.53±0.29**|**24.69±0.21**|**66.81±0.03**|16.60±0.23|44.62±0.12|**61.90±4.76**|
> |0.6 (Alpha-Pruning)|Wanda|40.03±0.04|29.19±0.20|50.24±0.27|**65.93±0.24**|35.95±0.01|**57.51±0.09**|51.09±0.16|24.32±0.36|66.03±0.15|**16.87±0.24**|45.39±0.03|38.10±4.76|
> |0.6 (Alpha-Pruning)|MOONSHOT-Wanda|**37.37±0.21**|**27.08±0.16**|**47.01±0.29**|63.38±0.65|**36.05±0.10**|57.51±0.56|**54.15±0.27**|**25.43±0.18**|**66.96±0.36**|16.27±0.18|**45.68±0.10**|**61.90±4.76**|
> |0.6 (OWL)|Wanda|37.35±0.14|27.93±0.23|44.25±0.74|**67.26±0.07**|37.18±0.13|**59.06±0.73**|51.89±0.34|25.54±0.23|66.47±0.10|**17.20±0.12**|**46.37±0.18**|33.33±9.52|
> |0.6 (OWL)|MOONSHOT-Wanda|**34.56±0.11**|**25.59±0.06**|**40.41±0.78**|63.73±0.92|**37.40±0.12**|58.93±0.35|**54.31±0.07**|**25.68±0.13**|**67.05±0.25**|17.00±0.23|46.30±0.13|**66.67±9.52**|
> |2:4|Wanda|49.79±0.26|35.90±0.37|68.16±0.29|**64.29±0.13**|34.16±0.06|**55.88±0.36**|50.88±0.23|25.28±0.03|65.25±0.10|17.13±0.24|**44.70±0.06**|38.10±12.60|
> |2:4|MOONSHOT-Wanda|**45.10±0.26**|**32.47±0.50**|**61.19±0.23**|61.60±0.86|**34.41±0.16**|55.51±0.66|**52.53±0.18**|**25.60±0.38**|**65.40±0.17**|**17.20±0.61**|44.61±0.09|**61.90±12.60**|
>
> MOONSHOT remains competitive in mean accuracy. In some settings, the baseline has a slightly higher mean accuracy, but the differences are small and not statistically significant. In contrast, MOONSHOT significantly improves over the baseline in many cases, with gains of up to 1.49 points.
> In terms of win rate, MOONSHOT shows clear gains. In this table, MOONSHOT achieves higher win rates across nearly all pruning algorithms and sparsity types.

---

### Author Response · Authors · 2026-02-09

We thank the reviewers for their detailed and constructive feedback. We have updated the paper accordingly and provide a revised version in which all requested changes are highlighted in blue.

As noted in the reviews, MOONSHOT is a framework for one-shot pruning of LLMs and vision models. Unlike existing pruning methods that optimize either the layer-wise reconstruction error or a second-order Taylor approximation of the training loss, MOONSHOT minimizes both objectives jointly via a multi-objective formulation. MOONSHOT acts as an efficient wrapper around state-of-the-art pruning methods and consistently improves their performance.

We have addressed all reviewer comments. Specifically, we applied MOONSHOT to the instruct-tuned versions of Llama-3.2-1B and Llama-3.2-3B (Reviewer EUyu), and to a model larger than 8B, Llama-2-13b-chat-hf (Reviewers EUyu and PmGe). We also clarified the tuning of $\lambda$ (Reviewer EUyu) and the use of MOONSHOT on projection matrices for LLMs (Reviewer PmGe). Finally, we adapted MOONSHOT to structured pruning (Reviewer 488S) and replaced median accuracy with mean accuracy (Reviewer 488S).

The additional experiments we conducted confirm that MOONSHOT yields significant performance improvements across model sizes, sparsity types, and single-objective one-shot pruning baselines.

---

### Decision · Action_Editor_R7kz · 2026-05-14

**Recommendation:** Accept as is

**Additional Comments:**

The paper is technically sound, clearly motivated, and relevant to the TMLR audience. The main idea is to use a convex combination of two losses previously considered in the literature, parametrized by a tunable parameter lambda. The reviewers raised important concerns about the method's scalability claims and the experimental protocol for lambda tuning. The authors addressed these concerns well through additional experiments and clarifications, and after the revision, all reviewers are in favor of accepting the paper.

**Audience:**

Yes

**Audience Explanation:**

The submission addresses post-training one-shot pruning, an important problem for model compression, efficient inference, and deployment of large neural networks. The proposed framework is relevant to researchers working on pruning and efficient deep learning.

**Claims And Evidence:**

Yes

**Claims Explanation:**

The main claims are supported by a clear technical formulation and an empirical evaluation. The revised manuscript substantially strengthened the evidence by adding results on Llama-2-13B-chat-hf, additional experiments on Llama-3.2 instruct models, extra experiments/clarifications on applying the proposed pruning method beyond attention layers, and clarifications about the lambda selection and computational trade-offs.